# Three-dimensional view of ultrafast dynamics in photoexcited bacteriorhodopsin

Gabriela Nass Kovacs[1], Jacques-Philippe Colletier[2], Marie Luise Grünbein[1], Yang Yang[3], Till Stensitzki [3], Alexander Batyuk [4], Sergio Carbajo[4], R. Bruce Doak[1], David Ehrenberg[3], Lutz Foucar[1], Raphael Gasper[5], Alexander Gorel[1], Mario Hilpert[1], Marco Kloos[1], Jason E. Koglin [4], Jochen Reinstein[1], Christopher M. Roome[1], Ramona Schlesinger [3], Matthew Seaberg [4], Robert L. Shoeman [1], Miriam Stricker [1], Sébastien Boutet[4], Stefan Haacke [6], Joachim Heberle[3], Karsten Heyne[3], Tatiana Domratcheva[1], Thomas R.M. Barends[1] & Ilme Schlichting [1]

Bacteriorhodopsin (bR) is a light-driven proton pump. The primary photochemical event upon light absorption is isomerization of the retinal chromophore. Here we used time-resolved crystallography at an X-ray free-electron laser to follow the structural changes in multiphoton-excited bR from 250 femtoseconds to 10 picoseconds. Quantum chemistry and ultrafast spectroscopy were used to identify a sequential two-photon absorption process, leading to excitation of a tryptophan residue flanking the retinal chromophore, as a first manifestation of multiphoton effects. We resolve distinct stages in the structural dynamics of the all-*trans* retinal in photoexcited bR to a highly twisted 13-*cis* conformation. Other active site sub-picosecond rearrangements include correlated vibrational motions of the electronically excited retinal chromophore, the surrounding amino acids and water molecules as well as their hydrogen bonding network. These results show that this extended photo-active network forms an electronically and vibrationally coupled system in bR, and most likely in all retinal proteins.

[1] Max-Planck-Institut für Medizinische Forschung, Jahnstraße 29, 69120 Heidelberg, Germany. [2] Univ. Grenoble Alpes, CNRS, CEA, Institut de Biologie Structurale, 71 Avenue des Martyrs, 38000 Grenoble, France. [3] Freie Universität Berlin, Department of Physics, Arnimallee 14, 14195 Berlin, Germany. [4] Linac Coherent Light Source (LCLS), SLAC National Accelerator Laboratory, 2575 Sand Hill Road, Menlo Park, CA 94025, USA. [5] Max-Planck-Institut für Molekulare Physiologie, Otto-Hahn-Str. 11, 44227 Dortmund, Germany. [6] Université de Strasbourg-CNRS, UMR 7504, IPCMS, 23 Rue du Loess, 67034 Strasbourg, France. Correspondence and requests for materials should be addressed to I.S. (email: Ilme.Schlichting@mpimf-heidelberg.mpg.de) or to T.D. (email: Tatjana.Domratcheva@mpimf-heidelberg.mpg.de)

Rhodopsins are retinal chromophore-containing photo-receptors that form a class of membrane proteins with a wide range of functionalities, including visual signalling and ion channelling. The primary photochemical event upon light absorption is the isomerization of retinal. This double-bond isomerization is one of the most studied processes in photo-biology. The quantum yield, regio-specificity and timescale of photoisomerization differ between retinal in solution and retinal bound to microbial and animal rhodopsins, respectively. Therefore, interactions between the chromophore and the protein matrix are expected to play a central role in controlling the evolution of the excited state through steric constraints and dynamic effects[1]. In view of the extremely short timescale of the reaction, it was hypothesized that coherent vibrational dynamics play a critical role in directing isomerization[2]. Although a great deal is known about the excited-state dynamics of photoexcited rhodopsins from ultrafast spectroscopy and computation, knowledge of the ultrafast structural dynamics of the protein and their role in the isomerization remains scarce. The best characterized system is bacteriorhodopsin (bR), a light-driven proton pump[3], due to its biochemical robustness. The retinal chromophore of bR (*Halobacterium salinarum*) is linked to the side chain of Lys216 via a protonated Schiff base (PSB, Fig. 1a). Photon absorption by all-*trans* retinal triggers a functional photocycle consisting of a series of distinct spectroscopic intermediates (I→J→K→L→M→N→O) with sub-ps (I), ps (J), μs (K, L) and ms (M, N, O) lifetimes. Deprotonation of the Schiff base of photoisomerized 13-*cis* retinal and protonation of Asp85 during the L→M transition ultimately results in proton translocation to the extracellular side of the membrane[4]. The all-*trans*→13-*cis* photoisomerization reaction (Fig. 1a) has been mapped out initially by ultrafast spectroscopy[5–7] and calculations[8]. These studies indicate that the Franck–Condon (FC) region of photo-excited all-*trans* retinal depopulates by relaxation along high frequency stretching modes to form the excited ($S_1$) electronic state I intermediate with a time constant $\tau$ of ~0.2 ps. It decays with $\tau \sim 0.5$ ps[9] through a conical intersection to the vibrationally hot 13-*cis* J-intermediate, before the K intermediate forms in 3 ps. Back-reaction to a hot all-*trans* ground-state occurs on the

timescale of 1–2 ps[6]. We here present a detailed study of these events, in the limit of multiphoton excitation by time-resolved serial femtosecond crystallography (TR-SFX). The latter is complemented by ultrafast ultraviolet/visible (UV/VIS) and mid-infrared (mid-IR) spectroscopy on bR in microcrystals and solution, as a function of excitation intensity. Transient absorption (TA) spectroscopy and quantum chemical calculations were used to characterize both single and multiphoton effects, which are expected to dominate in the TR-SFX experiments. The latter show distinct phases in the evolution of the twisting C13=C14 retinal bond and long-range correlated dynamics preceding retinal isomerization. We compare our comprehensive study with a recently published related investigation[10].

## Results

**Spectroscopic characterization.** When studying reactions by spectroscopy and crystallography, one has to bear in mind that reactions in solutions and crystals can differ due to effects of crystal packing and/or differences in composition of the buffer and crystallization mother liquor. Moreover, while spectroscopic signatures are often highly specific for a certain molecular species, allowing detection of even low concentration by difference spectroscopy, this is not the case for crystallography. Therefore, one tries to maximize intermediate state occupancy by adjusting experimental parameters. In case of light-triggered reactions, one typically aims at 1 photon per chromophore. Owing to the significantly higher protein concentrations in crystals than in solutions, this translates into use of much higher pump laser intensities for time-resolved crystallographic experiments than for spectroscopic ones. In particular, the use of intense ultrafast optical pump pulses can entail a number of undesired multi-photon effects. For these reasons, we performed ultrafast TA spectroscopy[11] on both bR in purple membranes (PMs) and microcrystals (Fig. 1b, c). In both cases, we observe the sub-ps decay of the $S_1$ state (visible by the loss of stimulated emission and excited-state absorption (ESA)) that gives rise to the photo-product. Additional relaxation dynamics occur on the picosecond timescale. Global exponential fitting results in time constants of

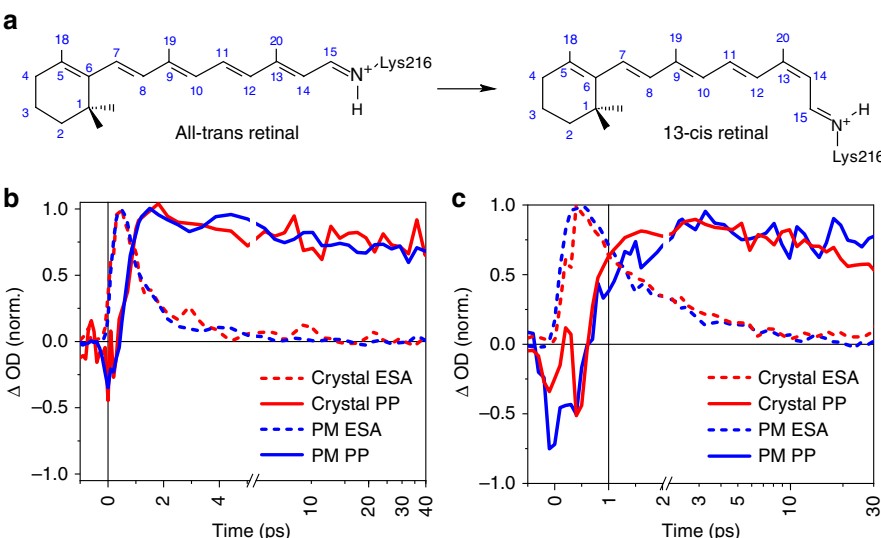

**Fig. 1** Retinal isomerization kinetics in purple membrane and bacteriorhodopsin (bR) microcrystals. **a** Schematic all-*trans* and 13-*cis* retinal covalently bound to Lys216 as a protonated Schiff base. **b**, **c** Direct comparison of kinetic traces in the ultraviolet/visible region for purple membrane (PM, blue) in $H_2O$, pH = 5.6, and the bR microcrystals in lipidic cubic phase (red) as a function of peak intensities, 35 and 88 GW cm$^{-2}$, respectively. Within the present signal-to-noise ratio, the dynamics of 13-*cis* isomer formation (PP, probe wavelength 670 ± 5 nm, solid lines) and excited-state decay (excited-state absorption (ESA), 480 ± 5 nm, dashed) are identical. See Supplementary Fig. 5 for a complete comparison as a function of excitation density. Source data are provided for Fig. 1b, c as a Source Data file

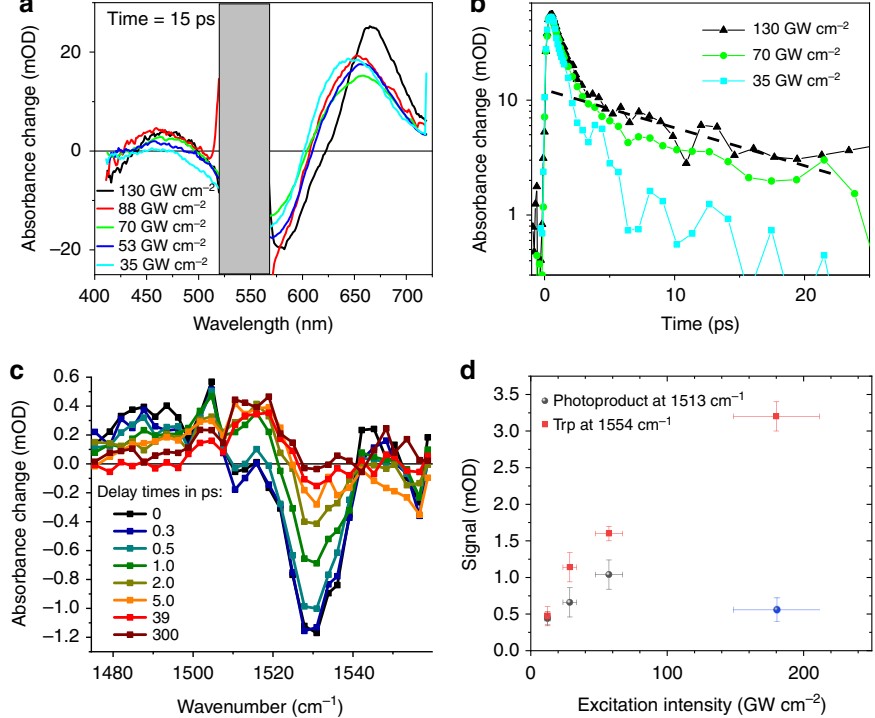

**Fig. 2** Power dependence of the photoreaction probed by transient absorption (TA) spectroscopy. **a** Ultraviolet/Visible TA spectra at 15 ps delay time in purple membranes. The 520–570 nm range (grey) is dominated by very strong pump light scatter. With increasing pump intensities, the photoproduct absorption band shifts to longer wavelengths (650–680 nm), and a long-lived weak absorption band appears peaking at ~460 nm. **b** Semi-log plot of kinetic traces at 463 nm, highlighting two decay components. For higher pump intensities, the dominant sub-picosecond excited-state absorption of the retinal protonated Schiff base increases slightly in duration, and a second smaller ~20 ps component appears (dashed line). **c** Isomerization dynamics tracked in the spectral range of the retinal C=C stretching vibration for different delay times. Excitation at 535 nm with 20 GW cm$^{-2}$ (or 0.5 μJ). The retinal bleaching signal at 1527 and the excited state around 1480 cm$^{-1}$ are formed instantaneously. The rise of the J and K product signal is reflected by the signal increase around 1505 and 1513 cm$^{-1}$, respectively. **d** Maximal signal of the isomerization product (red dots) tracked at 1513 cm$^{-1}$ as a function of excitation intensity (delay time 10–20 ps, see Supplementary Fig. 1) at 535 nm in the picosecond range; transients are shown in Supplementary Fig. 1a–e. At 180 GW cm$^{-2}$, other signals overlap with the product band at 1513 cm$^{-1}$, decreasing its intensity (blue dot), thus preventing its use to quantify photoproduct yield. The maximal negative signal of the picosecond tryptophan bleaching band (black squares) is also presented (delay time 2–5 ps, see Supplementary Fig. 1). All four measurements were performed under the same experimental conditions without changing the laser set-up, keeping the non-systematic errors small. The samples were taken from the same batch with nearly identical concentration (~1 optical density at 535 nm and 25 μm thickness). Deviation from linearity occurs at about 30 GW cm$^{-2}$. The errors of the absorption signals (y axis) were determined by the noise on the transients around the time point used to read out the signal strength. The intensity errors (x axis) are given by error propagation of the focal radius (strongest contribution), pulse length and pulse energy. Source data are provided for Fig. 2a–d as a Source Data file.

0.5–0.8 and 2.9 ps for PMs and of 0.6–0.8 and 2.8 ps for the microcrystals, respectively, with increased S$_1$ lifetimes observed for intensities >35 GW cm$^{-2}$ (Fig. 1c). We conclude that bR reacts similarly on the ultrafast timescale in PMs and in crystals.

High excitation power density is known to influence the ultrafast dynamics and spectra of bR[12–15]. At low excitation density (0.3 photon per retinal), the time courses of fluorescence emission spectra of bR exhibit a biphasic behaviour ($\tau < 0.15$ ps, $\tau \sim 0.45$ ps)[12]. At high excitation density (40 photons per retinal), the amplitude of the fast component increases strongly and the time constant of the slower component increases from ~0.45 to ~0.7 ps[12]. Moreover, a red shift of the visible product absorption spectrum of bR was identified upon increasing the pump intensity beyond the linear regime of <100 GW cm$^{-2}$[13, 15]. Therefore, we performed a power titration, varying intensities from 12 to 180 GW cm$^{-2}$ (max. available laser power) and compared bR transient dynamics in microcrystals and PMs in the VIS and in the mid-IR range (Fig. 2). Our findings match qualitatively the published non-linear spectral and temporal features (Supplementary Figs. 1–5). In the entire intensity range, we observe photoproduct formation upon retinal isomerization tracked by the product absorption around 630–680 nm (see Fig. 2a). As

expected[12], the isomerization time increases for the highest excitation intensities to $0.7 \pm 0.1$ ps, and a transient red-shift of the induced absorption band up to 670 nm is observed at 15 ps (Fig. 2a), which decays on a ≈15 ps timescale and leaves this band maximum at 650 nm (Supplementary Fig. 4d), forming the quasi-static 13-cis/all-trans difference spectrum. The spectral position of the 0.5–0.7 ps ESA in the visible range does not change, but an additional ESA band at 465 nm is observed (Fig. 2a). All these features are also observed in the time-resolved absorption spectra of microcrystals, though with significantly reduced signal-to-noise ratio (SNR) due to increased pump laser scattering (Supplementary Fig. 2). These high excitation-induced spectral features have a 15–20 ps lifetime (Fig. 2b), which is in line with the previously reported lifetime of tryptophan emission under multiphoton conditions[16]. In addition, their spectral positions, 465 and 670 nm, are consistent with ESA bands observed for tryptophan in water[17] yet red-shifted due to the different polarity of the protein environment.

The changes in vibrational absorption spectra in the linear power range are depicted in Fig. 2c. The mid-IR range shows, for $t \leq 0.5$ ps, bleaching of the retinal C=C stretching vibration at 1527 cm$^{-1}$, excited-state vibrations at 1480–1500 cm$^{-1}$ and Trp bleaching

signal at 1554 cm$^{-1}$. For $t \geq 0.5$ ps, photoproduct formation of J from 1490 to 1510 cm$^{-1}$ and K around 1515 cm$^{-1}$ induces positive signals. The photoproduct signal and Trp bleaching increase linearly with excitation intensity up to ~30–60 GW cm$^{-2}$. For higher intensities, the Trp signal shows a sub-linear dependence (Fig. 2d), while the photoproduct signal exhibits a strong decrease due to newly emerging overlapping vibrational bands (Supplementary Fig. 1f). Notably, the Trp signal decays with a time constant of ~20 ps (Supplementary Fig. 1a–e), much like the Trp-related ESA features identified in the UV/VIS range, and in agreement with the fluorescence decay of Trps in bR upon sequential two-photon excitation at 527 nm[16]. At 180 GW cm$^{-2}$, an additional sub-ps decay of the retinal C=C stretching vibration in the excited state at 1500 cm$^{-1}$ is observed, indicating the onset of additional ultrafast multiphoton effects (Supplementary Fig. 1f).

Most importantly, within the present SNR of the mid-IR data, the decay of the Trp band does not seem to increase the photoproduct signal around 1513 cm$^{-1}$ (Supplementary Fig. 1a–e), i.e. this sequential two-photon process appears to be non-productive for photo-isomerization.

The Trp dynamics is slower in microcrystals compared to PMs (Supplementary Fig. 2), but like in PMs, the mid-IR Trp bleaching signal is observed for all excitation intensities, indicating that the excited Trp and retinal are strongly coupled, as shown previously[16]. In contrast to the instantaneous retinal bleaching signal at 1527 cm$^{-1}$, a significant part of the Trp bleaching signal rises with a time constant of ~2 ps, indicating Trp excited state cooling after a sequential two-photon absorption process. The intensity range with negligible influence of the Trp pathway is <~30 GW cm$^{-2}$ corresponding to an average of less than one photon per bR trimer (Supplementary Methods). Nevertheless, even at 180 GW cm$^{-2}$, UV/VIS and mid-IR data display the characteristic sub-ps isomerization time and quasi-static K–bR difference spectra, attesting for isomerization even in the multiphoton limit. Finally, no broad spectral signatures of ionized water were observed between 1490 and 1570 cm$^{-1}$ (Supplementary Fig. 6). Nonetheless, we cannot rule out multiphoton ionization at higher pump intensities.

**Quantum chemistry analysis of retinal interactions with the protein.** To investigate the electronic coupling between retinal and protein in the excited state, we prepared a quantum mechanical (QM) model of the active site consisting of the lysine-bound retinal protonated Schiff base (RPSB), the aromatic triad (Trp86, Trp182 and Tyr185) together with other six residues and four water molecules, comprising the water cluster interacting with the PSB (Fig. 3a). The coupling of Tyr185 and Trp86 and the RPSB in bR has been characterized previously by computational studies revealing that charge transfer (CT) from these residues to the retinal significantly compensates the blue shift of the retinal absorption maximum, caused by the counterions[18, 19]. Despite this insight, the roles of Tyr185 and Trp86 in the excited-state dynamics have not yet been addressed computationally. Single-photon absorption populates the retinal $S_1$ state (computed energy 2.3 eV, Supplementary Table 1). Geometry relaxation of the $S_1$ retinal is dominated by bond-length alternation (Supplementary Fig. 7), in agreement with previous reports[20]. Moreover, a decrease of the positive charge on the PSB drives relaxation of hydrogen bonds, causing water W402 to shift. In contrast to previous studies, we included the molecular orbitals of the donor residues Tyr185 and Trp86 in the complete-active space wave function expansion (Supplementary Fig. 8). This enabled us to compute electronic states contributed by excitations from the highest occupied molecular orbital (HOMO) of each donor to the retinal lowest unoccupied molecular orbital (LUMO)—termed

here CT states CT(W86) and CT(Y185). The excited-state minima corresponding to the two CT states were found on the first excited-state potential energy surface (Supplementary Fig. 9). Transfer of a complete negative charge from Tyr185 or Trp86 to the retinal introduces a stronger CT character[20] of the CT states compared to the retinal $S_1$ state (Fig. 3b–d, Supplementary Table 1). In line with this, geometry relaxation in the CT states shows much larger changes in hydrogen bonds and the relative orientation of the donors and retinal than in the $S_1$ state (Supplementary Fig. 7). The energies of the relaxed CT states are lower than the retinal excitation energy but the energy gap to the ground state is rather large, suggesting that the CT states depopulate the FC region but do not provide de-excitation pathways competing with retinal isomerization. Importantly, the significant electronic coupling between the states (Supplementary Fig. 9c) indicates that interconversion of the $S_1$ and CT states could influence the initial ultrafast excited-state dynamics. This interconversion could correspond to the structurally observed low-frequency oscillations of the relative positions of the retinal and its neighbouring residues (i.e. the respective modes correspond to the differences in the $S_1$ and CT excited-state geometries, Supplementary Fig. 7).

These newly identified excited states corresponding to protein–retinal electron transfer can contribute to the observed multiphoton-excitation dynamics. In particular, the sequential two-photon process yielding excited Trp is a characteristic feature of the strongly coupled retinal and Trp86 and can be associated with photon absorption by either the retinal $S_1$ state or the CT(W86) radical pair (Fig. 4, Supplementary Tables 2 and 3). Indeed, the excited retinal $S_3$ state (Fig. 4b) and the excited Trp86-radical $D_2$ state of the CT(W86) manifold (Fig. 4c) are close in energy to the $S_1$ state of Trp86. The Trp86 $S_1$ state is coupled more strongly to the CT states than to the excited states of retinal (Supplementary Table 3), ruling out possible energy transfer back to retinal. In line with this, our spectroscopic data indicate that the excited-state relaxation of the Trp86 $S_1$ state occurs on the 10-ps timescale and does not increase the yield of the 13-*cis* retinal. Further work is needed to characterize possible conversion schemes after two-photon absorption.

Importantly, the identification and characterization of the CT states extends the previous experimental work describing ultrafast changes in Trp absorption by clarifying the origin of the absorption features ascribed to Trp86[7, 21]. In particular, the observed GSB in the UV[7] is consistent with the Trp86 $S_1$ energy of 4.24 eV in the ground-state model (Fig. 4a), and the positive signal[7] can be attributed to the absorption of the polarized Trp86 at 3.25 eV (Fig. 4b) or of the Trp86 radical cation at 3.75 eV (Fig. 4c). Furthermore, the computed $D_0$–$D_2$ transition of the Trp86 radical (Fig. 4c) is in line with the experimental Trp radical absorption >500 nm[22–24]. Finally, the calculated Trp86 $S_1$–$S_3$ transition may correspond to the transient two-photon induced 660–680 nm ESA band described above. Structurally, the two-photon Trp pathway is related to the same degrees of freedom as the relaxation pathways induced by single-photon excitation.

**Ultrafast serial femtosecond crystallography.** Dark-adapted bR exists as a ~50:50 mixture of all-*trans* and 13-*cis* retinal. Since only all-*trans* retinal enters a functional photocycle related to proton pumping, it is important to maximize its fraction. Light-adapted bR contains up to 100% all-*trans* retinal. This enrichment can be achieved by continuous illumination for several seconds, followed by a brief incubation in darkness. We decided against performing the light adaptation of bR microcrystals off-line, i.e. prior to injector loading as described previously[25], since this approach is relatively ineffective as detailed in the "Methods"

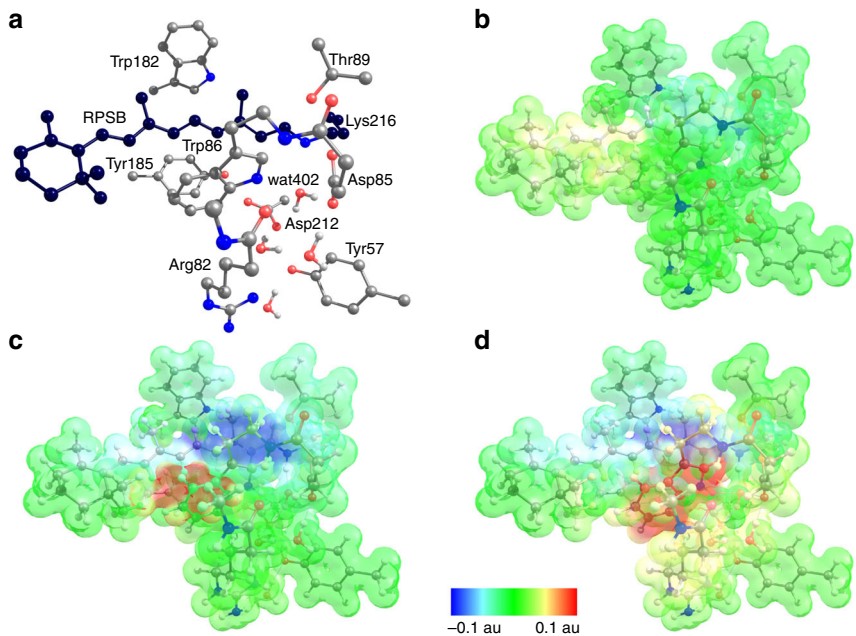

**Fig. 3** Electrostatic potential changes illustrate the charge-transfer (CT) character. The transitions $S_0$–$S_1$, $S_0$–CT(Y185) and $S_0$–CT(W86) are characterized. **a** Bacteriorhodopsin active-site cluster model (consisting of the 208 atoms in total). The carbon atoms of the retinal protonated Schiff base (RPSB)-Lys216 fragment are shown in black. Hydrogen atoms, except for the water molecules, are not shown. **b** The $S_0$–$S_1$ transition corresponds to the excited retinal; **c** The $S_0$–CT(Y185) and $S_0$–CT(W86) (**d**) transitions correspond to CT involving Tyr185 and Trp86 as electron donors to RPSB. In the $S_0$ state, the protonated Schiff base is positively charged. All electronic transitions decrease this positive charge as apparent from the light blue patch of the electrostatic potential surface in **b** and dark blue patches in **c**, **d**. The positive charge moves to the ionone ring of retinal as shown by the yellow patch in **b** or to the electron donor (Trp86, Tyr185) as shown by a red patch in **c**, **d**. The charge separation in the active site is much larger for the $S_0$–CT transitions than for the $S_0$–$S_1$ transition

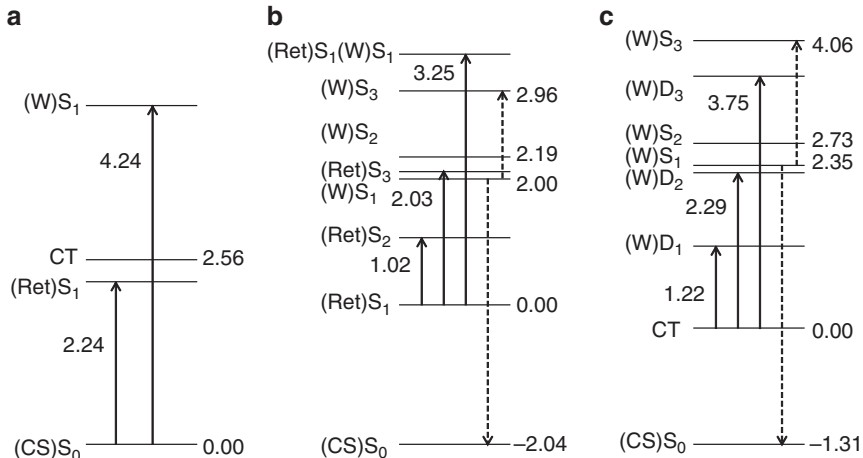

**Fig. 4** Computed energies characterizing electronic coupling between retinal and Trp86. Solid arrows indicate retinal and Trp86 absorption; dashed arrows indicate absorption and emission of excited Trp86 populated by a sequential two-photon process. All computed energies and transition dipole moments (tdm) are presented in Supplementary Tables 2 and 3, respectively. **a** Transitions of the retinal and Trp86 corresponding to the light-adapted state of bacteriorhodopsin. **b** Transitions from the relaxed retinal $S_1$ state. Absorption of a second photon may populate the retinal $S_3$ state at 2.03 eV (tdm = 5.37 au) coupled to the excited Trp86 (W)$S_1$ state at 2.0 eV (tdm = 0.15 au). **c** Transitions from the relaxed CT state. Photon absorption populates the excited Trp86 radical state (W)$D_2$ at 2.29 eV (tdm = 1.00 au) coupled to the excited Trp86 (W)$S_1$ state at 2.35 eV (tdm = 0.45 au)

section. Instead, we developed a protocol for on-line pre-illumination of the microcrystalline bR stream as it flowed through our high viscosity extrusion (HVE) injector[26]. A fully defined "dark zone" near the HVE exit allowed the crystals to return to the bR ground state prior to being photoexcited with the optical pump laser pulse. Pre-illumination increased the all-*trans* retinal content considerably, with compositions ranging from 65% to 80%

all-*trans* and from 35% to 20% 13-*cis* retinal, respectively, compared to 45% all-*trans* and 55% 13-*cis* retinal in the dark-adapted state.

TR-SFX data were collected at the coherent X-ray imaging (CXI) instrument[27] at the Linac Coherent Light Source (LCLS). bR microcrystals embedded in a high viscosity stream were injected in random orientations into the FEL beam. We employed

a pump–probe SFX data collection scheme. The stream velocity was adjusted (and controlled regularly) to ensure clearing of any illuminated material out of the FEL interaction zone before the next pump–probe cycle started. To help with this, the optical laser beam (99 μm $1/e^2$) was moved $25 \pm 5$ μm downstream of the X-ray focus. Photoexcitation was performed in a multiphoton regime. The time delays between the optical laser flash (145 fs, 532 nm, 5.9 μJ, ~500 GW cm$^{-2}$ in the interaction region, see "Methods" section) and the probing XFEL pulse were nominally 0.5, 1, 3 and 10 ps and 33 ms. Data from the nominally 0.5-ps delay were binned into 12 groups according to their actual time delay covering 0.24–0.74 ps (Supplementary Methods). For comparison, we also collected data of the dark state. The 33-ms time-delay data set, corresponding to an M-like intermediate, was collected to check whether the short-lived intermediates can advance along the bR photocycle.

**Data quality and analysis**. Diffraction and optical experiments have opposing demands on sample thickness. While large bR crystals diffract to significantly higher resolution than smaller ones, they cannot be photoexcited efficiently (1/e penetration depth at 532 nm is ~3.5 μm). As a compromise, we used bR microcrystals of $20$–$30 \times 20$–$30 \times 2$–$3$ μm$^3$. This corresponds to an average path length of ~5.9 μm through the crystals (Supplementary Methods). The crystals yielded 1.8 Å resolution SFX data (Supplementary Table 4). Analysis of electron density maps calculated from extrapolated structure factors with varying occupancy of the pumped state yielded an occupancy of ~15% for the sub-ps and 1 ps data (10% for 3, 10 ps) (Supplementary Methods). For the sub-ps and 1 ps data sets, we observe differences with respect to the dark-state structure after refinement against extrapolated structure factors. For the 3- and 10-ps data sets, changes of the torsion of the C13=C14 bond in the chromophore are apparent in both structure factor amplitude difference Fourier maps and extrapolated maps (Fig. 5a–f). Crystallographic statistics are shown in Supplementary Table 4. To estimate the uncertainties in the torsion angles and atom–atom distances, we used a jackknife-type method as described in Supplementary Methods. For the uncertainty in the time delays, we used the standard deviation of the time delays present in each data set. The temporal binning of the diffraction images into a series of data sets with distinct average time delays results in electron densities that correspond to the weighted average of the individual conformations present at the actual time delays being binned. Thus the amplitudes of the structural displacements are weighted with a binning function. In case of a known temporal dependence of the structural changes (oscillatory), the effect of binning can be addressed computationally (see Supplementary Methods). This results in changes of the magnitude of the measured displacements in particular of the early and late sub-ps time points.

To assess the information content of the extrapolated electron density maps and to obtain information on the time course of events, we performed singular value decomposition[28, 29] on the extrapolated maps for the 12 sub-ps time points as described in detail in Supplementary Methods. The first and strongest contribution corresponds to the average of the 12 maps. The other contributions, 2–12, are far smaller, with contributions 2–5 probably containing signal and number 6–12 probably mainly containing noise (Supplementary Fig. 10a). As expected, the time evolution of the first contribution, the average, is virtually constant with time. The time course of the second contribution is reminiscent of a step function, centred at ~0.5 ps, and that of the third is Gaussian-like, centred also at ~0.5 ps (Supplementary Fig. 10b). The second contribution shows the distortion at the C13-C14 bond described below, and the third displays "bending"

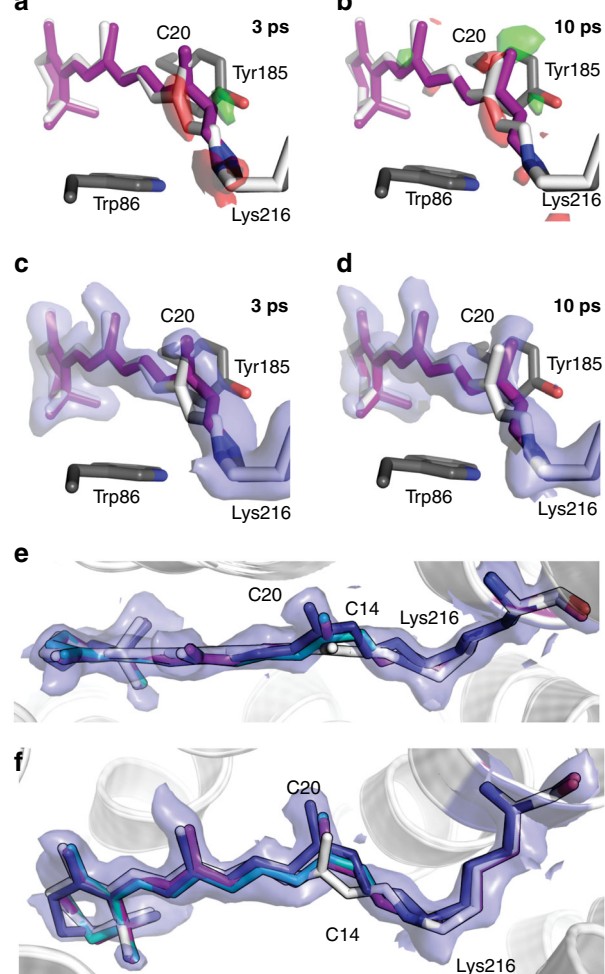

**Fig. 5** Structure of the retinal in bacteriorhodopsin on the ps timescale. **a** View along retinal at time delays of 3 ps (**a**, **c**) and 10 ps (**b**, **d**); the retinal is shown in purple and the dark-state structure as white sticks. Extrapolated electron difference maps (green, +3σ and red, −3σ) show clear deviations from the dark-state structure at the C20 methyl group as well as around the C13=C14 bond. Both the C20 methyl group and the C14 atom shift towards residue Tyr185 (shown as grey sticks). The 10-ps structure was refined using geometric parameters from a quantum mechanical (QM)-optimized model resulting in a −68° torsion angle, corresponding to a twisted 13-*cis* conformation of retinal. **e**, **f** Overlay of the unrefined QM model of the 10-ps time point with the electron density, the top view (**e**) and side view (**f**) of the retinal binding pocket are shown. Retinal dark state, 10 ps structure refined with the QM model, unrefined QM model and 10 ps structure from Nogly et al.[10] are shown as white, purple, teal and deep blue sticks, respectively. Extrapolated electron density of the 10-ps time point is contoured at 1σ

of the retinal. While these contributions by themselves should on no account be interpreted as separate species, modes or events, these results do establish that a major change occurs in the retinal structure at around 0.5 ps.

**Changes in the retinal**. In the dark state (Figs. 1a and 6a), the polyene chain of all-*trans* retinal is largely planar (C6-C13) with a slightly pre-twisted C13=C14 double bond when no planarity for the atoms around this bond is enforced during refinement (torsion angle −150°, "dark-unrestrained" structure), in line with computational work (−160°)[20]. Within the first time point of

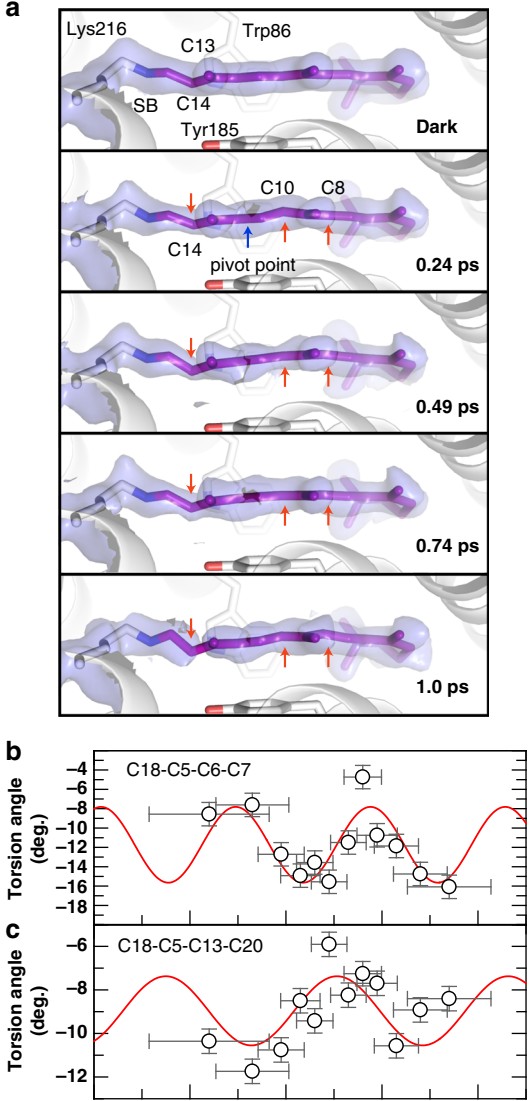

**Fig. 6** Sub-ps effects of photon absorption on retinal. **a** Close-up of the retinal (purple sticks) in the dark-unrestrained state, as well as at 0.24, 0.49, 0.74 and 1 ps. Distortions from the initial planarity of the retinal occur within 0.24 ps and evolve further in time, especially at the Schiff base bond, at the C13=C14 bond and at the C10-C11 bond, while the C10 and C14 atoms move in opposite directions (red arrows, a blue arrow marks the pivot point) away and towards Tyr185 (shown as sticks), respectively. Minor torsion occurs also at C8. The electron density maps around the retinal (refined for the dark state, 15% extrapolated for the other panels) are contoured at 1.0σ. At 1 ps, the density for the C13=C14 bond is only visible at lower contour levels. **b**, **c**, The torsion angles between the β-ionone ring and the beginning (C18-C5=C6-C7 torsion angle, **b**) and end (C18-C5-C13-C20 torsion angle, **c**) of the polyene chain are plotted in time. The oscillatory modulation of the angles with frequencies of 119 ± -9 cm$^{-1}$ (**b**) and 93 ± 14 cm$^{-1}$ (**c**) further visualize the twisting of the entire retinal. The error bars were derived as described in the Supplementary Methods. Source data are provided for Fig. 6b, c as a Source Data file

photoexcited retinal, the torsion angles are apparent values that describe the average conformation of any process contributing to retinal isomerization. The temporal evolution of the apparent C13=C14 dihedral angle can be separated in three distinct phases: 0–~500 fs (I), ~500–750 fs (II), 1–10 ps (III) (Fig. 7). In the single-photon regime, the excited state relaxes on the potential energy surface (the I state) followed by appearance of the J state with a vibrationally excited 13-*cis* conformation delayed by ~150 fs relative to time zero and a subsequent rise time of 450 fs[6, 30]. In parallel with the ps rise of K state, the all-*trans* ground state is repopulated (see Fig. 2c and Supplementary Fig. 1b). The J to K transition induces geometric changes, such as torsional relaxation, reduction of conformational heterogeneity and vibrational cooling[30].

In the present multiphoton regime, i.e. beyond the two-photon regime characterized by the spectroscopic data (≤180 GW cm$^{-2}$), the situation most likely involves population of several all-*trans* excited states S$_n$ followed by ultrafast multi-path electronic relaxation (≤0.5 ps, phase I, Fig. 7a, b) leading to hot populations in S$_0$ and S$_1$, with distorted retinal conformations and longer vibrational relaxation times on the electronic excited potential energy surface, compared to the single-photon regime. In phase II (Fig. 7a, c), the excited-state decays and transitions to the 13-*cis* J state and hot all-*trans* ground state take place. However, in this multiphoton regime, the hot all-*trans* ground state may be accessed through additional internal conversion pathways from the hot excited state or hot J state. Indeed, such additional non-reactive channels would be consistent with the low occupancy of isomerized retinal and the reported reduction of the isomerization quantum yield in the high excitation regime[14]. The existence of structurally non-characterized multiphoton ionized species cannot be excluded and would equally contribute to a reduced quantum yield. Nevertheless, the majority population relaxes into the all-*trans* ground state with large vibrational excess energy, with an average dihedral angle close to the geometry of unpumped all-*trans* retinal. Hence, in phase II (0.5–0.75 ps, Fig. 7a, c), the small reduction of the dihedral angle is interpreted as a combination of hot 13-*cis* J state formation and effective back-reaction to the hot all-*trans* ground state. This is followed by a slower phase (ps, III, exponential, Fig. 7a, d) reflecting J to K transition and K state relaxation, accompanied by reduction of conformational heterogeneity and cooling of the 13-*cis* product. Vibrational relaxation is significantly longer for multiphoton excitation due to increased excess energy.

Local distortion between the β-ionone ring and the polyene chain is manifested by the oscillation of the torsion angle C18-C5=C6-C7 at a frequency of ~120 cm$^{-1}$ (Fig. 6b). Globally, skeletal torsion is apparent from e.g. oscillations of the torsion angle between terminal methyl groups (C18 and C20; angle C18-C5-C13-C20) of ~90 cm$^{-1}$ (Fig. 6c). The C12-C13=C14-C15 torsion angle changes from −150° to −135° during 0.24–0.74 ps and refined to −112° at 1 ps and −100° at 3 ps (Supplementary Table 5). The 10 ps data are too weak to be refined based solely on the electron density. Therefore, we started the refinement using geometric parameters provided by our QM-optimized model (10 ps, torsion angle −41°) which resulted in an angle of −68°. However, the density can also very well accommodate the unrefined QM model (Fig. 5e, f).

Calculations have revealed that the isomerization mechanism in bR comprises an aborted double bicycle–pedal motion, involving a concurrent twisting conformation of three adjacent double bonds C11=C12, C13=C14 and C15=N in alternate directions (counter clockwise, clockwise, and counter clockwise, respectively)[20]. This proposal agrees qualitatively with our observations in the multiphoton regime; however, we also note differences. In our structures, the C11=C12 bond undergoes

0.24 ps, the C9=C10 bond also distorts and undulates on the sub-ps timescale, while the C13=C14 bond torsion mostly progresses. The largest displacements, in opposite directions, are at atoms C14 and C10, while C8 undergoes minor distortion in the same direction as C10. Like all other features describing the

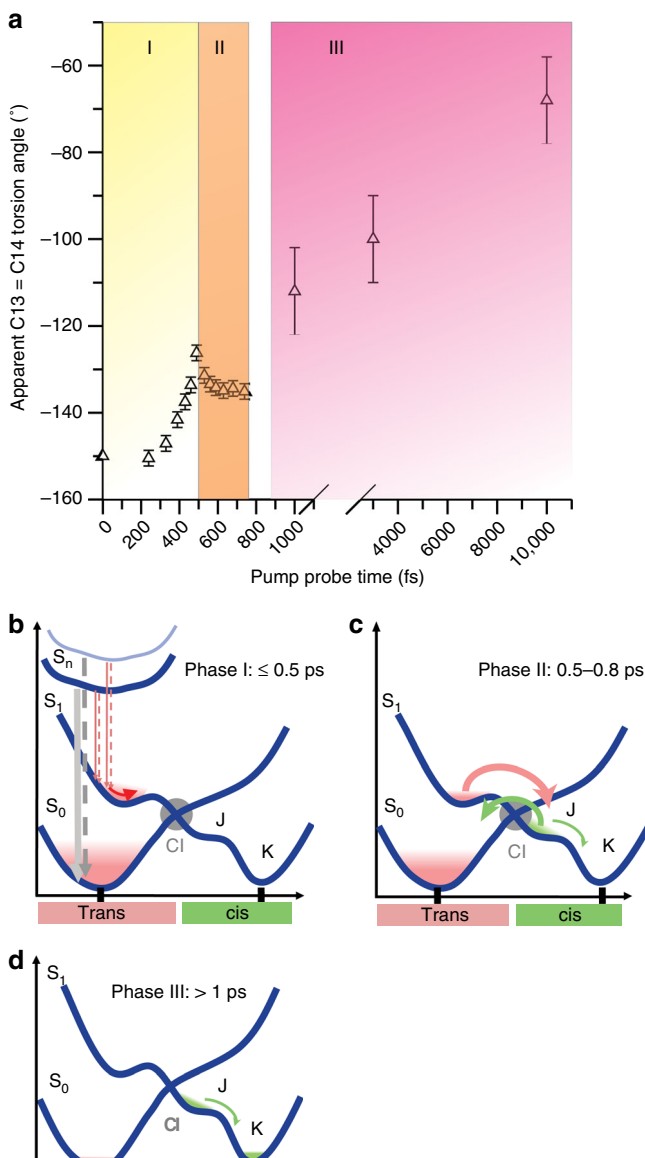

**Fig. 7** Temporal evolution of the apparent C13=C14 retinal torsion angle.
The structural dynamics of the twisting bond can be divided into three distinct phases (**a**). **b** The first (I, 0–~500 fs, yellow) reflects the initial torsional motion in $S_1$ after relaxation from multiphoton excited high energy states $S_n$ and internal conversion into $S_0$. **c** During the second phase (II, ~500–750 fs, beige), the torsion angle decreases, indicating transition not only to the hot J state but also to a hot all-*trans* ground state. The small reduction of the dihedral angle is due to an average over the populated vibrationally hot distorted $S_1$, the 13-*cis* J state and the all-*trans* ground state. The 0.5–0.8 ps timescale is consistent with the decay of the excited state (cf. "spectroscopic characterization"). This is followed by phase III (1–10 ps, pink, **d**) displaying the increase of the torsion angle, as expected for J and K state relaxation, ultimately yielding the relaxed 13-*cis* isomer of retinal. The error bars were derived as described in Supplementary Methods. Source data are provided for Fig. 7a as a Source Data file

hardly any twisting, whereas the twisting of the C15=N double bond is much more pronounced compared to the calculated structures[20]. This may be explained by the limited model used in the QM calculations[20], which included—in addition to the retinal PSB—only the terminal $\varepsilon$-carbon atom of Lys216. Therefore, all

torsional changes required for the isomerization can occur only in this fragment. However, in our complete system, we see substantial distortions in the lysine side chain, which has more degrees of freedom to twist around its single bonds, accommodating the twisted retinal much more easily. We also see tilting of the methyl group C20 towards Tyr185 during the isomerization reaction, with the distance between C20 and the hydroxyl group of Tyr185 decreasing by ~0.5 Å at 10 ps (Fig. 5c, d). Further motion is likely limited sterically. Restricting the C20 motion could be a means of the protein to stabilize the all-*trans* configuration and restrain isomerization to only the C13=C14 double bond. Indeed, experiments on bR reconstituted with 13-demethylretinal showed that the protein contains predominantly the 13-*cis* isomer and undergoes neither light–dark adaptation nor a functional photocycle[31].

**Ultrafast changes in the protein**. The data also allow us to study the sub-ps temporal evolution of the conformation of residues surrounding the retinal (see Fig. 8a). In the dark state, PSB forms a hydrogen bond with Wat402 (2.8 Å). Upon photoexcitation of the retinal, the PSB–Wat402 distance increases within 0.4 ps to 3.2 Å and oscillates in time with a frequency of ~80 cm$^{-1}$ (Fig. 8b), similar to the oscillation of the retinal skeletal torsions and the Lys216 $\chi_4$ torsion angle (both ~90 cm$^{-1}$, Figs. 6c and 8c). To accommodate the shifted Wat402, the carboxylate group of Asp85 reorients within 0.24 ps; the Asp212 $\chi_2$ torsion angle as well as distances between the carboxylate oxygen atoms of Asp212 and Wat402 show an oscillatory behaviour (~80 cm$^{-1}$, Fig. 8d, e, Supplementary Fig. 11a). These displacements likely cause the rearrangements of Wat401 and Wat406, propagating the motion to Arg82 (Supplementary Fig. 11b) and to Glu194 and Glu204: the distances between their carboxylate groups and Wat403/Wat404, which are part of the proton release group on the extracellular side[4], are modulated with frequencies of ~80–100 cm$^{-1}$ (Fig. 8f, Supplementary Fig. 11c–g).

In addition to the above oscillations of residues and waters directly involved in the later proton translocation, we clearly observe oscillations in the positions of residues lining the retinal (Fig. 8g–i and Supplementary Fig. 11h–j). These include Trp86 ($\chi_1$ and $\chi_2$ oscillation period ~150 and ~125 cm$^{-1}$, respectively) and Tyr185 ($\chi_1$ oscillation period about 80 cm$^{-1}$) as well as Met118 and Met145 lying within 4 Å of the retinal. Met118 $\chi_2$ and $\chi_3$ and Met145 $\chi_2$ oscillate with ~100, ~100 and ~90 cm$^{-1}$, respectively. We also performed singular value decomposition (SVD) on the maps using only a region within 5 Å around the retinal. Here, too, the SVD analysis revealed oscillations with frequencies of about 83 and 113 cm$^{-1}$, further supporting the observed oscillations in the coordinates (Supplementary Table 6).

The magnitude of the damping time constants of the observed oscillations can be estimated to several 100 fs by correcting the amplitudes of the oscillations for the effects of binning as described in "Methods". We then fitted an exponentially decaying harmonic oscillation corrected for binning to the data. The accuracy of this approach is limited by: (i) correcting the refined amplitudes by effect of temporal binning will result in the lower bound of the real amplitude (see "Methods"), which affects the initial estimate of the decay constant, and (ii) owing to limited beamtime, we could only collect data for binned time points 0.24 ps $\leq \Delta t \geq 0.74$ ps covering not more than 1.5 oscillation cycles. Nevertheless, fitting a binning-corrected exponential decay to the observed amplitudes, there is an indication that in most cases the resulting damping constant is on the order of 0.2–0.3 ps (Supplementary Fig. 12). However, owing to the limited data, this value has to be considered with caution (see "Methods"). Whether or not these changes also occur in the single-photon

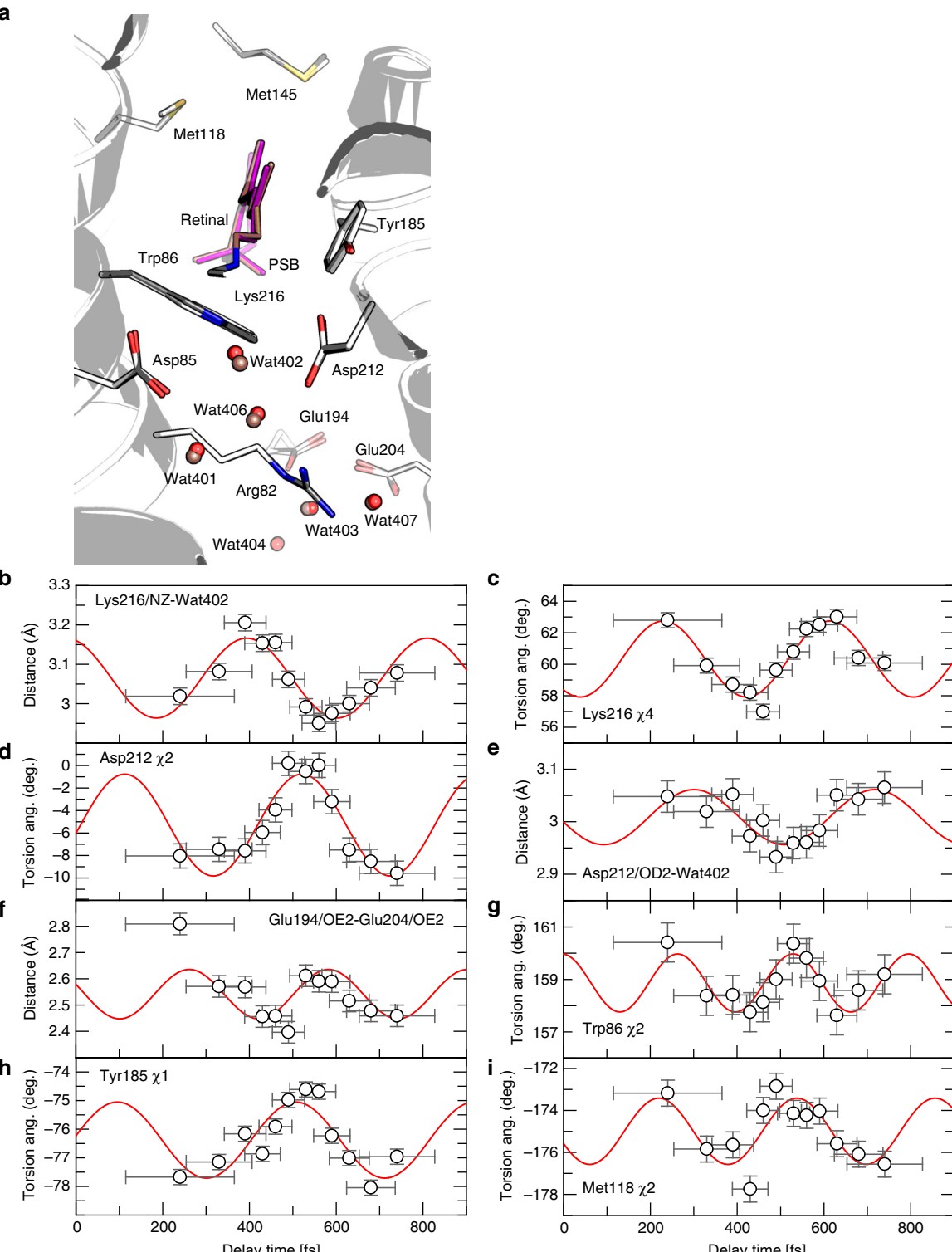

excitation regime and are thus of functional relevance needs to be tested by future experiments.

In any case, our data show torsional oscillations not only in the retinal PSB (Fig. 6) but also in the surrounding residues and in their relative distances (Fig. 8, Supplementary Fig. 11). This suggests that the photoexcited retinal and its neighbouring residues form a vibrationally coupled network. The extent and direction of the conformational changes occurring in the retinal and its surroundings are visualized and summarized in Fig. 9.

**Comparison with published spectroscopic data**. Retinal isomerization in bR in the high-intensity regime was studied by Florean et al.[13] who reported that photoproduct yield increases approximately linearly well beyond the saturation of the initial one-photon transition. This was disputed by Prokhorenko et al.[14] who showed a monotonic decrease of the isomerization yield >80 GW cm$^{-2}$, which was attributed to nonradiative loss of the excitation through re-excitation of the excited-state $S_1$ to higher excited states in the multiphoton regime. Such a re-excitation is identified spectroscopically here in its simplest form in the

**Fig. 8** Motions of the water network and surrounding residues on the sub-ps timescale. **a** View along the retinal chain towards the extracellular site. The dark-unrestrained state and 0.46 ps structures of retinal (purple and dark-salmon sticks, respectively), of selected side chains (white and dark grey sticks, resp.) and of water molecules (red and dark-salmon spheres, respectively) are superimposed. In the all-*trans* dark state, the protonated Schiff base (PSB) forms a hydrogen bond (HB) with Wat402 (2.8 Å), which is wedged between counter ions Asp85 (2.5 Å) and Asp212 (3.0 Å). These are part of an extensive network of HB extending to the extracellular side (EC) of the membrane. Upon photoexcitation, displacements of waters (Wat402, Wat401, Wat406, Wat403, Wat407 and Wat404) and HB-connected residues (Asp85, Asp212, Arg82, Glu194, Glu204) and nearby residues (Trp86, Tyr185, Met118) are observed. **b–f** Within the HB network, these displacements likely propagate from the PSB region towards the EC as coupled motions, as observed in the oscillatory modulations of torsion angles and distances within the network. **b** Upon photoexcitation of the retinal, the PSB–Wat402 distance initially increases within the first 0.4 ps to 3.2 Å and then seems to oscillate with time with a frequency of $80 \pm 5$ cm$^{-1}$. **c** Oscillation of the Lys216 $\chi_4$ torsion angle with a frequency ($87 \pm 6$ cm$^{-1}$) similar to the retinal skeletal oscillations (Fig. 6c). **d**, **e** Asp212 $\chi_2$ angle and Asp212/OD2–Wat402 distance oscillation of $82 \pm 6$ and $80 \pm 9$ cm$^{-1}$, respectively. **f** The distance of Glu194/OE2 and Glu204/OE2 undulates with a frequency of $104 \pm 17$ cm$^{-1}$. **g–i** Residues lining the retinal cavity also show oscillatory modulation of their torsion angles with the following frequencies: Trp86 $\chi_2$ ($125 \pm 9$ cm$^{-1}$, **g**), Tyr185 $\chi_1$ ($81 \pm 9$ cm$^{-1}$, **h**), Met118 $\chi_2$ ($105 \pm 12$ cm$^{-1}$, **i**). The error bars were derived as described in Supplementary Methods. Source data are provided for Fig. 8b–i as a Source Data file

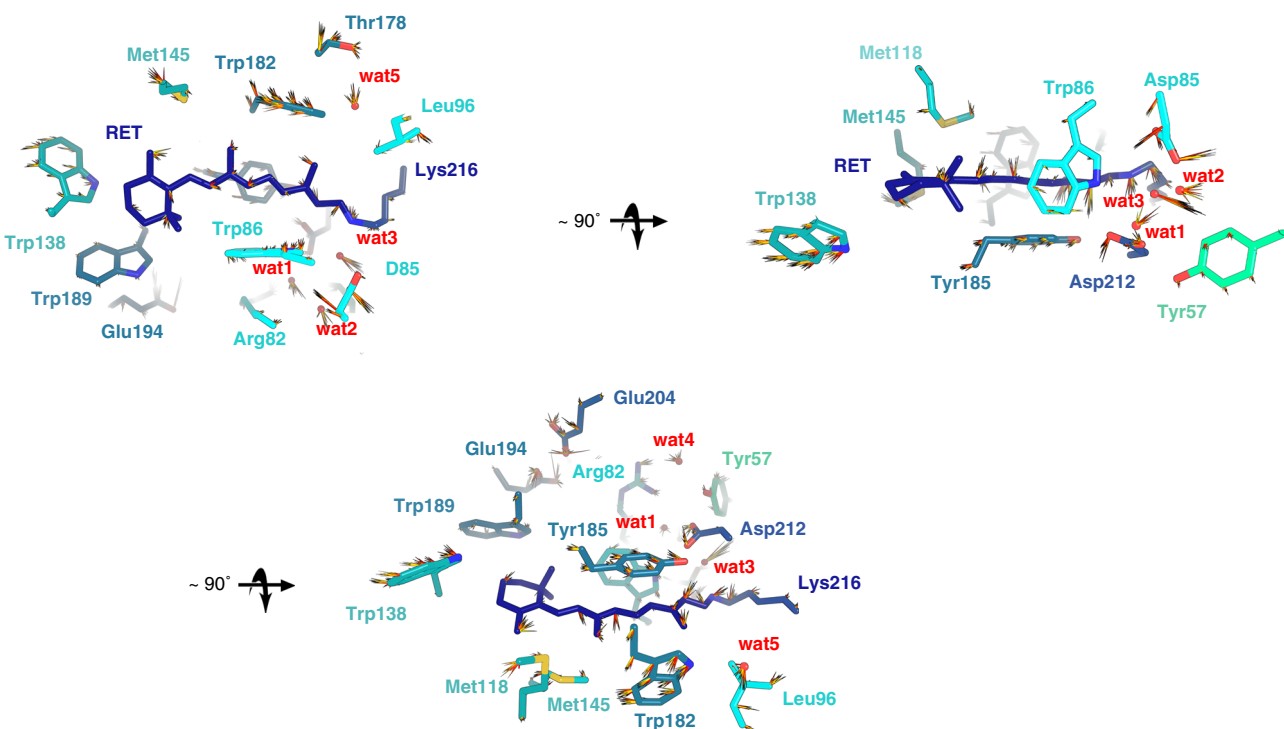

**Fig. 9** Ultrafast changes of residues and waters in the immediate vicinity of retinal. Selected residues from the dark "unpumped" structure are shown as sticks, with the retinal pigment coloured in dark blue; carbon atoms of surrounding residues coloured in shades of cyan and blue (N-terminal to C-terminal); and nitrogen, oxygen and sulfur atoms coloured in marine blue, red and yellow, respectively. Arrows indicate the movement of atoms on the fs timescale, with the magnitude of motions illustrated by the length of arrows (multiplied five times). Arrows of different colours correspond to different pump–probe delays, with increasingly red-shaded arrows corresponding to longer pump–probe delays, in the 240–740-fs range

regime of sequential two-photon excitation and by the onset of sub-ps excited-state reduction for the highest pump intensities (180 GW cm$^{-2}$).

Although our SFX data were obtained in the multiphoton excitation regime, it is interesting to compare them to spectroscopic data obtained in the single-photon regime. As inferred from vibrational spectroscopy[5, 6, 30, 32, 33] and computations[8], the photoinduced structural changes of retinal in bR involve ultrafast intramolecular bond length alterations, launching C=C and C-C stretching modes and out-of-plane and torsional degrees of freedom. In line with these findings, we experimentally observe and computationally predict immediate changes along the entire retinal PSB. The strongly bent retinal conformation observed on the sub-ps timescale reflects torsional relaxation, expected in the single-photon regime during the transition from the I to J intermediate. It has been predicted that collective torsional

motions drive the population on the S$_1$ potential energy surface towards the conical intersection and thus induce isomerization[20]. Excited-state processes are accompanied by low-frequency vibrational coherences (80–177 cm$^{-1}$) detected in both optical[34, 35] and impulsive vibrational spectroscopy[33, 36, 37]. Most importantly, and highlighting the role of the protein, they differ between free and protein-bound retinal[35, 37].

## Discussion

Ultrafast time-resolved spectroscopic experiments have provided unprecedented insight into the early events following photon absorption of light-sensitive systems. Analogous SFX experiments are only emerging, and while conceptually similar, there are crucial differences between spectroscopic experiments on protein solutions and SFX experiments on microcrystals. Typically, the

latter aim to maximize the occupancy of intermediate states. To this end, a power titration should be performed to establish the linear range of pump intensity increase on intermediate occupancy. This, however, has not been done for the published experiments due to paucity of XFEL beamtime. To date, all sub-ps time-resolved optical pump X-ray probe SFX experiments have used laser power densities of 360–500 GW cm$^{-2}$[10, 29, 38, 39], which results in far >1 photon per chromophore even if a safety margin for potential pump laser drifts or scattering effects is considered.

During the review of our manuscript, another study describing a time-resolved SFX experiment on the ultrafast steps in retinal isomerization in bR was published by Nogly et al.[10]. In contrast to our experiment, Nogly et al.[10] performed light adaption offline, photoexcited the sample at 30 Hz and collected the diffraction data at 120 Hz (i.e. four X-ray pulses after each pump laser pulse, see Supplementary Fig. 1A in Nogly et al., Supplementary Table 7). Importantly, the same pump laser flash was used for two data sets: crystals illuminated at position $x_0$ were probed by the first XFEL pulse after (sub)-ps time delays; crystals illuminated at position $x_1$ further upstream in the lipidic cubic phase (LCP) stream and reaching the interaction zone 8.3 ms after the first X-ray pulse were probed by the second XFEL pulse (Supplementary Fig. 13). The second data set corresponds to an M-like intermediate with 10% occupancy, compared to 16% for the 10 ps time-delay intermediate[10]. Assuming single-photon excitation, as stated by the authors, this implies that the photon intensity at position $x_1$ is ~60% of that at position $x_0$. However, this high intensity is very difficult to reconcile with the reported illumination geometry (see Supplementary Fig. 13, Supplementary Table 7). The power of the pump laser is 11% of the peak power at $x_0$ but negligible at $x_1$. Since this is incompatible with any photoexcitation at $x_1$, it is very difficult to assess the experimental description including the pump laser power density used in the Nogly et al. experiment[10]. Similarly, a reduction of the incident laser light by 80% due to scattering at the LCP jet was stated, but no experimental details or data were provided[10]. This value does not agree with our findings (20% of light is scattered by the LCP jet itself and the embedded crystals, Supplementary Note 1) and is most likely a lensing effect due to the cylindric jet. Despite these many experimental inconsistencies, if one assumes a 50-μm displacement of the pump laser[10], the two studies used a comparable pump power density of ~500–600 GW cm$^{-2}$, resulting in multiphoton excitation (Supplementary Table 7). This contradicts the assessment of Nogly et al.[10] of being in the single-photon excitation regime.

Nogly et al.[10] collected significantly more data at higher spatial resolution (1.5–2.0 Å) that allowed binning with much shorter temporal (~120 fs) spacing. Interestingly, as in our case, negative difference densities are much stronger than the corresponding positive densities, indicating significant disorder of the developing states compared to the initial state. This effect is expected in multiphoton processes in which there will be a distribution of potential energy surfaces explored in the relaxation process. Moreover, multiphoton ionization may result in electrons being trapped at polar sites, resulting in further disordering. Peak heights of 18 difference electron densities were listed as a function of time, however, without describing and depositing the corresponding structures and data, respectively. While difficult to compare, some of temporal changes in peak height are reminiscent of the structural oscillations we describe here, with periods of 300–400 fs (e.g. torsion angle C20-C13-C5-C18 Fig. 6c versus Supplementary Fig. 20[10]). Qualitatively, the structural changes of retinal and the surrounding water network seem to be similar to what we observe. In general, the changes reported by Nogly et al.[10] are larger than the ones that we observe (see Fig. 10 for the

counter-ion H-bonding network). In particular, we do not observe a protein quake[10]. It is unclear whether this is due to the lower resolution and SNR of our data, differences in refinement protocols or photoexcitation or a combination thereof. In this context, it is interesting to note that increasing the pump laser power above a value where complete photolysis was attained resulted in larger displacements in photolysed carbonmonoxy myoglobin[38]. A quake-like motion of the protein occurring within ps after multiphoton excitation (800 photons per chromophore) was described for the photosynthetic reaction centre[40]. Protein quakes have been proposed as a means to quickly dissipate energy. However, other relaxations channels exist, such as intramolecular vibrational energy redistribution and vibrational cooling.

Since it is known that high photoexcitation intensity influences the excited-state spectra and dynamics of bR[12–15], we investigated the impact of high photoexcitation power on the bR photocycle and dynamics in PMs and bR microcrystals by a spectroscopic power titration. Ultrafast TA spectroscopy in the UV/VIS and mid-IR reveals red-shifted absorption signals due to transitions to higher excited states above ~30 GW cm$^{-2}$ or above one photon per bR trimer. Our quantum chemical calculations indicate that a sequential two-photon absorption process populating the Trp excited-state manifold (Fig. 4) could be the cause for the observed and previously reported[12–14] red-shifts in the visible spectral range and the bleaching of Trp vibrational bands (Fig. 2). On this basis, we ascribe the intensity-dependent changes of the photoproduct region around 630–650 nm and the long-lived 460 nm ESA at least partially to the sequential two-photon absorption process populating the excited state most probably of Trp86. The relaxation dynamics in the Trp excited-state manifold takes place on a timescale of tens of ps, without producing additional 13-*cis* photoproduct (Supplementary Fig. 1a–e). Other multiphoton processes on the sub-ps timescale alter the retinal excited-state dynamics (Supplementary Fig. 1f) without significant influence on the 13-*cis* isomer formation time.

The insight gained from the spectroscopic experiments cannot be applied directly to the SFX experiments, since the latter were performed at much higher pump laser intensity. There we observe three distinct phases in the evolution of the apparent C13=C14 torsion angle following multiphoton excitation of retinal, likely reflecting the fast decay of different highly excited species with a distribution of rate constants to very hot $S_1$ and $S_0$ states (I and II, respectively, in Fig. 7), with only the former contributing to formation of the 13-*cis* isomer (III, Fig. 7). Again, multiphoton-induced ionization cannot be excluded. Under these conditions, we not only observe sub-ps and ps structural changes in the retinal but also in surrounding protein residues and water molecules. These include damped oscillations of torsional angles in retinal and surrounding amino acids as well as oscillations in interaction distances. While we can only give a tentative first interpretation of the effects of multiphoton excitation on retinal isomerization and thus on the oscillations, their presence and similarity with those observed for retinal in one-photon excited transient spectroscopy is remarkable. Most important is the previously undescribed observation of oscillations of nearby amino acids under multiphoton excitation, which require an impulsive triggering most likely due to the sub-50 fs dipole polarization of excited-state retinal[7]. These rapidly damped but coherent oscillations at these low-energy eigenfrequencies are a manifestation of a robust and strong vibrational coupling of retinal and its environment. It is conceivable that these eigenfrequencies are also activated in the single-photon regime. Finally, the similarity of the structures of the M-like intermediate determined from data collected at 33 ms and the M-like intermediate structure collected at 1.7 ms[25] (Supplementary Fig. 14) shows that

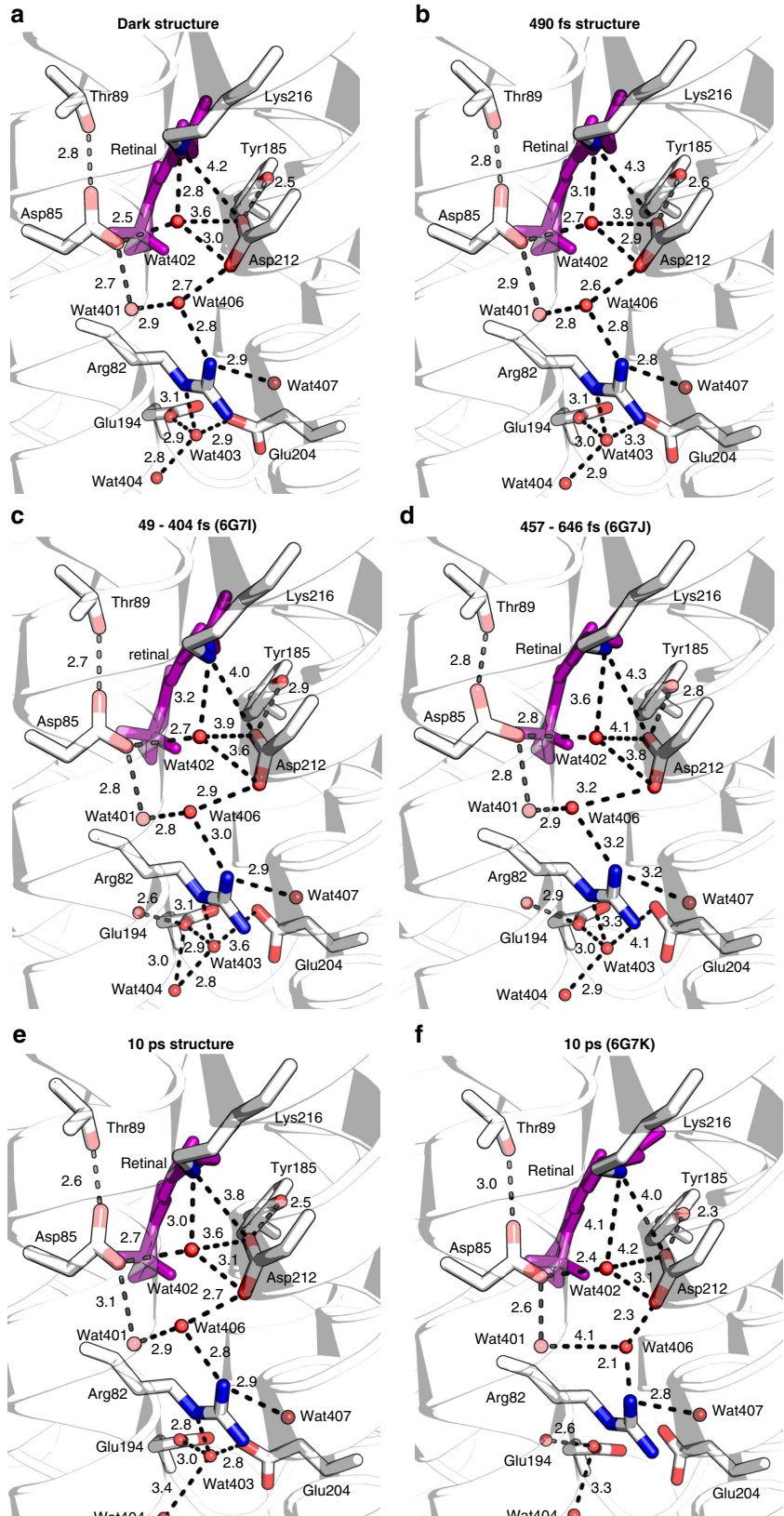

**Fig. 10** Evolution of distances between residues and waters along the proton pathway in time. Retinal is shown in pink and waters as red spheres. Distances are given in Å. **a** Dark-unrestrained structure. **b** The 490 fs structure. **c** The 49–404 fs structure from Nogly et al.[10] (6G7I). **d** The 457–646 fs structure from Nogly et al.[10] (6G7J). **e** The 10 ps structure. **f** The 10 ps structure from Nogly et al.[10] (6G7K). While similar changes in the H-bonding network occur in both studies, the elongation of the retinal protonated Schiff base and water 402 distance is much more pronounced in the Nogly et al.[10] work, as well as a disorder of water 401 and 406

the ultrafast intermediates formed after multiphoton absorption can proceed along the photocycle.

Our multi-disciplinary study addresses single and multiphoton excitation in bR. The combination of time-resolved X-ray crystallography with ultrafast molecular spectroscopy and quantum chemistry provides a new dynamic picture for bR, underscoring the coupled dynamic electronic and structural response of retinal, neighbouring amino acids and water molecules, which sets the stage for isomerization, yet observed under multiphoton excitation. Whether the observed crystallographic changes are of relevance for the biological single-photon-excited isomerization reaction needs to be studied in further experiments performed at significantly lower laser excitation intensity.

## Methods

**Sample preparation.** bR enriched in PMs of *H. salinarum* was expressed, isolated and purified as described[41, 42]. Briefly, PMs (0.9 mg ml$^{-1}$) were mixed with 1.7% (*w*/*v*) *n*-octyl-β-D-glucoside (β-OG; Anagrade, Anatrace), 50 mM K$_2$HPO$_4$/NaH$_2$PO$_4$ pH 6.9, sonicated and solubilized for 24 h. The pH was adjusted to 5.5 prior to ultracentrifugation (Ti70, 50,000 rpm, 4 °C, 30 min) and the supernatant was subjected to size exclusion chromatography (HiLoad 16/600 Superdex 75 pg, GE Healthcare) in 25 mM K$_2$HPO$_4$/NaH$_2$PO$_4$ buffer pH 5.6, 1.2% β-OG. bR eluting in the second peak was concentrated to 35–50 mg ml$^{-1}$. The protein concentration was determined spectrophotometrically at 560 nm, using an extinction coefficient of 63,000 M$^{-1}$ cm$^{-1}$. Purification and crystallization[43] was performed in the dark or under dim red light. bR was crystallized in LCP using monoolein (NuChek Prep) and gas-tight Hamilton syringes (1710 or 1725 RN) connected with a coupler (TTP Labtech). A string of LCP was extruded into a Hamilton syringe filled with precipitant (32% PEG 2000, 0.1 M K$_2$HPO$_4$/NaH$_2$PO$_4$ pH 5.6)[43, 44]. Very densely packed purple hexagonal plate-shaped bR microcrystals (longest dimension 20–30 μm, average thickness 2–3 μm) appeared in a strongly birefringent mesophase within 1–3 days. Prior to injection, the crystal-loaded mesophase (pooled from multiple syringes) was mixed with an equal volume of monoolein (Nu-Chek Prep) and diluted (4+1) with Pluronic F-127 40% (*w*/*v*) to improve flow properties of the stream[42].

**Pre-illumination.** Dark-adapted bR exists as a ~50:50 mixture of all-*trans* and 13-*cis* retinal. 100% of the functionally relevant all-*trans* retinal (light-adapted bR) can be accumulated by continuous illumination for several seconds, followed by a brief incubation in darkness. To measure the kinetics of the return of light-adapted bR microcrystals to the dark-adapted state, the crystal-loaded LCP mesophase was squeezed between two BaF$_2$ windows separated by a 1-mm Teflon spacer. Light adaption was accomplished by 30 s of illumination with a white light-emitting diode (KL 2500 LED from Schott). UV/VIS measurements were performed with a Shimadzu UV-2450 spectrometer. The decay of the visible difference absorbance signal at 600 nm is mono-exponential with a time constant of 24 min (Supplementary Fig. 15), which is similar to the value for detergent-solubilized PMs[45]. One possibility for light adaption for SFX experiments is to illuminate the crystals just before injector loading[10, 25]. However, given that the typical time needed to proceed from injector loading to actual data collection is about 15–30 min and thus similar to the dark-adaptation halftime in bR microcrystals, diffraction data would be collected on an essentially dark-adapted sample. Moreover, owing to the high optical density (OD) of bR crystals, it is necessary that the sample be sufficiently thin to allow most crystals to be exposed to light. For these reasons, we decided against performing the light adaptation of bR microcrystals prior to injector loading as described previously[25], since the sample thickness is then set by the inner diameter (ID) of the Hamilton syringe in which it is illuminated (ID 1.4 mm for a 100-μl syringe). Instead, we developed a protocol for on-line pre-illumination. To this end, samples were loaded into a 340-μl reservoir of our HVE injector[26] equipped with a 76-mm-long UV–VIS transparent capillary (100 μm ID) over which a cylindrical metal mask was slid into place. A 22-mm long window in this sleeve admitted continuous pre-illumination light perpendicular to the plane defined by the sample stream and the X-ray beam. The bR microcrystals were pre-illuminated as they flowed through this illuminated region. Since back-lighting for the side-view camera required a hole on this side of the nozzle shroud, the terminal 4–5 mm of the mask was left solid to provide a fully defined "dark zone" near the HVE exit to allow the crystals to return to the bR ground state prior to being photoexcited with an optical pump laser pulse entering collinearly with the X-ray beam (downstream of the HVE tip and passing behind the "dark zone" mask; Supplementary Fig. 16). With a sample flow rate of 1.9 μl min$^{-1}$, the stream velocity in the capillary averaged 4 mm s$^{-1}$. Thus the microcrystals spent 5.5 s in the illuminated section and 1–1.25 s in the dark section of the capillary. The 520 nm pre-illumination laser (Oxxius LBX-525-800-HPE-PP) was coupled into a continuous 10-m multimode fibre (Thorlabs, FT1500UMT, 1.5 mm core diameter, TECS clad), which entered the vacuum chamber through a custom-built clamping O-ring seal. The polished end of the fibre was positioned 90 mm from the sample stream, at which distance the intrinsic angular spread of light emerging from the

fibre slightly over-filled the 22-mm-long pre-illumination slit in the masking sleeve. By running the fibre directly into the vacuum chamber and using its polished end as the light source, optical losses were considerably reduced relative to the use of standard fibre-optic vacuum feedthroughs. The spot from the pre-illumination laser was measured to be ~4 cm in diameter at 90 mm from the fibre end. The laser power was set to 200 mW (measured output of the multimode fibre after passing through the vacuum feedthrough). The nozzle shroud itself was specially constructed with a long narrow slit on the pre-illumination side, allowing the pre-illumination light to fully fill the slit in the sleeve mask while still minimizing the amount of light entering the shroud.

To measure the pre-illumination efficacy in house or on-site in the vacuum chamber, the sample from the mounted, illuminated and aligned HVE injector was extruded into a black Eppendorf tube and immediately subjected to retinal extraction and analysis in the dark using established high-performance liquid chromatographic procedures[46]. The average composition of a non-illuminated, dark-adapted sample analysed with this method was 45% all-*trans* and 55% 13-*cis* retinal. Pre-illumination increased the all-*trans* retinal content considerably, with compositions ranging from 65% to 80% all-*trans* and from 35% to 20% 13-*cis* retinal, respectively.

**SFX data collection.** Diffraction data were collected using 9.8 keV photons. The photon pulse length was ~30 fs, the average pulse energy was 3.4 mJ and the X-ray focus was ~1.3 μm in diameter. The X-ray beam was attenuated to ~14% transmission. Taking into account the beamline transmission of ~50%, this resulted in a pulse energy of ~0.2 mJ at the sample position. Time-resolved SFX data were collected at 10 Hz with nominal time delays between the optical pump and probing FEL pulses of 0.5, 1, 3 and 10 ps. In addition, two so-called dark data sets were collected at 120 Hz from pre-illuminated but not pumped bR microcrystals to verify that the light-adapted sample had returned to the all-*trans* ground state. These data sets also served as the reference for the time-resolved data of photoexcited bR. To check whether the short-lived intermediates represent functional steps of the bR photocycle, we collected a 33-ms time-delay data set, corresponding to an M-like intermediate (Supplementary Table 7). All diffraction patterns were recorded using a CSPAD[47] detector operated in dual-gain mode with a low and high gain in the low and high resolution region of the detector, respectively.

**Stream velocity measurements.** Knowledge of the actual stream velocity is critical for TR-SFX experiments, especially in case of HVE injection, since a too low stream velocity prevents flushing of the fairly large illuminated stream slug after one X-ray pulse and prior to the beginning of the next pump probe sequence. Therefore, the actual stream velocity was determined both periodically and for each change in flow conditions (e.g. new sample) by recording short movies of the stream using an off-axis camera at a repetition rate of 120 Hz. The imaged pixel size of the camera was calibrated by imaging a capillary of known 360 μm outer diameter. In each movie, features in the extruded stream (e.g. microcrystals) were tracked over multiple frames as they moved downstream. For a given movie, such a measurement was repeated every 250–1000 frames for a total of 11–43 regularly spaced time points per movie. From these measurements, the mean stream velocity and its spread were calculated. The measurements verified that the actual stream velocity was, on average, close to the expected value of 4 mm s$^{-1}$ for a 100-μm jet based on mass conservation calculations (Supplementary Fig. 17).

**Femtosecond pump laser excitation of bR microcrystals for TR-SFX.** The bR microcrystals were photoactivated using circularly polarized 532 nm laser light from an optical parametric amplifier pumped by a Ti-Sapphire laser regenerative amplifier system. The laser pulses were stretched to 145 ± 5 fs. The laser beam had a Gaussian profile; the laser focus was 99 μm (1/$e^2$) at the sample stream position. To allow for accurate timing, the sample stream was kept at the same position along the beam direction using an optical camera. The optical laser was collinear with the X-ray beam. The centre of the pump laser was offset 25 ± 5 μm downstream of the X-ray focus for all (sub)-ps pump–probe measurements. Thereby only a shorter stretch of illuminated microcrystal-containing LCP stream must clear the interaction region before the next pump–probe cycle can be initiated. We had insufficient beamtime to perform a power titration to determine the optimal laser power and so chose a laser energy of ~5.9 μJ per pulse, resulting in a laser power density of 500 ± 50 GW cm$^{-2}$ (nominal upper limit, see Supplementary Table 7). In doing so, we knowingly erred on the high rather than the low side in order to accommodate any drift of the pump laser position resulting in poorer spatial overlap of the laser focus with the XFEL beam at the interaction region as well as to compensate losses due to scattering off the LCP stream (at the time uncharacterized). Nevertheless, by not properly accounting for the Gaussian profile, the laser power density was much higher than we intended (although previous experiments[10, 29, 38, 39] have used similar power densities). Taking into account the experimental parameters, the protein concentration in the crystals (27 mM) and an average crystal thickness of 5.9 μm, these illumination conditions result in ~21 photons per retinal on average (see Supplementary Methods, Supplementary Table 7). This value necessarily has a very large variation given the plate-like shape of the crystals (1/$e$ penetration depth is ~3.5 μm, assuming an extinction coefficient of 45,600 M$^{-1}$ cm$^{-1}$) and the fact

that we had no control over crystal orientation. Crystal shielding is unlikely since the majority of hits contained one diffraction pattern.

**Data processing**. Online monitoring of diffraction quality, hit identification, background subtraction and file conversion were performed with CASS[48]. Given the jitter in the difference between the FEL and pump laser arrival times of ~0.3 ps, the exact time delay between the pulses was determined using the LCLS timing tool[49] as described previously[38]. The sorted ~43,000 images were binned in 12 overlapping data sets of 10,000 images each[38], with average time delays of 0.24, 0.33, 0.39, 0.43, 0.46, 0.49, 0.53, 0.56, 0.59, 0.63, 0.68 and 0.74 ps (see Supplementary Methods). For indexing and integration, CrystFEL[50] (version 0.6.3) was used.

**Refinement and map calculation using extrapolated structure factors**. $q$-weighted structure factor amplitude difference electron density maps[51] were calculated using custom-written scripts. Because the occupancy of the illuminated states was only 10–15%, extrapolated structure factors were calculated, in which the occupancy of the illuminated state is extrapolated to 100%[52, 53]. Because traditional reciprocal-space refinement against these extrapolated structure factor amplitudes proved unstable, real-space refinement against extrapolated density maps was performed using Phenix dev-3063[54, 55]. Maps and structures were inspected using Coot[56] and Pymol. To assess the reproducibility of the results, all crystallographic analyses were performed independently by two researchers, using separate implementations of the $q$-weighting scheme and the structure factor extrapolation. This resulted in very similar maps and structures. Details of the crystallographic analyses are given in Supplementary Methods.

**Effect of the temporal binning on the data**. The values $y(t_{delay})$ of the dihedral angles and interatomic distances change with the average delay times $t_{delay}$ as a function of time that can be described by a function $H(t_{delay})$. However, the observed values $y(t_{delay})$ are weighted by the temporal binning of the diffraction images: Each crystallographic data set $ds(t_{delay})$ contains $n$ images $i$ collected at actual delay times $t_i$. The electron density calculated from $ds(t_{delay})$ corresponds to the weighted average of the individual conformations $Y(t_i)$ at the actual time delays $t_i$. In other words, the structural fit $y(t_{delay})$ of this electron density accounts for the binning on $Y(t)$:

$$y\left(t_{delay}\right) = \frac{1}{n}\sum_{i=0}^{i=n} Y(t_i) \qquad (1)$$

We can attempt to correct the observed values for the effects of binning directly, by making only one reasonable assumption about $Y$, namely that it consists of a harmonic oscillation (sine function) $H(t)$ that is modulated by the binning as shown above. Fitting the observed data to a sine function

$$H(t) = A \sin\left(\frac{2\pi(t - t_0)}{t_{period}}\right) + C \qquad (2)$$

yields the period $t_{period}$, the amplitude $A$ and an offset $C$. This allows us to calculate to what extent this function would have been affected by the binning:

$$h\left(t_{delay}\right) = \frac{1}{n}\sum_{i=0}^{i=n} H(t_i) = \frac{1}{n}\sum_{i=0}^{i=n}\left(A \sin\left(\frac{2\pi(t_i - t_0)}{t_{period}}\right) + C\right) \qquad (3)$$

Depending on the binning, this value will differ from the predicted value of the original function $H$ at the average time delay for each point, $H(t_{delay})$.

Thus we now know what the effect of the binning would have been on a simple sine function; to a first approximation, the amplitude of each point is scaled by the ratio between the value $H(t_{delay})$ without binning and the binned value $h(t_{delay})$. This scaling can then be applied to the actual, observed values $y(t_{delay})$ to correct the harmonic oscillator part of the actual function $Y(t)$ for the effects of binning. By plotting the results of this correction for each of the oscillations together (after subtraction of $C$ and normalization by $1/A$), this should, within the limits of the method, reveal the values $H(t_{delay})$. If $H(t_{delay})$ corresponds to a damped sinusoidal function, the binning-corrected extrapolated values are underestimated and correspond to the lower bounds. An estimate of the damping constant can be obtained by fitting $A \sin(2\pi t \tilde{\nu} c + \phi)e^{-at} + b$ to $H(t_{delay})$ where $A$ is the amplitude, $t$ is time, $\tilde{\nu}$ is the wavenumber, $c$ is the speed of light, $\phi$ is the phase, $a$ is the damping constant and $b$ is the offset.

A different method of estimating the decay constant uses the explicit assumption that the function $Y$ underlying the observed data is in fact an exponentially decaying harmonic oscillation:

$$Y(t) = A(1 - e^{t/\tau})\sin\left(\frac{2\pi(t - t_0)}{t_{period}}\right) + C \qquad (4)$$

Applying the effect of the binning to this function yields:

$$y\left(t_{delay}\right) = \frac{1}{n}\sum_{i=0}^{i=n}\left\{A(1 - e^{t_i/\tau})\sin\left(\frac{2\pi(t_i - t_0)}{t_{period}}\right) + C\right\}. \qquad (5)$$

This latter function can then be fitted to the observed amplitudes, yielding a value for the decay constant $\tau$, using the estimate obtained by the previously described method as a starting point for optimization.

**Microcrystal preparation for spectroscopic experiments**. bR microcrystals grown in LCP were pooled directly from Hamilton syringes into a 1.5-ml centrifuge tube and gently centrifuged to collect the LCP in an upper layer above the precipitant. The LCP was then transferred into a 300-μl tube and centrifuged for 1–3 h at maximum speed in a table top centrifuge. Residual precipitant was at the bottom and the upper LCP layer was separated into a lower layer highly enriched with crystals and a top layer consisting of LCP essentially without crystals. After removing the latter with a spatula, the enriched microcrystalline layer was used for spectroscopy. For IR measurements in $D_2O$, precipitant was first removed using a syringe from 200 to 500 μl of the LCP collected by gentle centrifugation. Then it was resuspended in 1.5 ml $D_2O$ crystallization buffer (32% ($w/v$) PEG 2000, 0.1 M $K_2HPO_4$/$NaH_2PO_4$ pD ~ 5.6) by vortexing and recollected by gentle centrifugation. This step was repeated 3–5 times and then an enriched crystal layer was prepared as described above.

**Ultrafast VIS spectroscopy on bR microcrystals**. Enriched bR microcrystals in LCP were mixed with monoolein and F-127 until clear (2.5+1+1) and mounted between two $CaF_2$ windows using a 50- and 100-μm spacer for the crystals and PMs at pH 5.6, respectively. The experimental set-up was essentially as described in ref. [11]. The pump pulse wavelength was centred at 535 nm with a full-width at half-maximum (FWHM) of 13 nm with varying energy. The pump pulse at 535 nm was generated by cascaded tuneable optical parametric amplifiers (OPAs) with a FWHM of 13 nm and pulse duration of $(130 \pm 20)$ fs and a diameter on the sample of 150 μm. The maximal absorption around 570 nm was 0.15 OD and 0.3 OD for the LCP crystals and PMs, respectively. We used our Ti:Sa fundamental at 810 nm to generate super-continuum probe pulses by focussing the fundamental into liquid water[11]. Transients in the spectral range from 410 to 719 nm were detected with a spectral resolution of about 1.5 nm. Pump energies at 535 nm were varied from 0.2 to 1.0 μJ and from 0.4 to 1.5 μJ for the LCP crystals and PMs, respectively. The polarization of the pump beam was chosen to be parallel to the polarization of the probe beam, and a notch filter at 537 nm was used to suppress scattering. A step size of 50 fs was used for the −0.5 to 4 ps delay times; logarithmic steps were used for longer delay times.

**Ultrafast IR spectroscopy on bR in PMs and microcrystals**. Highly enriched bR microcrystals in LCP and PMs were mounted between two $CaF_2$ windows each using a 25-μm spacer. Visible pump beams at 535 nm were generated by cascaded tuneable OPAs generating fs output pulses of ~6 μJ and $(130 \pm 20)$ fs. In detail, a collinear BBO OPA pumped at 400 nm generated near-IR pulses at 1070 nm, which were amplified by two KTA (potassium titanyle arsenate (KTiOAsO4)) amplification stages pumped with the fundamental at 800 nm. The 1070 nm pulses were frequency doubled in a BBO crystal. The pump pulses at 535 nm exhibit a FWHM of 13 nm and were sent through a filter (Schott KG4) and a polarizer to filter near-IR frequencies. A $\lambda/2$ plate was used to set the polarization of the pump beam at the magic angle with respect to the probe beam. The diameter of the pump beam on the sample was $(220 \pm 20)$ μm, the diameter of the two probe beams on the sample was $(120 \pm 20)$ μm. IR probe beams with energies <50 nJ were generated as reported elsewhere[57]. Two reflections of the fs mid-IR pulse were taken as probe beams at the same time in the same sample volume to detect absorbance changes. One probe beam was used to probe the sample spot 1.5 ns before and the other beam fs to ps after pump excitation. The system response was about 400 fs. Probe pulses were dispersed with an imaging spectrograph and recorded with a $2 \times 32$ element MCT array detector. The spectral resolution was 2–3 cm$^{-1}$. The sample was moved with a Lissajous scanner to ensure a fresh sample volume between the pump beams. The power titration for excitation energies of 0.3, 0.7, 1.4 and 4.4 μJ was performed under the same laser set-up conditions with samples from one batch and nearly identical absorption of about 1 OD at 535 nm. Here the non-systematic errors are small. Measurements with different pump–probe overlapping change the absolute signal values presented in Fig. 2. Several measurements in the spectral range from 1490 to 1570 cm$^{-1}$ upon excitation at 535 nm were performed at different excitation energies (0.3–4.4 μJ). Each transient measurement was repeated several times (scans). Each scan consists of 187 delay time points, before and after time overlap of pump and probe pulses. Every single data point at a specific delay time in a scan is the averaged result of 600 shots. Time zero was determined by the sigmoidal rise of the signal in a Germanium wafer. Measurements with excitation energies of 0.3 μJ were performed in one sample with 70 scans, resulting in an average of about 42,000 shots per data point. In these measurements, no change of the sample was observed. Measurements at 0.7 μJ were performed with two samples and 30 scans per sample resulting in 36,000 shots per data point. For longer measurement times, the sample starts to change its colour. With increasing pump energy, the sample was exchanged sooner. At 1.4 μJ excitation energy, 6 samples and 8 scans per sample were used resulting in 28,800 shots per data point. At 4.4 μJ, eight samples were investigated. We analysed the first or the first two scans (15 scans in total) resulting in 9000 shots per data point. Sample degradation, i.e. a loss in visible

absorption, starts moderately at 0.7 μJ and is significantly sped up at higher excitation energies.

**Light scattering from a cylindrical sample jet**. The refractive index of LCP was determined with a refractometer to be 1.42. Using this value, scattering of light from the surface of a circular LCP jet was determined with ray tracing simulations. A collimated light beam perpendicularly impinged on a 100-μm diameter jet cross-section and the reflection was calculated using Fresnel equations as described in Supplementary Note 1.

To determine the scattering of light by LCP containing bR microcrystals, the transmittance of an empty cuvette, a cuvette filled with a 65% glycerol solution (refractive index 1.42), with LCP prepared from monoolein and water supplemented with 4+1 with F-127 and with bR crystals in LCP prepared as used in SFX, respectively, was measured using a Jasco V-760 spectrophotometer. To disentangle the absorption and scattering components of the signal due to passage through bR microcrystals in LCP, the crystals were bleached as described in Supplementary Note 1. In total, ca. 20% of incoming photons are lost due to scattering (14% by crystals, 2% by LCP) and reflection (5%) at the various interfaces (see Supplementary Note 1). This result does not agree with the 80% scattering loss reported by Nogly et al.[10], who did not, however, present any details as to how their value was obtained.

**QM calculations**. A cluster model of the active site was prepared using the coordinates of the light-adapted dark state. In addition to the RPSB chromophore, residues Tyr57, Arg82, Asp85, Trp86, Thr89, Trp182, Tyr185, Asp212 and Lys216 and four water molecules Wat401, Wat402, Wat406, Wat407 were included in the model (Fig. 3a). Standard protonation states of the groups forming the bR active site were assumed. The final cluster model consisted of 208 atoms and had a net charge of zero. The geometry was optimized in the ground state at the B3LYP/cc-pvdz-D3 level of theory[58–61]. To mimic the constraints of the active-site fragments in the protein, the pairwise distances between selected terminal atoms were kept as in the starting geometry during the geometry optimization. The geometry was optimized in delocalized internal coordinates to the maximum energy gradient component being <0.0001 Hartree/Bohr. Excited-state geometries were obtained using the TD-DFT-B3LYP/cc-pvdz-D3 geometry optimization. The same geometry constraints as for the ground-state geometry optimization were applied. The retinal $S_1$ and CT(W86) states were optimized to the maximum energy gradient being <0.0003 Hartree/Bohr. The geometry optimization in the CT(Y185) state was performed until the CT(Y185) state crossed with the $S_0$ state. Therefore, relaxation of the CT(Y185) state is probably underestimated compared to the relaxation of the retinal $S_1$ and CT(W86) states.

At the optimized geometries, the excitation energies were computed with the XMCQPT2- CASSCF(8,5)-SA5 method[62]. The CASSCF active space (8, 5) included 8 electrons in 5 molecular orbitals (MOs), i.e. the HOMOs of RPSB, Tyr185 and Trp86, HOMO-1 of Trp86 and LUMO of RPSB (Supplementary Fig. 8). Five states were computed: the closed-shell $S_0$ state and four excited states dominated by single-electron excitations from the four occupied MOs included in the active space to the LUMO of RPSB. CASSCF energy averaging was performed with equal weights for the five states (-SA5). Such five state solutions were obtained for all geometries of interest, enabling straightforward energy comparison. The effect of the active space selection on the computed energies is discussed in Supplementary Note 2, see Supplementary Tables 8 and 9.

In order to estimate the excited-state energies of Trp86, we performed XMCQDPT2'-CASSCF(12,12)-SA20 calculations at the geometries optimized in the ground state ($S_0$-min), in the retinal $S_1$ state ((Ret)$S_1$-min) and in the CT(W86) state (CT(W86)-min). The CASSCF active space (12, 12) consisted of 12 electrons and 12 MOs, of which 4 electrons and 4 MOs correspond to the RPSB and 8 electrons and 8 MOs correspond to the indole moiety of Trp86. State-averaging was performed for 20 states with equal weights (-SA20); all these 20 states were included in the XMCQDPT2 calculations. As we used a rather large cluster model, lack of size consistency of the quantum chemical method may become a source of errors. To minimize such errors, we considered the energies obtained with the approximately core-separable XMCQDPT2'[62] theory in Fig. 4. The original XMCQDPT2 and core-separable XMCQDPT2' results for different active spaces and sizes of the cluster model are compared in Supplementary Note 2. The XMCQDPT2 and XMCQDPT2' energies are presented in Supplementary Table 10. The calculations were performed using a denominator energy shift[63] of 0.002 Hartree to avoid the intruder-state problem. The standard cc-pvdz basis set[58] was used throughout.

Using the data obtained from the XMCQDPT2 calculations, the electronic coupling $V_{el}$ between two states of interest was estimated using the generalized Mulliken–Hush scheme[64] according to the formula

$$V_{el} = \frac{\mu_{12}\Delta E_{12}}{\sqrt{d_{12}^2 + 4\mu_{12}^2}} \qquad (6)$$

where $d_{12}$ is the difference of the dipole moments of the two states, $\mu_{12}$ is the transition dipole moment matrix element of the two states and $\Delta E_{12}$ is the two-state energy difference at a given geometry.

To obtain a model of the retinal at 10 ps after photoexcitation, a larger cluster model consisting of Tyr57, Arg82, Gly83, Ala84, Asp85, Trp86, Thr89, Thr90

Trp182, Tyr185, Asp212, Lys216, Wat401, Wat402, Wat406 and Wat407 (241 atoms, net charge zero) was used in addition to the above described 208-atom cluster. The starting coordinates of the non-hydrogen atoms were taken from the initial SFX model of 10 ps structure. The geometry of the smaller (208 atom) and larger (241 atom) clusters containing the distorted retinal was optimized in the ground state using the B3LYP/cc-pvdz-D3 method in delocalized internal coordinates using the pairwise distance constraints for selected terminal atoms similar to the optimization of the dark-state structure. None of the atoms comprising the conjugated π-system of RPSB was constrained. All quantum chemistry calculations were performed using the Firefly program version 8.0 (http://classic.chem.msu.su/gran/firefly/index.html), which is partially based on the US GAMESS[65] source code.

**Reporting summary**. Further information on research design is available in the Nature Research Reporting Summary linked to this article.

## Data availability
Structures and diffraction data have been deposited in the protein databank. The accession codes are 6GA1 for dark-state cell 1, 6GA2 for dark-state cell 2, 6RMK for dark unrestrained state, 6GA3 for the 33-ms structure, 6GA4 for 1 ps, 6GA5 for 3 ps, 6GA6 for 10 ps, 6GA7 for 0.24 ps, 6GA8 for 0.33 ps, 6GA9 for 0.39 ps, 6GAA for 0.43 ps, 6GAB for 0.46 ps, 6GAC for 0.49 ps, 6GAD for 0.53 ps, 6GAE for 0.56 ps, 6GAF for 0.59 ps, 6GAG for 0.63 ps, 6GAH for 0.68 ps and 6GAI for 0.74 ps. The Cartesian coordinates of the QM-optimized cluster models are available upon request from Tatiana Domratcheva. Raw diffraction data will be made available at cxidb.org. The source data underlying Figs. 1b, c, 2a–d, 6b, c, 7a and 8b–i and Supplementary Fig. 11a–j are provided as a Source Data file. Other data are available from the corresponding authors upon reasonable request.

## Code availability
The custom-written scripts used to calculate the Q-weighted structure factor amplitude difference Fourier maps are available from J.-P.C. and T.R.M.B. on request.

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

## Acknowledgements

Portions of this research were carried out at the Linac Coherent Light Source, a National User Facility operated by Stanford University on behalf of the US Department of Energy, Office of Basic Energy Sciences. Parts of the sample injector used at LCLS for this research were funded by the National Institutes of Health, P41GM103393, formerly P41RR001209. Use of the Linac Coherent Light Source (LCLS), SLAC National Accelerator Laboratory is supported by the U.S. Department of Energy, Office of Science, Office of Basic Energy Sciences under Contract No. DE-AC02-76SF00515. We thank the support staff at LCLS. We acknowledge support from the Max Planck Society. We thank Michel Sliwa and Rolf Diller for discussions, Dorothea Heinrich, Kirsten Hoffmann and Elisabeth Hartmann for preparation of purple membranes and Miroslaw Tarnawski, Katerina Dörner and Grant Mills for assistance at LCLS. The work was supported by the Max Planck Society, the Agence

Nationale de la Recherche (grant ANR-17-CE11-0018-01) to J.-P.C. and the German Research Foundation through the SFB-1078, projects A1 and B3 to J.H. and B4 to R.S. and B7 to K.H. and project HE 5206/3-2 to K.H.

## Author contributions

G.N.K., R.B.D., T.D. and I.S. designed the experiment; R.S. provided the bR samples; G.N.K. purified bR; G.N.K. and R.L.S. performed the retinal HPLC analysis; G.N.K. and R.G. crystallized bR. G.N.K., M.L.G., D.E. and J.H. performed spectroscopic characterization of the light adaptation in crystals, T.S., Y.Y. and K.H. performed ultrafast transient UV–VIS and IR spectroscopy for which G.N.K. prepared the samples; S.H. contributed to the analysis; G.N.K., M.L.G., R.L.S., M. Stricker and R.B.D. established the pre-illumination protocol; M.L.G., M. Stricker, G.N.K., I.S. and S.H. analysed the light scattering contribution, T.R.M.B., A.B., S.B., S.C., R.B.D., L.F., A.G., M.L.G., M.H., M.K., J.E.K., G.N.K., C.M.R., R.G., M. Seaberg, M. Stricker, I.S. and R.L.S. collected the SFX data; M.H., C.M.R. and L.F. processed the SFX data; J.P.C. and T.R.M.B. analysed the SFX data; G.N.K., J.-P.C. and T.R.M.B. interpreted the electron densities; G.N.K., J.P.C., T.R.M.B. and I.S. analysed the structures; T.D. performed quantum chemistry calculation; J.R., S.H. and J.H. contributed to discussions; G.N.K. and I.S. wrote the manuscript with input from all authors.

## Additional information

**Competing interests:** The authors declare no competing interests.

