## [Peer Review File · Nature Communications]

Reviewers' comments:

Reviewer #1 (Remarks to the Author):

First, although this review poses many questions, most general, some specific, I do not have definitive answers to them myself. We are all exploring new territory; but it's useful to realize that at this stage, conclusions must remain tentative.

As the authors state, "This double bond isomerization [of the retinal chromophore in rhodopsins and bacteriorhodopsin] is one of the most studied processes in photobiology." The interest is even more general. The explicit structural details of how isomerization is accomplished are not well understood in chemical terms, particularly in a complicated molecule with many double bonds. And in the present case, the details - or even the fundamentals - differ between free retinal and retinal incorporated in bacteriorhodopsin; retinal - protein interactions are evidently key. So, the wide significance of and interest in this study is in no doubt whatsoever.

The problem lies in the severe experimental and data-analytical difficulties of examining the structural basis of this reaction on tiny crystals, illuminated/pumped by a high power visible laser and probed by an exceptionally high power hard X-ray laser. The authors are well aware of (at least most) of these difficulties, describe heroic attempts to overcome them, and present a rather detailed atomic model for the time course of isomerization on the time scale from 240 fs to 10 ps. This time scale, unlike an earlier study by Nango et al., covers the critical phase of isomerization itself.

Unfortunately, I'm not convinced by their detailed structural interpretation. The most I will concede is that their structural model MAY be consistent with their X-ray observations, though here too there are difficulties I note below. No model based on dynamic experiments can yield a unique mechanism and hence a unique set of intermediate spectra or structures. That's only possible if all competing mechanisms are considered and all but one shown to be incompatible with the data. Very rarely attempted let alone accomplished. Discriminating among competing mechanisms then requires external data e.g. chemical plausibility or (as here) calculations; or simply Occam's Razor. The last normally leads to the simplest linear mechanism, A -> B -> C..... That's fine if the system is pretty much near equilibrium throughout; but highly dubious when the reaction is initiated - as in all photoreactions - by absorbing an energetic photon which drives it far from equilibrium.

Is this a counsel of despair? No; but it does mean that all conclusions have to be cautiously phrased, and not leave the impression that they are the definitive truth.

The authors seem to realize this when they state (in specific reference to their SVD results (main text p. and Supplementary Results p.6, but applicable to the interpretation more generally) that "...these contributions by themselves should on no account be interpreted as separate species, modes or events. These results do establish that a major change occurs in the structure at around 0.5 ps". I readily agree with these two sentences. In fact, the second is the most conservative statement of their results.

Substantial experimental problems:

Massive averaging of intensities/amplitudes necessitated by weak scattering compromised here by anisomorphism of the crystals: different cell dimensions. Was a given crystal batch always of one form or the other? Or, did a run on one batch display both crystal forms, so that individual diffraction patterns had to be assigned to one form or the other? If this assignment is in error, what are the consequences?

Necessity to maximize light-driven signal by an unusual illumination protocol. Yield is increased, but leaves this reader uneasy that artefacts might have been introduced, perhaps by slow, unsuspected structural changes in the seconds range. What evidence these do not exist?

Possibility - even likelihood - of radiation damage from the high power pump visible laser and/or the probe X-ray laser. Presence could not be investigated e.g. by a power titration. How might conclusions be affected?

Certainty of large variation from crystal to crystal of the extent of isomerization arising from range of crystal sizes, orientation in the laser beam, and position in the sample stream. These tend to mask the inevitably small time-dependent changes in structure amplitudes in the fs to ps time range. Consequences?

Substantial problems in data analysis :

Several novel difficulties arise in refining a light-driven structure in the fs to ps time range which certainly affect and may compromise the final interpretation.

As the authors note, time-dependent difference maps can be hard to interpret if the structural differences are complex, as here. Much then depends on appropriately converting these to more readily interpretable derivative maps, based on extrapolated Fourier coefficients. The authors do this carefully, explore many parameters and approaches, and describe this in detail. Their maps seem plausible and exhibit time-dependent changes to the eye. To me, it's unexpected that they would be so prominent.

In this type of time-resolved experiment that (probably? involves incoherent evolution of structures, a time-dependent mixture of structural states is present at all time points. A priori the number of states, the mechanism they comprise, their individual structures and the time course of their evolution are all unknown. In this reaction there is a wealth of ultrafast spectroscopic data in solution which the authors show is quantitatively closely similar to data on their crystals. These data have been interpreted as arising from a linear mechanism comprising spectroscopically distinct states I, J, K, L...., which at least provides a starting framework for interpreting their time-resolved structural data. However, one spectroscopic state could encompass many different structural states; a single electronic state of the retinal could be coupled to many protein structures/states. Conversely, electronically distinct spectroscopic states could have such small structural differences between them that they're crystallographically indistinguishable. How to decide what the mechanism and states are in fact in the crystallographic data? Might deviations from the spectroscopically-derived model be evident in their data? How to decide how many states are present in any time point?

The authors chose to refine a single structure into each extrapolated map, in which that single structure is in some sense an average over the unknown, individual structures actually present. How would breakdown of the implicit assumption that a single structure can represent a structurally heterogeneous mixture affect the structural conclusions on the mechanism of isomerization?

After absorbing a visible photon, the retinal is highly excited - "hot", with a vibrational temperature equivalent to many hundreds of deg K. Bond lengths and torsion angles may change from that effect alone, even with no isomerization. Atoms are "smeared out"; and there are no standard bond lengths and angles available to conduct crystallographic refinement in real space. Yes, the system rapidly cools - but how fast? What "standard" bond lengths and angles were used (if any)? What are the effects on refinement of this temperature jump and the large deviation from equilibrium?

With all these effects that conspire against confident detection of small structural changes, I'm not convinced that the oscillations are real; few cycles, poor signal-to-noise. Shouldn't the also be evident in one or more of the right singular vectors in the SVD? Particularly if SVD were applied only to the volume including the chromophore and the immediately surrounding protein? (I didn't see where the volume actually used was given.)

Finally, isomerization has been studied by closely similar pump-probe FEL techniques in single crystals of photoactive yellow protein (ref 19). Their results do not display oscillations, but agree in showing a major structural transition in the chromophore at around 500 fs which the authors attribute to isomerization. It's plausible that the earliest steps of isomerization are identical - or at least closely similar - in all systems. Worth adding a sentence?

Reviewer #2 (Remarks to the Author):

The authors report a time-resolved x-ray diffraction experiment at a free electron laser of bacteriorhodopsin (bR), in which they recorded diffraction pattern from ~250 fs up to 10 ps (and 11 ms) after photoexcitation with a 150-fs laser pulse. Structural refinement against extrapolated difference densities enabled them to visualize the time-series of structural changes affecting both the retinal co-factor and the nearby protein residues. Interestingly, the authors could resolve (coherent?) vibrations of modes involving atoms in and around the retinal, and linked these to oscillations seen in various spectroscopy signals. The refined structures suggest a mechanism of

isomerization that is very much in line with previous computational predictions, but not quite the same. Since this is the first time the sub-ps regime of the photoactivation in bR is addressed with tr-sfx, I think the results will be of very high interest to the community and beyond. I therefore would very like to see this study appear in Nature Communication, but not before a number of (small) concerns, in particular about the computational aspects, have been addressed.

Although the paper is all about visualizing the first event in the bR photocycle, there is no image showing the complete isomerization process. Instead, the time evolution of the retinal conformation has to be inferred from the text and from combining figures 2 and 3 that show the retinal pocket from different angles. I suggest therefore to include a figure showing the complete time-evolution of the isomerizing retinal (corrected for the 2:1 branching ratio) and environment from $t=0$ to $t=10$ ps. Also in the same figure I would like to see the (corrected) torsion angles in the retinal, again from 0 to 10 ps. Currently, supporting information Figure 8 only shows information on the sub-ps timescale, making it difficult to see when (or how) the actual isomerization occurs.

The authors have tried multiple approaches to refine intermediate structures against the experimental data. This makes the paper (methods and SI) a bit difficult to read, but at the same time increases confidence. However, perhaps due to the many details, some aspects were lost on me. For example, the authors write that for the refinement of the 10 ps intermediate, they used geometric parameters from the QM calculations (page 6). However, in the description of the QM methodology, they write that "The starting coordinates of the non-hydrogen atoms were taken from the refined SFX model" (page 16). If the QM parameters could only be obtained after the refinement, while the refinement relied on the QM parameters, this must have been an iterative process? If so, can the authors rule out a potential QM bias in these refinements? I think some additional details on how the QM results were used for the refinements and vice versa is needed here. In particular, because the QM and refined models seem not too similar, with a difference in the C11=C13 torsion angles of 20 degrees (refined SFX: -61, QM: -41).

Furthermore, to show that there is indeed a good agreement between the QM-optimized 10-ps structure and the refined 10-ps SFX structure, I suggest to include a new figure, in which the two structures are superposed, perhaps in Supporting Information. This will make it a lot easier for the reader to see how well the two models agree. In the current version no figure(s) of the QM optimized 10-ps structure is included anywhere, unless I've missed that.

As in previous SFX studies, a very high pump laser power was used. Although my concerns are relieved by the agreement between the 33 ps refined structure and a previously published structure of late-M intermediate (Fig. S1), it might be informative to compare also the difference density maps at 33 ps with the map of that late-M intermediate.

The authors have also performed a series of quantum chemistry calculations to address what happens prior to the first measured timepoints ($< \sim 250$ fs). Unfortunately, I have not completely understood the role of these calculations in the present study. What time point do these computations represent, and are the structural changes seen in the optimized excited states comparable to changes observed in the refined structures?

The calculations suggest that rather than photo-isomerization, electron transfer from nearby Trp86 and Tyr185 residues into the retinal occurs instead. Although the involvement of charge transfer (CT) states in the photo-chemistry of bR had been suggested before (see e.g. Hayashi et al, Biophysics 8 (2012) 115), there seems no support from experiment (yet). Instead, previous experiments, including the present one, as well as computational studies, all suggest that retinal photo-isomerizes in the S1 excited state. Although in most earlier computational studies only retinal was included in the QM region (and thus excluding the CT states), isomerization timescales that are very much in line with time-resolved spectroscopies were observed. Population of CT states, therefore, seems not necessary for an efficient isomerization and intuitively, I'd even expect these CT states to reduce rather than enhance the isomerization. Unfortunately, the authors do not discuss by what mechanism the CT states contribute to the photo-activation. In particular, it would be important to discuss (or speculate at least) how population of CT state would lead to the 13-cis retinal configuration observed at 10 ps.

Along the same lines, the authors state that the coherent oscillations are due to crossings between the S1 and the CT states (page 9 manuscript, and page 12, SI). I, however, don't see the connection. How would spreading of population over multiple excited states lead to coherent nuclear vibrations? Please explain.

Also, unless I got it wrong, the authors speculate that the CT states are involved in the proton translocation process later in the photocycle (page 10). How? Are the authors suggesting that these CT states are that long lived to affect the dynamics in the L state?

The authors observe that in the CT states the excited state dipole moment is much larger, and refer to Science 309 (2005) 917 and PNAS 106 (2009) 7718 (refs 8 and 38) to support this finding. However, I have understood that in those works, the changes in Trp absorption were used to probe the charge migration in the retinal, while here, it is claimed that the Trp donates an electron and is thus actively involved in the photo-activation. Would the formation of the CT state lead to similar changes in absorption signals of TRP as seen in those previous works? If so, this could be support for the involvement of the CT states. I suggest therefore to compute the absorption of the CT state and see if it fits the previous observations.

Can the authors rule out that the charge transfer (CT) state are only accessible due to a rather limited active space, which includes exactly those orbitals required to get the CT states? To rule out potential artifacts, can the authors include more states (and orbitals) and then compare the energies, as was done for instance in JCTC 8 (2012) 1912 for GFP chromophore pocket?

I think the original GMH approach was first described in Chemical Physics Letters 249 (1996) 15-19, rather than in ref. 60, which is sort of a review of multiple approaches to compute couplings.

Reviewer #3 (Remarks to the Author):

The manuscript entitled: "Three-dimensional view of ultrafast dynamics in photoexcited bacteriorhodopsin (bR)" by Kovacs represents an impressive and highly innovative approach to femtosecond time resolved crystallography. The structural correlation to known excited state dynamics are similar in time scales, however, there are some notable exceptions. The authors have advanced theoretical studies that seem to indicate that charge transfer in the excited state involves the protein residues and this can explain some of the discrepancies with the current picture for the excited state dynamics of bR. This new picture could have relevance to other photoactive rhodopsin systems. There are however a number of technical issues that seriously question the conclusions of this work. It needs to be stated that this is a real tour de force for the field. I support publication in Nature Communications but there needs to be some reconsideration of the main conclusions especially in light of a few points that limit the ability to draw the conclusions made.

The main concerns are:

1. The oscillations seen in the data involving assignments to structural induced changes that are biologically relevant needs to be challenged. First, there needs to be a direct comparison to the actual frequencies that are known to the Franck-Condon active modes that involved in the initial optical excitation and are anharmonically coupled along the motion to the conical intersection (CI) to the coupling (C13=C14 bond elongation) and tuning modes (low frequency torsion) directing the key photoisomerization step. The authors should compare what reactive modes have been observed using optical, Raman, and impulsive Raman detection to those they observe and the assignments. (Note the reference in relation to coherent motions by Matthias et al is incorrect as this was for rhodopsin.) Please note that the key modes that have been identified so far for bR are the HOOP mode at around 860 cm⁻¹, the low frequency skeletal torsion at 200-230 cm⁻¹, and higher frequency modes involving C=C stretch around 1600 cm⁻¹. None of these are consistent

with the motions observed.

2. The very fact that the authors observe oscillations in their diffraction data that involve nearby residues points to artifacts. The motion to the CI and associated change in the time evolving potential is too slow to impulsively excite the specific vibrations observed (600fs is not sufficiently short driving force for such motion – see further below). The observation of coherent oscillations in optically observed spectra are due to the short pulse excitation of vibrational wavepackets on the excited state surface (displacive excitation) or ground state Raman processes (spectator modes.) To observe surrounding protein residues oscillate is in contradiction to other measurements as there is no transition dipole moment coupled to these motions. The force would have to come from the reaction itself not some superimposed optical preparation step. The force then must be evolving on a time scale shorter than $\frac{1}{2}$ period of the excited vibration. It is telling that the reported oscillations are effectively the pulse width. The time scale of the observed oscillations is rather consistent with their pulse duration and associated photophysical processes. This observation and induced motions of waters, polar residues is consistent with ionization of the retinal chromophore, a well known problem with the peak power conditions used for this experiment. I don't see how a 500 fs-600 isomerization process can impulsively excite 100 cm^{-1} modes involving neighbouring residues in the protein. (The argument may be to emphasize charge transfer contributions but then this should have been seen previously spectroscopically – see below).

3. Given the above, it is essential the authors go back and thoroughly do a power dependence. They assert they do not have peak power problems with their excitation conditions, as they observe similar dynamics as their pump probe studies. (The authors need to state the excitation conditions for the solution phase and crystal fs studies in figure 1c.) The authors need to redo the all optical pump-probe studies of the crystals under the exact same conditions as the XFEL studies and they must directly compare transient spectra. This means a global fit using decay associated spectra to the data to ensure they are seeing the same excited state dynamics as the biologically relevant processes. The dynamics may be somewhat different but the related decay spectra should be the same.

The red edge of the spectra are most useful in this comparison. There will likely be a red shift due to ionization of the retinal in a fully resonant multi-photon process. The peak power of the excitation is reported to be $>250 \text{ GW/cm}^2$. This peak power is enough for nonresonant multiphoton ionization of water never mind fully resonant multi-photon artifacts. The fact that the x-ray probe spot size was on the order of 1 micron relative to the 100 micron excitation spot size means that the peak power probed by the x-rays is related to the centre of the laser excitation not the spatially averaged intensity. The locally probed peak power is likely over 500 GW/cm^2 . There are some beam position overlap issues pointed out but the power of $>250 \text{ GW/cm}^2$ at least up to 500 GW/cm^2 should be studied to be sure there are no artifacts. For completeness sake, the authors should identify the peak power where changes occur ($> 500 \text{ GW/cm}^2$ if need be).

4. On a related issue, the authors report the laser excitation conditions correspond to 1 photon absorbed per retinal chromophore. It is not possible to obtain 100% excitation and this point is again part of the problem. There is a units problem in their reported extinction coefficient. I calculate at the excitation conditions used there are well over 1 photon per retinal (unless the protein concentration is off by an order of magnitude). The number of absorbed photons points out the problem again, without even considering peak power problems.

To get this correct, one has to include stimulated absorption from the excited state to higher lying electronic states with nearly the same oscillator strength as well as stimulated emission during the pulse duration (with exactly the same oscillator strength with minimal motion along the CI). To get the correct fraction excited is nontrivial and this number is critical to how they inferred the fraction excited and ensuing branching ratio.

The authors need the fraction excited to analyze their data. They must do a proper estimate of the true fraction excited at the peak power used and then apply the same logic in branching ratios. The problem will be taking into account the excited state absorption. The fraction excited, independent of light adaption issues, is not 20 as reported%.

5. The above problem is important as the structural changes depend critically on the assumed fraction excited. This reviewer is concerned about multiphoton ionization artifacts that would better explain their data and motions of residues in the protein. The excited state potential is localized on the retinal (see below re: theory). Some discussion of the importance determining the fraction excited is needed as this greatly affects the extrapolated changes in the structure as pointed out by Coppens et al. The crystal thickness varies, as does the fraction excited in the probed crystal volume. This effect and multiphoton absorption will significantly affect the true fraction excited that undergoes the photobiological process of interest. The authors need to examine the shot to shot variation on the fraction excited and how this would affect their inferred structural changes. There is no way around this problem affecting the results. It may be impossible to determine the fraction excited but the effect on the inferred structural changes with error bars for the fraction excited needs to be clearly stated with a more rigorous determination of the fraction excited.

Some analysis of the variance in the fraction excited and the effect on the extrapolated structural changes is essential for readers to properly gauge the degree of confidence in these reported structural changes. The fraction excited can not simply be assumed at these peak powers and then use the cis-trans branching ratio to disentangle the differential changes in diffraction intensities from the dark controls.

This point is doubly important as there are also significant variations in the light adapted bR process they did not take into account. The crystal thickness is too thick on average to uniformly illuminate the entire sample volume. Most of the light is absorbed at the front of the crystal, which will block most of the light from properly adapting the back side of the crystal in the light adaptation process. This point is important despite as it will affect the fraction of ground state trans and cis as an additional question on their observed structural changes.

The real problem is likely the femtosecond laser excitation but the entire problem needs to be treated with some analysis of error in the structure determinations.

6. The theory is a welcome contribution to the problem. It is interesting to see some ideas that there may be charge transfer processes involving the surrounding protein as part of the protein structure optimization. However, charge transfer processes depend critically on the bath as this normally involves long range correlated motions, many small motions, in medium (protein in this case) repolarization of the charge separated product state. It is difficult to understand a strong charge transfer character without a more realistic model for the protein. The wavefunction overlap at a specific site or residue will be greatly reduced by a more complete treatment of the protein. I believe this level of theory is in response to the experiment and not a prediction in advance of the experiment per se. The magnitude of the motions observed experimentally are quite large. These observations may be more in line with multiphoton ionization of retinal than charge transfer contributions to the residues.

All previous experimental and theoretical work view the process to be driven by the excited state potential (with protein involvement at this level) with relaxation central or localized to the retinal chromophore. It seems the argument is for a much stronger charge transfer process not within the retinal but to the protein itself, which would then now involve polar residues. Such an effect should be seen experimentally as very different in terms of transient spectra. It is for this reason I am also asking for a comparison of the observed oscillations inferred from the structural changes (diffraction analysis) to that observed previously with all optical methods. Spectroscopy should definitely observe these effects in transient spectra if not the specific motions involved. The electronic states would be very strongly modulated by such a CT process in contrast to the great similarity of retinal (and analogues) photophysics in a large number of different systems (different proteins and solvents where this CT process would not occur).

The above points cover the major concerns. The authors must go redo their control experiments as this is a critical point. This is not an unreasonable request as it requires no additional XFEL beam time. This study can be done in labs currently set up within this collaboration for such studies. It is not sufficient to do all optical pump-probe transient absorption studies on their crystals to assert that based on the similarity to the observed dynamics (low power) that these results (at orders

higher peak power) can be compared to solution studies under closer to physiological conditions. The lower power optical studies can be used to argue that photobiology is not strongly affected by the crystal packing forces but not then extrapolate to high power. (Given the initial motions are localized in the protein, this point was known before hand.) In terms of going to higher peak powers, it is well known that at peak powers above 100 GW/cm² that multiphoton ionization and other artifacts begin to contribute for bR. (High power optics for lasers are usually specified for 10 GW/cm² to put this issue into perspective.) If so, any inferred motions are likely dominated by ionization effects as these motions involved are much larger amplitude and would correspond to the main motions observed within their resolution. For this reason, the authors need to compare the transient excited state spectra using a global fit (decay associated spectra). This requirement is now standard practice. It will show if the excited states are indeed the same under the much higher excitation conditions used in the XFEL experiments. If there is a pronounced red shift for example in solution phase studies under the same peak power (as a necessary comparison), it means there is even nonresonant ionization of waters to create solvated electrons and given the relative small absorption cross section would indicate a problem. This process would be occurring in addition to fully resonant ionization of the retinal chromophore in Br. If the decay associated spectra agree at the two peak power regimes then there is sufficient proof that the structural changes can be related to biologically relevant processes. I would further like to stress this condition is for the short time dynamics (0-10 ps range of most interest in this work). Germinate recombination will occur on longer time scales to excited state potentials and will be complicated to sort out.

If there is a problem, here it should be noted that the fraction of systems undergoing ionization would dominate the statistics for the short time scales in recovering the structural changes relative to the very small changes involving the retinal chromophore (a point again relevant to the fraction excited issue on the extracted structural changes). The structural changes in the case of ionizing the retinal or even protein residues would be expected to involve larger changes of the parent and solvation of a full charge (parent ion + e). These motions would be larger than the localized motions of the twisted I, J, K intermediates. The authors need to redo their control studies under the exact same laser excitation conditions and do a detailed comparison of the excited state spectra to show they are measuring biologically relevant processes under the excitation conditions used. This comparison should be made for both solution phase and similar crystal size distributions (re: light adapted bR problem).

Simply put, the excitation conditions used in these experiments are not acceptable in femtosecond spectroscopy. There was a long history early on in the development of picosecond lasers that involved problems with excessive peak power and multi-photon absorption artifacts. It is a normal requirement to show that you are in the linear domain with respect to absorption and independence of spectral changes to ensure one is observing the excite state processes of interest. The same standards need to be applied to femtosecond time resolved x-ray diffraction experiments to avoid these pitfalls.

This work is truly exceptional. I will fully support publication in Nature Communications if the peak power and fraction excited issues can be resolved.

Reviewers' comments:

Reviewer #1 (Remarks to the Author):

◆ *First, although this review poses many questions, most general, some specific, I do not have definitive answers to them myself. We are all exploring new territory; but it's useful to realize that at this stage, conclusions must remain tentative. As the authors state, "This double bond isomerization [of the retinal chromophore in rhodopsins and bacteriorhodopsin] is one of the most studied processes in photobiology." The interest is even more general. The explicit structural details of how isomerization is accomplished are not well understood in chemical terms, particularly in a complicated molecule with many double bonds. And in the present case, the details - or even the fundamentals - differ between free retinal and retinal incorporated in bacteriorhodopsin; retinal - protein interactions are evidently key. So, the wide significance of and interest in this study is in no doubt whatsoever.*

We agree with the referee.

◆ *The problem lies in the severe experimental and data-analytical difficulties of examining the structural basis of this reaction on tiny crystals, illuminated/pumped by a high power visible laser and probed by an exceptionally high power hard X-ray laser. The authors are well aware of (at least most) of these difficulties, describe heroic attempts to overcome them, and present a rather detailed atomic model for the time course of isomerization on the time scale from 240 fs to 10 ps. This time scale, unlike an earlier study by Nango et al., covers the critical phase of isomerization itself. Unfortunately, I'm not convinced by their detailed structural interpretation. The most I will concede is that their structural model MAY be consistent with their X-ray observations, though here too there are difficulties I note below. No model based on dynamic experiments can yield a unique mechanism and hence a unique set of intermediate spectra or structures. That's only possible if all competing mechanisms are considered and all but one shown to be incompatible with the data.*

We fully agree with the referee.

◆ *Very rarely attempted let alone accomplished. Discriminating among competing mechanisms then requires external data e.g. chemical plausibility or (as here) calculations; or simply Occam's Razor. The last normally leads to the simplest linear mechanism, A -> B -> C..... That's fine if the system is pretty much near equilibrium throughout; but highly dubious when the reaction is initiated - as in all photoreactions - by absorbing an energetic photon which drives it far from equilibrium.*

Is this a counsel of despair? No; but it does mean that all conclusions have to be cautiously phrased, and not leave the impression that they are the definitive truth.

Again, we fully agree with the referee. This is one of the reasons we were careful how we phrased the interpretation of the structural data and why we performed time-resolved spectroscopy both in solution and in crystals.

◆ *The authors seem to realize this when they state (in specific reference to their SVD results (main text p. and Supplementary Results p.6, but applicable to the interpretation more generally) that "...these contributions by themselves should on no account be interpreted as separate species, modes or events. These results do establish that a major change occurs in the structure at around 0.5 ps". I readily agree with these two sentences. In fact, the second is the most conservative statement of their results.*

Substantial experimental problems:

Massive averaging of intensities/amplitudes necessitated by weak scattering compromised here by anisomorphism of the crystals: different cell dimensions. Was a given crystal batch always of one form or the other? Or, did a run on one batch display both crystal forms, so that individual diffraction patterns had to be assigned to one form or the other? If this assignment is in error, what are the consequences?

We describe in the Supplementary Methods section "Anisomorphism" that we had different crystal batches with different crystal forms that differ in unit cell lengths. Fortunately, each batch contained only one crystal form/ set of unit cell constants. This makes it very easy to assign the two crystal forms and to pair the correct dark /light data sets. The referee may have missed this. This non-isomorphism is well known for bR as we mentioned in the manuscript.

◆ *Necessity to maximize light-driven signal by an unusual illumination protocol. Yield is increased, but leaves this reader uneasy that artefacts might have been introduced, perhaps by slow, unsuspected structural changes in the seconds range. What evidence these do not exist?*

Light adaption by extended low light illumination is a standard protocol for bR. If this introduces artefacts, they plague the whole field. Importantly, since we are looking at differences between the light-adapted dark state and the photo-excited light-adapted dark state any "slow unsuspected structural changes in the seconds range" would cancel out. We were careful to allow for sufficient time for the photo-excited molecules to return to the ground state of the light-adapted state (i.e. no M, N, O intermediates present).

◆ *Possibility - even likelihood - of radiation damage from the high power pump visible laser and/or the probe X-ray laser. Presence could not be investigated e.g. by a power titration. How might conclusions be affected?*

Multiphoton excitation by the high power of the pump laser is indeed a potential problem as we have pointed out already in the first version of the manuscript. Based on the comments of referee 3 we revisited our power density calculation and found a mistake (mix-up of radius/diameter of the laser beam as well as circular vs Gaussian beam profile) that resulted in significantly higher power density than intended for our experiment. It is thus rather likely that we observe multi-photon effects. This is now addressed explicitly in the manuscript and in the detailed response to referee 3 (page 27). Importantly, however, the combination of the ultrafast spectroscopy and our quantum chemical calculations strongly indicate that the multiphoton effects do not contribute to the crystallographically observed structural features.

In a nut shell, we performed “power titration” experiments with mid-IR and UV/VIS transient absorption spectroscopy (TAS), carried out on both purple membrane (PM) suspensions and on the microcrystals for peak intensities up to 1/3 of the ones used for the TR-SFX experiments. They reveal a new sequential two-photon process, which leads to the excitation of a Trp residue, most likely Trp86. This is in particular evident from the transient mid-IR spectra, showing the bleach of Trp vibrations. This Trp electronically relaxes within ~20ps and is at the origin of induced absorption bands in the UV/VIS TAS spectra, rationalizing the observed red-shift in the 600-700 nm region, previously reported by other authors. Most importantly, the relaxation of Trp does NOT lead to re-excitation of retinal and NOT to additional 13-cis isomers, as unambiguously shown by the vibrational bands of this species in the mid-IR. In conclusion, our TAS and QM data show that single and multiphoton events are temporarily and spatially decoupled. Thus, the sub-ps structural data reflect only features related to single-photon absorption processes.

It is highly unlikely that the X-ray laser causes significant radiation damage (see e.g. Chapman HN, Caleman C, Timneanu N. 2014 Diffraction before destruction. Phil. Trans. R. Soc. B 369: 20130313. <http://dx.doi.org/10.1098/rstb.2013.0313>)

◆ *Certainty of large variation from crystal to crystal of the extent of isomerization arising from range of crystal sizes, orientation in the laser beam, and position in the sample stream. These tend to mask the inevitably small time-dependent changes in structure amplitudes in the fs to ps time range. Consequences?*

This is an inherent, unavoidable feature of these kind of diffraction experiments. The range of crystal sizes and the apparent difference in thickness caused by different orientations (perpendicular to a thin wide plate-like crystal versus along the wide side) will likely result in a different extent of photoexcitation. Different orientation may display different photo-selection rules, which is why we have used circularly polarized laser

light. Different positions in the stream only matter for very short pump probe delays, the difference in distance travelled by the pump laser light does not matter in our case. If a crystal is at the back end of the LCP stream and other crystals are in front of it, it will not be photoexcited efficiently. However, the probability for this to have occurred is very low in our experiments since we have observed very few double hits (multiple diffraction patterns in one image). All of these effects may result in a lower apparent photoexcitation yield than expected based on calculation. This would not affect our interpretation.

◆ *Substantial problems in data analysis:*

Several novel difficulties arise in refining a light-driven structure in the fs to ps time range which certainly affect and may compromise the final interpretation.

As the authors note, time-dependent difference maps can be hard to interpret if the structural differences are complex, as here. Much then depends on appropriately converting these to more readily interpretable derivative maps, based on extrapolated Fourier coefficients. The authors do this carefully, explore many parameters and approaches, and describe this in detail. Their maps seem plausible and exhibit time-dependent changes to the eye. To me, it's unexpected that they would be so prominent.

Why? The difference in photoactive yellow protein (PYP) described by Pande *et al* are of similar magnitude if not larger. Moreover, during the revision of our manuscript a related study was published in Science (Nogly *et al.*, online version: DOI: 10.1126/science.aat0094) that shows even larger differences.

It is well established that difference electron density maps are exquisitely sensitive to small changes in a structure, as long as the phases are good enough.

◆ *In this type of time-resolved experiment that (probably? involves incoherent evolution of structures, a time-dependent mixture of structural states is present at all time points.*

This depends on what the referee calls coherent/incoherent. In the context of ultrafast spectroscopy experiments referred to in our manuscript, a coherent oscillation implies a phase relationship between the pump laser and the observed spectroscopic response. This is possible in the case of the well-controlled spectroscopic experiments but not in case of SFX experiment where the crystalline sample changes position, orientation etc from shot to shot.

The ultrafast structural changes occur “in step” within the time- and spatial resolution of our experiment. Dephasing due to coupling to the thermal bath occurs later. Indeed, spectroscopic studies show that the dephasing occurs on the picosecond time scale.

◆ *A priori the number of states, the mechanism they comprise, their individual structures and the time course of their evolution are all unknown. In this reaction there is a wealth of ultrafast spectroscopic data in solution which the authors show is quantitatively closely similar to data on their crystals. These data have been interpreted as arising from a linear mechanism comprising spectroscopically distinct states I, J, K, L..., which at least provides a starting framework for interpreting their time-resolved structural data.*

Small correction: the mechanism is not linear due to the back-reaction (see Figure 1b in the manuscript).

◆ *However, one spectroscopic state could encompass many different structural states; a single electronic state of the retinal could be coupled to many protein structures/states.*

We agree.

◆ *Conversely, electronically distinct spectroscopic states could have such small structural differences between them that they're crystallographically indistinguishable. How to decide what the mechanism and states are in fact in the crystallographic data? Might deviations from the spectroscopically-derived model be evident in their data? How to decide how many states are present in any time point?*

These are important questions that one needs to ask oneself when interpreting electron densities, in particular in case of time-resolved experiments. We used the following approach: We checked whether the protein is active in the crystalline state and established the time-constants of the reaction by spectroscopy on the crystals. Since they are very similar to the ones obtained by PM suspension experiments, we felt confident to include published information on kinetics (including the expected number of states) obtained by various spectroscopies. We fitted as many structures as permitted by the quality of the electron density (which turned out to be one) - irrespective of knowing that there should be more than one species. Specifically, we fitted apparent conformations of the isomerizing retinal, despite the fact that at any given time delay < 3ps the electron density corresponds to (at least) a species associated with the forward reaction and one that corresponds to the backward reaction. Using a simplified partitioning scheme we calculated the isomerization time constant and C13-C14 torsion angle. We do see deviations of the spectroscopically known model. Spectroscopy has observed oscillations associated with the retinal only, in addition to these we observe oscillations of the conformation of protein residues as well as water molecules.

◆ *The authors chose to refine a single structure into each extrapolated map, in which that single structure is in some sense an average over the unknown, individual structures actually present. How would breakdown of the implicit assumption that a single structure can represent a structurally heterogeneous mixture affect the structural conclusions on the mechanism of isomerization?*

We are fully aware of this problem and are careful to describe the models as apparent conformations. We decided to use a conservative approach for the interpretation of the electron densities as opposed to fitting structures based on prior spectroscopic knowledge that are not demanded by the electron density itself. Given the prior comments of the referee concerning over-interpretation, we suspect that he/she agrees with this approach.

In this context, it is interesting to note that the manuscript by Nogly et al that appeared in Science after we obtained the referee comments also fits the data with a single structure despite the fact that Nogly et al has higher resolution data. This would imply that the structural changes between the two species are rather similar.

◆ *After absorbing a visible photon, the retinal is highly excited - "hot", with a vibrational temperature equivalent to many hundreds of deg K. Bond lengths and torsion angles may change from that effect alone, even with no isomerization. Atoms are "smeared out"; and there are no standard bond lengths and angles available to conduct crystallographic refinement in real space. Yes, the system rapidly cools - but how fast? What "standard" bond lengths and angles were used (if any)? What are the effects on refinement of this temperature jump and the large deviation from equilibrium?*

The atoms are not necessarily "smeared out". In case of very high spatial resolution the bond length and angles can be obtained from the electron density. This is not the case for our data. We tried to loosen the restraints for the carbon—carbon bond lengths of the retinal chain. This works well for the light-adapted dark state structure, double bonds refine to shorter distances, single bonds to longer distances. We had hoped that we would be able to resolve the bond-softening taking place during the very short time-delays but our time-resolved SFX data were not good enough to allow this. Therefore we used standard bond length and bond angle restraints (or rather, prior probabilities...) but released the planarity and torsion angle restraints for the apparent conformation of the retinal (Δt 0.24 ps \leq 0.74 ps), allowing the torsion angles to change in order for the retinal to fit the electron density. We also used bond lengths and angles derived by quantum chemistry for some of the structures corresponding to longer pump delays as indicated in the text (in Supplementary Methods "Retinal geometry" and in the Supplementary Note 2.).

◆ *With all these effects that conspire against confident detection of small structural changes,...*

The referee was right if these effects were to introduce random uncorrelated movements. This would indeed wash out any correlated response of the system.

◆ *I'm not convinced that the oscillations are real; few cycles, poor signal-to-noise.*

We think the oscillations are real (see also below), corresponding to correlated movements of the system. The structures were refined independently by two different co-authors, one in Heidelberg, one in Grenoble, using different refinement protocols and analysis tools. We have also performed a jackknife-type method to determine the error in coordinates, which is smaller than the observed changes (see “Error estimates” in Supplementary Methods). The referee may be aware of our previous observation of oscillations in photodissociated carbonmonoxy myoglobin (Barends et al Science 2015). We agree that the signal-to-noise ratio is not very high, yet our error estimates give us confidence in our observation of the oscillations.

The manuscript by Nogly et al that appeared in Science after we obtained the referee comments shows oscillations in the intensity of difference electron density peaks (light minus dark) as a function of time. The oscillations were not further described, fitted or functionally interpreted. While it is difficult to compare changes in e.g. distances between pairs of refined coordinates with the intensities of the difference electron densities between two different time points, it is interesting to note that they have a similar period of about 350-400 fs, confirming and supporting our findings.

◆ *Shouldn't the also be evident in one or more of the right singular vectors in the SVD? Particularly if SVD were applied only to the volume including the chromophore and the immediately surrounding protein? (I didn't see where the volume actually used was given.)*

Indeed they are. We added the information as Supplementary Table 2.

◆ *Finally, isomerization has been studied by closely similar pump-probe FEL techniques in single crystals of photoactive yellow protein (ref 19). Their results do not display oscillations, but agree in showing a major structural transition in the chromophore at around 500 fs which the authors attribute to isomerization. It's plausible that the earliest steps of isomerization are identical - or at least closely similar - in all systems. Worth adding a sentence?*

We thought about adding a paragraph about PYP but concluded that it would be beyond the scope of the current manuscript. The observation of a structural transition /

photoisomerization around 500 fs for bR and PYP is most likely a coincidence. Visual rhodopsin photoisomerizes significantly faster.

=====

Reviewer #2 (Remarks to the Author):

◆ *The authors report a time-resolved x-ray diffraction experiment at a free electron laser of bacteriorhodopsin (bR), in which they recorded diffraction pattern from ~250 fs up to 10 ps (and 11 ms) after photoexcitation with a 150-fs laser pulse. Structural refinement against extrapolated difference densities enabled them to visualize the time-series of structural changes affecting both the retinal co-factor and the nearby protein residues. Interestingly, the authors could resolve (coherent?) vibrations of modes involving atoms in and around the retinal, and linked these to oscillations seen in various spectroscopy signals. The refined structures suggest a mechanism of isomerization that is very much in line with previous computational predictions, but not quite the same. Since this is the first time the sub-ps regime of the photoactivation in bR is addressed with tr-sfx, I think the results will of very high interest to the community and beyond. I therefore would very like to see this study appear in Nature Communication, but not before a number of (small) concerns, in particular about the computational aspects, have been addressed.*

◆ *Although the paper is all about visualizing the first event in the bR photocycle, there is no image showing the complete isomerization process. Instead, the time evolution of the retinal conformation has to be inferred from the text and from combining figures 2 and 3 that show the retinal pocket from different angles. I suggest therefore to include a figure showing the complete time-evolution of the isomerizing retinal (corrected for the 2:1 branching ratio) and environment from $t=0$ to $t=10$ ps. Also in the same figure I would like to see the (corrected) torsion angles in the retinal, again from 0 to 10 ps. Currently, supporting information Figure 8 only shows information on the sub-ps timescale, making it difficult to see when (or how) the actual isomerization occurs.*

We have now included a table with the apparent and corrected torsion angle values and a plot of the exponential function fitted to the data to show the time evolution (Supplementary Figure 2). We would like to point out that the isomerization proceeding through the conical intersection is essentially instantaneous and cannot be visualized by crystallography. The corrected torsion angles correspond to the average of the population (without the contribution of the back reaction), but not to actual structures accumulating at a specific time point. Therefore, the requested figure would not correspond to actual events. While we can extrapolate the C13=C14 torsion angle, we cannot extrapolate the other torsion angles. We therefore feel that a figure showing structures containing a mixture of corrected and apparent torsion angles is not helpful.

Therefore, it is unfortunately not possible to have a figure similar to e.g. Fig. 4 in Altoe et al, PNAS 2010.

◆ *The authors have tried multiple approaches to refine intermediate structures against the experimental data. This makes the paper (methods and SI) a bit difficult to read, but at the same time increases confidence. However, perhaps due to the many details, some aspects were lost on me. For example, the authors write that for the refinement of the 10 ps intermediate, they used geometric parameters from the QM calculations (page 6). However, in the description of the QM methodology, they write that "The starting coordinates of the non-hydrogen atoms were taken from the refined SFX model" (page 16). If the QM parameters could only be obtained after the refinement, while the refinement relied on the QM parameters, this must have been an iterative process? If so, can the authors rule out a potential QM bias in these refinements? I think some additional details on how the QM results were used for the refinements and vice versa is needed here. In particular, because the QM and refined models seem not too similar, with a difference in the C11=C13 torsion angles of 20 degrees (refined SFX: -61, QM: -41).*

The referee probably refers to the C13=C14 torsion angle. We agree with the referee that this was not described very clearly (refined SFX gave -68°). We have now expanded this part (main manuscript: page 5, "Changes in the retinal" and page 24). In the 10 ps data the electron density around the C13=C14 region was weak and difficult to fit. We therefore fitted the rest of the electron density and geometry optimized this model by QM. The C13=C14 torsion angle in the starting unrefined SFX structure was -87°. The optimization was performed in the delocalized intrinsic coordinates, and therefore, none of the atomic Cartesian coordinates was explicitly frozen. Constraints were applied to the distances (in total 69 distances) between selected terminal atoms of the fragments comprising the cluster (in total 25 atoms out of 241 atoms). Four of the so contained atoms belong to the RPSB-Lys216 fragment (the C3 and two C atoms of the methyl groups in the ionone ring and the C β atom of Lys216). Notably none of bond lengths, angles, or dihedral angles of RBSB was frozen or explicitly constrained during the optimization. The constraints are weak and did not introduce any undesired bias in the obtained QM geometry of the twisted 13 *cis* retinal. (Similar constrains were used during the optimization of the all-*trans* retinal model.) Subsequently, we used this QM-optimized model for another round of refinement against the X-ray data. This resulted in a relaxation of the C13=C14 torsion angles, resulting in the discrepancy mentioned above by the referee. We do not know whether this difference is real or related to the quality of the electron density map in this region.

◆ *Furthermore, to show that there is indeed a good agreement between the QM-optimized 10-ps structure and the refined 10-ps SFX structure, I suggest to include a*

new figure, in which the two structures are superposed, perhaps in Supporting Information. This will make it a lot easier for the reader to see how well the two models agree. In the current version no figure(s) of the QM optimized 10-ps structure is included anywhere, unless I've missed that.

We have now included an additional figure showing the QM-optimized 10-ps retinal overlaid with the 10-ps SFX structure (Figure 2 e, f). We added the model obtained by Nogly et al for the 10 ps time-delay as well.

◆ *As in previous SFX studies, a very high pump laser power was used. Although my concerns are relieved by the agreement between the 33 ms refined structure and a previously published structure of late-M intermediate (Fig. S1), it might be informative to compare also the difference density maps at 33 ms with the map of that late-M intermediate.*

We agree that this is informative. We calculated the difference electron density between our 33 ms time-delay data and the published 1.725 ms data (Nango et al) and see no significant difference density peaks. We added this comparison as well as the one with the 8.3 ms time-delay data published by Nogly et al to the Supplement (Fig. 13). In this case, our density contains more M-like structural features.

◆ *The authors have also performed a series of quantum chemistry calculations to address what happens prior to the first measured time points (< ~250 fs). Unfortunately, I have not completely understood the role of these calculations in the present study. What time point do these computations represent, and are the structural changes seen in the optimized excited states comparable to changes observed in the refined structures?*

Our quantum-chemistry calculations address the electronic structure of the bR active site, in particular we concentrate on the electronic coupling between the RPSB and protein. Such a coupling, mediated by the electron-donor residues interacting with the retinal, is formally described by the presence of low-energy charge-transfer (CT) states strongly coupled to the S₀ and retinal S₁ states in the excitation spectrum. Along with the optically excited S₁ state, the CT states govern the initial excited state dynamics on bR. These calculations play the same role as the calculations addressing the S₀-S₁ transition of the RPSB that are widely cited and commonly accepted as useful and insightful. Our calculations, however, extend the computational approach to the problem of bR photoactivation, i.e. of the full active site including neighbouring residues and waters in the quantum-mechanical description. The obtained energy estimates (both state energies and electronic-state couplings) suggest that the CT states along with the S₁ state are involved in or govern initial excited-state evolution (on the femtosecond timescale) of bR. This time-window was addressed by our time-resolved structural study.

◆ *The calculations suggest that rather than photo-isomerization, electron transfer from nearby Trp86 and Tyr185 residues into the retinal occurs instead.*

The calculations indicate that ultrafast photoinduced electron transfer is highly likely to occur: the calculations did not suggest that electron transfer occurs rather than or instead of isomerization. We did not address photoisomerization as such by our current calculations, but focus on different electronic relaxation paths prior to the isomerization event. This path, as our calculations suggest, is defined by coherent interconversion of the retinal S1 and CT states. As our structural results indicate that changes of the protein occur prior to RPSB isomerization – in line with ultrafast spectroscopy – we focused our QM study on electronic transitions between the chromophore and protein that may explain the observed ultrafast long-range structural changes.

◆ *Although the involvement of charge transfer (CT) states in the photo-chemistry of bR had been suggested before (see e.g. Hayashi et al, Biophysics 8 (2012) 115), there seems no support from experiment (yet).*

We thank the reviewer for pointing out the work by Hayashi et al. This study presented computational evidence for a role of aromatic residues close to RPSB in the active site in modulating the opsin shift (the S0-S1 energy in bR). The finding of this publication is closely related to our computational results, although this study did not introduce the CT electronic states as part of the bR excitation spectrum. The paper “only” discussed the CT interactions between Tyr185 and RPSB in the S0 and S1 states of bR. Nonetheless, enhanced charge transfer from Tyr185 to RPSB in the S1 state is an indication of electronic coupling between the of the S1 and CT(Y185) states explicitly characterized by our calculations.

We performed ultrafast spectroscopy in the visible and infrared spectral region to explore the involvement of CT states. The infrared data taken in the region of the retinal C=C stretching vibration demonstrate retinal isomerization accompanied with bleaching of the Trp band at 1554 cm^{-1} . Moreover, the visible data show an excited state signal around 463 nm (Fig. 6d) decaying on the time-scale of ~ 20 ps, assigned to Trp excited state absorption. Our computational study links the appearance of these spectral signatures to the CT(W86) state manifold (Fig. 8), and its strong coupling to the retinal and Trp86 excited states. This is the first study combining quantum chemical calculations, with visible, vibrational, and crystallographically derived structural dynamics to substantiate the involvement of charge transfer states in the photo-chemistry of bR.

◆ *Instead, previous experiments, including the present one, as well as computational studies, all suggest that retinal photo-isomerizes in the S₁ excited state. Although in most earlier computational studies only retinal was included in the QM region (and thus excluding the CT states), isomerization timescales that are very much in line with time-resolved spectroscopies were observed.*

This is a misunderstanding. We agree that photoisomerization occurs once retinal is in the S₁ excited state and the role of CT states for the isomerization remains to be examined. However, previous computational studies that did not consider the presence of the low-energy CT states in bR do not provide a complete picture of the bR early light response. We note that it is technically easier to consider only excited states of RPSB and ignore coupling of the RPSB and protein. Excluding the CT states from computational studies does not provide us with evidence that the CT states are of no relevance. Indeed, as clearly demonstrated by refs 38 and 39, the charge-transfer interactions with electron donors Trp86 and Tyr185 significantly affect properties and energies of S₀ and S₁ states, therefore, the CT interactions must affect photoisomerization. We stress again, the charge-transfer interactions in the S₀ and S₁ states and the CT excited-state manifold characterize the same phenomenon. The computed isomerization timescale matching the experimental ones unfortunately cannot be used as a criterion if one seriously takes into account the approximations involved in the computations.

◆ *Population of CT states, therefore, seems not necessary for an efficient isomerization and intuitively, I'd even expect these CT states to reduce rather than enhance the isomerization.*

Currently, this cannot be concluded. The role of CT states in bR function needs to be carefully investigated before conclusions can be made. As we point out above, the role of the charge-transfer interactions in the S₀ and S₁ states has been previously demonstrated by quantum-mechanical calculations of extended active-site models. The S₀/S₁ conical intersection mediating isomerization should be also affected by the charge transfer interactions. Thus, it is rather likely, that the charge-transfer interactions play a role in efficient isomerization. The charge-transfer interactions in the S₀ and S₁ states and the CT excited-state manifold characterize the same phenomenon.

◆ *Unfortunately, the authors do not discuss by what mechanism the CT states contribute to the photo-activation. In particular, it would be important to discuss (or speculate at least) how population of CT state would lead to the 13-cis retinal configuration observed at 10 ps.*

We expanded our calculations and rewrote the description of quantum chemical results. We now suggest how the CT states contribute to the excited-state dynamics that initialize the isomerization (see next point). A more comprehensive study of the CT states contributions is currently beyond the scope of this manuscript, but would be indeed highly interesting, as the high sequence and structure conservation of the electron-donor residues in rhodopsins indicates a possible role of these residues in the biological function of these proteins.

◆ *Along the same lines, the authors state that the coherent oscillations are due to crossings between the S₁ and the CT states (page 9 manuscript, and page 12, SI). I, however, don't see the connection. How would spreading of population over multiple excited states lead to coherent nuclear vibrations? Please explain.*

We discussed not just crossings of the states, but their interconversion - S₁→CT, CT→S₁. This interconversion is supported by the large electronic coupling at the S₁- and CT-relaxed geometries. The magnitude of the coupling suggests a fs process that is faster than isomerization. On the other hand, structural relaxation in the CT states leaves the energy gap with the ground state rather large, indicating that excited state decay via CT relaxation does not occur and that isomerization at the S₁/S₀ state crossing is the dominating excited-state decay pathway. In such a case, state interconversion of the S₁ and CT states should result in coherent nuclear motions preceding isomerization.

◆ *Also, unless I got it wrong, the authors speculate that the CT states are involved in the proton translocation process later in the photocycle (page 10). How? Are the authors suggesting that these CT states are that long-lived to affect the dynamics in the L state?*

This is a misunderstanding. We imply that the CT state influences the hydrogen-bonding network, including the water molecules, which may set the scene for later events (e.g. proton transfer).

◆ *The authors observe that in the CT states the excited state dipole moment is much larger, and refer to Science 309 (2005) 917 and PNAS 106 (2009) 7718 (refs 8 and 38) to support this finding. However, I have understood that in those works, the changes in Trp absorption were used to probe the charge migration in the retinal, while here, it is claimed that the Trp donates an electron and is thus actively involved in the photo-activation. Would the formation of the CT state lead to similar changes in absorption signals of TRP as seen in those previous works? If so, this could be support for the involvement of the CT states. I suggest therefore to compute the absorption of the CT state and see if it fits the previous observations.*

As requested by the referee, we computed the absorption of the S1 and CT(W86) states. The results are summarized in Fig.8 of the manuscript. The results indeed, support involvement of the CT states in the early dynamics of bR as well as in the multiphoton excitation process

◆ *Can the authors rule out that the charge transfer (CT) states are only accessible due to a rather limited active space, which includes exactly those orbitals required to get the CT states? To rule out potential artifacts, can the authors include more states (and orbitals) and then compare the energies, as was done for instance in JCTC 8 (2012) 1912 for GFP chromophore pocket?*

We performed extensive calculations testing the effect of the active space size on the CT energies. The results presented in Supplementary Note 2 clearly demonstrate that the CT energies obtained with the extended active spaces are consistent with the energies obtained with the small active space. Thus, we exclude that the predicted ultrafast photoinduced electron transfer is an artifact of our small active space.

◆ *I think the original GMH approach was first described in Chemical Physics Letters 249 (1996) 15-19, rather than in ref. 60, which is sort of a review of multiple approaches to compute couplings.*

We thank the referee for bringing this paper to our attention; we added it to the manuscript.

Reviewer #3 (Remarks to the Author):

◆ *The manuscript entitled: “Three-dimensional view of ultrafast dynamics in photoexcited bacteriorhodopsin (bR)” by Kovacs represents an impressive and highly innovative approach to femtosecond time resolved crystallography. The structural correlation to known excited state dynamics are similar in time scales, however, there are some notable exceptions. The authors have advanced theoretical studies that seem to indicate that charge transfer in the excited state involves the protein residues and this can explain some of the discrepancies with the current picture for the excited state dynamics of bR. This new picture could have relevance to other photoactive rhodopsin systems. There are however a number of technical issues that seriously question the conclusions of this work. It needs to be stated that this is a real tour de force for the field. I support publication in Nature Communications but there needs to be some reconsideration of the main conclusions especially in light of a few points that limit the ability to draw the conclusions made.*

We are happy to see that the referee appreciates our work. We are very grateful to the

referee for his/her very careful analysis of our manuscript that resulted in the identification of some serious errors. We have corrected the mistakes in the manuscript and adjusted the interpretation related to this. In addition, we significantly extended our ultra-fast spectroscopic investigations and QM analysis to address in particular also multiphoton effects.

In the initially submitted version of our manuscript, we made a mistake in the description of the geometry of the pump laser setup. In order to reduce the distance that the segment of the jet that was illuminated by the pump needs to clear so that the next laser pump beam excites only material that has not been accidentally illuminated by the previous pump, we displaced the pump laser $\sim 25 \pm 5 \mu\text{m}$ below the X-ray interaction region. We did not displace it further because we were worried about too large changes in pump energy in the case of drifts in the laser position. The other mistake we made was in the calculation of the laser power since we mixed up radius and diameter and did not take into account the Gaussian beam profile.

The peak power of the laser is $\sim 1 \text{ TW}/\text{cm}^2$ (as pointed out by the referee). Taking into account the laser offset of $\sim 25 \mu\text{m}$, and the experimentally determined loss by reflection and scattering (20 %), the laser power is $\sim 500 \text{ GW}/\text{cm}^2$ in the X-ray interaction region (Supplementary Table 6). As now described in detail in the supplement, we calculated the average thickness of the plate-shaped hexagonal crystals, taking into account the information about their orientation in the X-ray beam which was derived from the orientation matrix of the indexed diffraction patterns. The average thickness is $\sim 5.9 \mu\text{m}$. Taking into account the experimentally determined loss of photons due to scattering/reflection (ca 20 %, see Supplementary Note 1) this results in an average of approx. 21 photons/retinal. Obviously, some retinal chromophores will absorb several photons, others will absorb only a single photon. The “power titration” experiments carried out for the re-submission (see page 27 below) indicate that the crystallographically observed sub-ps structural changes are related to single photon absorption.

In the following, we gather the different referee comments into topical sections for an easier reading.

❖ **Oscillations:**

1. The oscillations seen in the data involving assignments to structural induced changes that are biologically relevant needs to be challenged. First, there needs to be a direct comparison to the actual frequencies that are known to the Franck-Condon active modes that involved in the initial optical excitation and are anharmonically coupled along the motion to the conical intersection (CI) to the coupling (C13=C14 bond elongation) and tuning modes (low frequency torsion) directing the key photoisomerization step . The authors should compare what reactive modes have been observed using optical, Raman, and impulsive Raman detection to those they observe

and the assignments. (Note the reference in relation to coherent motions by Matthias et al is incorrect as this was for rhodopsin.)

We realize that the Wang *et al* reference refers to rhodopsin. We cite this work in the introduction in the general context recalling the hypothesis that coherent vibrational dynamics may be important for directing isomerization. We do not cite this work in the specific comparison of modes.

◆ *Please note that the key modes that have been identified so far for bR are the HOOP mode at around 860 cm⁻¹, the low frequency skeletal torsion at 200-230 cm⁻¹, and higher frequency modes involving C=C stretch around 1600 cm⁻¹. None of these are consistent with the motions observed.*

The referee is right that there needs to be a comparison between spectroscopically observed modes and the ones seen in our data. We had done this, but apparently not with enough clarity. Given the relatively long pulse width of 150 fs, high frequency vibrational wave packets as those quoted by the referee cannot be excited. However, low-frequency oscillations as were reported in femtosecond pump-probe, modulating e.g. the stimulated emission, or four-wave mixing experiments are most relevant data to compare. In (Hou et al. *Chem Phys Lett* (2003)), report for bR excited state oscillations observable during the first picosecond with dominant frequencies of 100 and 170 cm⁻¹. That these frequencies are protein-specific is highlighted in the same paper. Other references like (Kahan et al., *JACS* (2007), Shim et al., *JACS* (2009), Liebel et al., *Phys Rev Lett* (2014)) concentrate on the higher frequency vibrations, even though Liebel et al., *Phys Rev Lett* (2014) confirms the 170 cm⁻¹ vibration, since the FFT procedure performed on residuals removing the incoherent multi-exponential dynamics deliberately avoid the low-frequency part (< 150 cm⁻¹) as this is subject to too large errors induced by the subtraction procedure. Kraack et al. (*ChemPhys Chem* **12**, 1851-1859 (2011)) report similar low-frequencies for excited state vibrations, but limit their discussion, owing to the above technical reasons, to frequencies ≥ 160 cm⁻¹.

Clearly, our fs X-ray data identify non-monotonous forward and backward motion of the torsion angle between the β -ionone ring and the polyene chain (C18-C5=C6-C7) and of the torsion angle between terminal methyl groups (C18 and C20; torsion angle C18-C5-C13-C20). These are approximated by sinusoidal functions with ~ 120 cm⁻¹ and ~ 90 cm⁻¹ frequencies, respectively, (Fig. 3), which bear obviously large error bars due to data scattering and the limited number of oscillations (1.5 periods).

In conclusion, the 90-120 cm⁻¹ oscillations we report for motion of the PSBR are consistent with the data of (Hou et al. *Chem Phys Lett* (2003), Wand et al., *JACS* (2011), Liebel et al., *Phys Rev Lett* (2014)) for bR, and with similar values reported for Anabaena Sensory rhodopsin in (Wand et al., *J Phys Chem* (2013), Wand et al., *JACS* (2011)). However, due to the above technical reasons they were only marginally identified and received less attention in ultrafast optical experiments, even though they are expected to modulate the differential absorption and polarizability of the PSBR in a similar fashion as the high frequency modes. This is less obvious for the oscillatory

motion we report for W402 and nearby amino acids, and will be discussed together with the mechanism that launches them below.

2. The very fact that the authors observe oscillations in their diffraction data that involve nearby residues points to artifacts. The motion to the CI and associated change in the time evolving potential is too slow to impulsively excite the specific vibrations observed (600 fs is not sufficiently short driving force for such motion – see further below).

Yes, we do not claim that isomerization drives the oscillations, as they precede it. The opposite is likely to be true, in line with our QM calculations.

◆ *The observation of coherent oscillations in optically observed spectra are due to the short pulse excitation of vibrational wavepackets on the excited state surface (displacive excitation) or ground state Raman processes (spectator modes.) To observe surrounding protein residues oscillate is in contradiction to other measurements as there is no transition dipole moment coupled to these motions.*

We do not agree with the statement that transition dipole moments activating motions of the surrounding residues do not exist. We report such transition dipole moments in Supplementary Table 3 (e.g. tdm S1-CT(W86) or tdm S1-CT(Y185)). The substantial electronic couplings mentioned in the QM result subsection and presented in Supplementary Fig. 12c are also derived from these tdms. Our QM calculations indicate that the coherent oscillations in early bR photodynamic could be due to the so-called electronic coherences (see for instance a theoretical model by Egorova and Domcke in J. Photochem., Photobiol. A (2004) the coherent dynamics of the S1 and CT states is presented in their Fig. 2 as the dashed population probability).

The multiphoton excitations of the protein (which the reviewer prefers to our proposed “single photon” excitation) strictly speaking, depend on similar tdms, but the density of such states mediating the chromophore/protein coupling increases at high energies facilitating de-coherence. As we found (by QM computations of the energies and tdms) the CT states can be populated by photoexcitation, we suggested that these states were observed in spectroscopy experiments and may play a role in bR biological function.

◆ *The force would have to come from the reaction itself not some superimposed optical preparation step. The force then must be evolving on a time scale shorter than $\frac{1}{2}$ period of the excited vibration. It is telling that the reported oscillations are effectively the pulse width.*

The time scale of the observed oscillations is rather consistent with their pulse duration and associated photophysical processes. This observation and induced motions of waters, polar residues is consistent with ionization of the retinal chromophore, a well known problem with the peak power conditions used for this experiment.

This point is partly addressed in the reply to the previous point. We address the issue of multiphoton effects in detail (see page 27 of the response).

◆ *I don't see how a 500 fs-600 isomerization process can impulsively excite 100 cm⁻¹ modes involving neighbouring residues in the protein. (The argument may be to emphasize charge transfer contributions but then this should have been seen previously spectroscopically – see below).*

This is a misunderstanding. We do not claim that the isomerization process excites modes of protein residues.

❖ **High excitation power and excitation rate:**

3. Given the above, it is essential the authors go back and thoroughly do a power dependence. They assert they do not have peak power problems with their excitation conditions, as they observe similar dynamics as their pump probe studies. (The authors need to state the excitation conditions for the solution phase and crystal fs studies in figure 1c.) The authors need to redo the all optical pump-probe studies of the crystals under the exact same conditions as the XFEL studies and they must directly compare transient spectra. This means a global fit using decay associated spectra to the data to ensure they are seeing the same excited state dynamics as the biologically relevant processes. The dynamics may be somewhat different but the related decay spectra should be the same.

The red edge of the spectra are most useful in this comparison. There will likely be a red shift due to ionization of the retinal in a fully resonant multi-photon process. The peak power of the excitation is reported to be >250 GW/cm². This peak power is enough for non-resonant multiphoton ionization of water never mind fully resonant multi-photon artifacts. The fact that the x-ray probe spot size was on the order of 1 micron relative to the 100 micron excitation spot size means that the peak power probed by the x-rays is related to the center of the laser excitation not the spatially averaged intensity. The locally probed peak power is likely over 500 GW/cm². There are some beam position overlap issues pointed out but the power of >250GW/cm² at least up to 500 GW/cm² should be studied to be sure there are no artifacts. For completeness sake, the authors should identify the peak power where changes occur (> 500 GW/cm² if need be).

We agree with the referee. The X-ray interaction point was upshifted with respect to the pump laser center by ~25 um, which reduces the power density to about half of the peak power. We are fully aware of the multiphoton issues, which will be addressed below in the section “Request for power titration experiments using TAS”

◆ 4. *On a related issue, the authors report the laser excitation conditions correspond to 1 photon absorbed per retinal chromophore. It is not possible to obtain 100% excitation and this point is again part of the problem.*

We have corrected the number of photons/retinal in view of the updated laser power. On average we likely have 21 photons per retinal chromophore.

The referee is right that it is not possible to obtain 100 % photo excitation and that the analysis of low occupancy data is very difficult.

◆ *There is a units problem in their reported extinction coefficient.*

Thank you for catching this. We corrected the typo.

◆ *I calculate at the excitation conditions used, there are well over 1 photon per retinal (unless the protein concentration is off by an order of magnitude). The number of absorbed photons points out the problem again, without even considering peak power problems.*

To get this correct, one has to include stimulated absorption from the excited state to higher lying electronic states with nearly the same oscillator strength as well as stimulated emission during the pulse duration (with exactly the same oscillator strength with minimal motion along the CI). To get the correct fraction excited is nontrivial and this number is critical to how they inferred the fraction excited and ensuing branching ratio.

The authors need the fraction excited to analyze their data.

This is not correct. We do not need to derive this number (e.g. by spectroscopy) for the analysis of the X-ray data. In fact, we obtained an estimate of the fraction of excited molecules from comparing the data of the initial light-adapted dark state with the data of the pumped state. How we did this is described in detail in the Supplementary Methods. We can only identify populations that differ significantly in structures and therefore structure factors / electron densities. At our resolution, we cannot identify populations that differ in only electronic structure, hydrogen bonding or other small conformational differences.

◆ *They must do at a proper estimate of the true fraction excited at the peak power used and then apply the same logic in branching ratios. The problem will be taking into*

account the excited state absorption. The fraction excited, independent of light adaption issues, is not 20 % as reported.

As described above, this number is consistent with the X-ray data/electron densities. It does not exclude the occurrence of excited state populations that have very similar structures to the light-adapted dark state.

◆ *5. The above problem is important as the structural changes depend critically on the assumed fraction excited.*

This is not correct and likely a misunderstanding of the referee. The fraction excited was derived from the analysis of the diffraction data; it was not derived from other considerations and then used to analyze the diffraction data. The referee is right that the apparent extent of the structural changes does depend on the assumed excited fraction used when calculating the extrapolated structure factors.

❖ ***Evaluating the excitation conditions and possible effects of multi-photon excitation:***

◆ *This reviewer is concerned about multiphoton ionization artifacts that would better explain their data and motions of residues in the protein. The excited state potential is localized on the retinal (see below re: theory). Some discussion of the importance determining the fraction excited is needed as this greatly affects the extrapolated changes in the structure as pointed out by Coppens et al.*

Unfortunately, we do not know to which publication of Philip Coppens the referee refers.

Nevertheless, we agree with the referee. For this reason, we spent a lot of time and care on the best analysis of the fraction excited. We described this in detail in the Supplementary Methods. Maybe the referee had overlooked it. We address multiphoton effects now explicitly in the manuscript and below.

◆ *The crystal thickness varies, as does the fraction excited in the probed crystal volume. This effect and multiphoton absorption will significantly affect the true fraction excited that undergoes the photobiological process of interest. The authors need to examine the shot to shot variation on the fraction excited and how this would affect their inferred structural changes.*

The referee is correct; there are many strongly varying parameters in this experiment. Not only in the fraction excited in the probed crystal volume but also in the data quality and X-ray intensity for each shot, ... In general, serial femtosecond crystallography relies on highly redundant data so that the fluctuations converge to an average value. It

is thus in the nature of the method that the shot-to-shot variations do not matter as much as in an experiment with few measurements. We determine an apparent excitation yield that is consistent with the changes in the electron density compared to the initial state and thus apparent structural changes.

◆ *There is no way around this problem affecting the results. It may be impossible to determine the fraction excited but the effect on the inferred structural changes with error bars for the fraction excited needs to be clearly stated with a more rigorous determination of the fraction excited.*

As the referee indicated, the peak intensity has been corrected in the resubmitted version (500 GW/cm⁻² at the X-ray interaction region). This corresponds to ~35 photons per absorption cross section of bR at the crystal surface (see page 10 of the Supplement). However, depending on the crystal orientation, and owing to the short penetration depth the intensity decays to zero inside the large majority of crystals. We therefore deal with a large distribution of excitation yields, with an average value of ca. 21 photons/retinal in the volume probed by X-rays as detailed in the supplement (Supplementary Table 6).

Again, the average fraction of excited proteins displaying a structurally detectable change, was derived solely from an analysis of the X-ray data. We describe in detail in the supplement how we did this. The analysis is independent of any knowledge of excitation power, number of retinals, absorption cross sections etc. Since it seems not have to been clear, we have now emphasized this point in the revised manuscript. We also used a very conservative approach rather underestimating the structural changes than overestimating them. Due to the depth- and orientation dependent relative laser penetration into the crystals, this number is obviously smaller than the above optical excitation at the crystal surface.

More generally, we thank the referee for his request to improve the discussion of the fraction of excited molecules since this is indeed a central point, even though extremely difficult to ascertain. Even if we restrict the discussion to the spatial laser peak intensity, the excitation conditions are strongly inhomogeneous within one crystal (X-ray propagation length \gg laser penetration depth of ~3.6 μ m), and from one crystal to another due to random crystal orientation.

We address the issue of multiphoton effects on page 27 of the response.

◆ *Some analysis of the variance in the fraction excited and the effect on the extrapolated structural changes is essential for readers to properly gage the degree of confidence in these reported structural changes. The fraction excited cannot simply be assumed at these peak powers and then use the cis-trans branching ratio to*

disentangle the differential changes in diffraction intensities from the dark controls.

The referee is right that this is a very crude approximation, but we do not think we can do better with the data available to us. We think this approximation is better than not addressing this point at all as was the case in a recent publication by Nogly et al that was published during the revision of this manuscript. Nogly et al report an isomerization angle of 90° degree for the time interval of 457 fs to 646 fs after photoexcitation without any further comment despite the fact that this configuration is energetically unstable and cannot accumulate during a time interval of 90 fs.

◆ *This point is doubly important as there are also significant variations in the light adapted bR process they did not take into account.*

Again, the referee is correct. There are actually two issues. The first is variations in the light adaption yield. These will average out and simply manifest themselves in an apparent light-adaption yield. The second is that the dark adapted form can also undergo a photocycle. The present data do not allow addressing this issue at all.

◆ *The crystal thickness is too thick on average to uniformly illuminate the entire sample volume. Most of the light is absorbed at the front of the crystal, which will block most of the light from properly adapting the back side of the crystal in the light adaption process. This point is important despite as it will affect the fraction of ground state trans and cis as an additional question on their observed structural changes.*

We agree with the referee's concern and are fully aware of the factors limiting complete light-adaptation, including the ones mentioned by the referee. The LCP embedded bR crystals spend about 5.5 seconds in the capillary section that is illuminated by the pre-illumination laser beam. The capillary is inside a metal sleeve (Supplementary Fig. 14), whose polished inner surface will reflect both the direct laser beam as well as light scattered by the circular capillary. In this manner, some light will reach also the "back" of the crystals. Given that the crystal concentration yields mainly single crystal hits, shielding of crystals is not a major concern. . We determined a light-adaptation level of 65-80 % of the crystals illuminated *in situ* in the experimental chamber (which is consistent with offline measurements performed in our laboratory). This value is based on retinal extraction and chromatographic separation results and we believe that due to the reasons pointed out by the referee, 100% light-adaptation in crystals is very difficult to achieve, including with our light-adaptation set-up. We would like to point out that in contrast to other groups (Nango et al 2016, Nogly et al 2018), we actually quantified the extent of light-adaptation.

This issue also applies to photoexcitation and is likely one reason for the low excitation yield that the referee commented on earlier (here, the reflection of the pump laser back onto the jet is not possible).

◆ *The real problem is likely the femtosecond laser excitation but the entire problem needs to be treated with some analysis of error in the structure determinations.*

As pointed out above, whenever possible, we estimated the errors.

❖ ***The new computational studies highlighting the formation of CT states***

6. The theory is a welcome contribution to the problem. It is interesting to see some ideas that there may be charge transfer processes involving the surrounding protein as part of the protein structure optimization. However, charge transfer processes depend critically on the bath as this normally involves long range correlated motions, many small motions, in medium (protein in this case) repolarization of the charge separated product state.

We provide a “model of the protein” by including the residues surrounding the retinal in our multireference-multistate QM treatment. All our theoretical conclusions are reached using the prediction of such a model. “...repolarization of the charge separated product state” is accounted for in our model by considering a rather large cluster of residues for the computations of the excited states as well as by performing excited state geometry optimization. The classical molecular-mechanical model of the protein is entirely inappropriate for addressing electronic coupling between the chromophore and the protein. Interactions with the bath (the parts of the protein and solvent missing from our model) cause de-coherence, but do not affect initial population of the CT states. So, it is not clear to us on which grounds the referee challenges our computational results.

◆ *It is difficult to understand a strong charge transfer character without a more realistic model for the protein. The wavefunction overlap at a specific site or residue will be greatly reduced by a more complete treatment of the protein. I believe this level of theory is in response to the experiment and not a prediction in advance of the experiment per se.*

The level of theory is consistent with the quantum-mechanical nature of the photoinduced ultrafast electron transfer in a protein: a large cluster model allows addressing the between the chromophore and protein on the same theoretical basis as the excitations within the chromophore. Our results demonstrate that the larger the model the larger the electronic coupling (i.e. wavefunction overlap). Specifically, the S1/CT(W86) electronic coupling at the CT(W86)-min geometry is 88 cm^{-1} in our small

cluster RPSB-Trp86-W402 (Suppl. Fig. 22a) compared to 223 cm^{-1} in our large cluster (Fig. 7a).

It is not correct that the calculations were performed in response to the experiment. In fact, We do not agree with the referee, the predictions from the calculations formed the basis of our beamtime proposal to the LCLS. This was the main reason that we got beamtime. The feedback from the Proposal Review Panel was as follows: "The team proposes to use bacteriorhodopsin (bR) from *Halobacterium salinarum* crystalized in lipidic cubic mesophase as a model system to study ultrafast processes. The pump-probe, TR-SFX experiments will explore whether or not photoinduced electron transfer from Tyr185 to the retinal protonated Schiff base is coupled to proton transfer from Asp212 on the ultrafast time scale. There was significant enthusiasm/discussion from the panel. The panel liked the hypothesis drive approach and that they have measured the crystalline bR photocycle kinetics in LCP using time-resolved spectroscopy to rationally develop a pump-probe time delay series that covers the J to K transition. It was agreed that 1.8 Å resolution and high redundancy will be critical to address their mechanistic questions. The panel expressed high confidence in the team and noted significant pressure for beamtime from many excellent proposals." Unfortunately, the latter point resulted in a reduction of the amount of beamtime that we had applied for and considered essential for the experiment.

◆ *The magnitude of the motions observed experimentally are quite large. These observations may be more in line with multiphoton ionization of retinal than charge transfer contributions to the residues.*

We address the issue of multiphoton isomerization in detail in the revised manuscript (see page 27 of the reply), in particular also the relevance of the structurally observed changes. Our spectroscopic and QM analysis strongly suggest that in bR the multiphoton effects are decoupled in time-scale from the one-photon process. Nevertheless, the experiment will have to be repeated at lower laser power in the future.

◆ *All previous experimental and theoretical work view the process to be driven by the excited state potential (with protein involvement at this level) with relaxation central or localized to the retinal chromophore. It seems the argument is for a much stronger charge transfer process not within the retinal but to the protein itself, which would then now involve polar residues. Such an effect should be seen experimentally as very different in terms of transient spectra.*

The referee is right. We address this specifically in our revised manuscript. We stress that the amount of CT within retinal in the S1 state (without retinal twisting) is reduced by the HOMO-LUMO overlap. The overlap is zero in the 90° twisted geometry (the CI geometry), therefore the amount of CT upon twisting should increase. However, our early experimental structures (prior to isomerization) do not show significant twisting of

retinal, which is consistent with a moderate increase of dipole moment (8 Debye, as compared to 12 and 19 Debye in the CT states). A change in the Trp absorption is observed immediately after photoexcitation, before retinal twisting (Schenkl et al., *Science* (2005), Leonard et al., *PNAS* (2009)). This is visible at 1554 cm^{-1} in Fig. 6b, Supplementary Fig. 4., Supplementary Fig. 5, and Supplementary Fig. 6. Within our present model, the population of CT states offers an alternative explanation for the Trp-absorption changes.

◆ *It is for this reason I am also asking for a comparison of the observed oscillations inferred from the structural changes (diffraction analysis) to that observed previously with all optical methods. Spectroscopy should definitely observe these effects in transient spectra if not the specific motions involved. The electronic states would be very strongly modulated by such a CT process in contrast to the great similarity of retinal (and analogues) photophysics in a large number of different systems (different proteins and solvents where this CT process would not occur).*

A detailed comparison of the oscillations described here and observed previously by spectroscopy is given in point 1 raised by the referee.

The processes in free retinal in solution and in different proteins are considered to differ greatly (e.g. isomerization specificity and quantum yield, highlighting the role of the protein) Given that Trp and Tyr residues are highly conserved in the active site of many rhodopsins, it is highly likely that these CT states are also present there and modulate the electronic states.

❖ **Request for off-line “power titration” experiments using TAS**

◆ *The above points cover the major concerns. The authors must go redo their control experiments as this is a critical point. This is not an unreasonable request as it requires no additional XFEL beam time. This study can be done in labs currently set up within this collaboration for such studies. It is not sufficient to do all optical pump-probe transient absorption studies on their crystals to assert that based on the similarity to the observed dynamics (low power) that these results (at orders higher peak power) can be compared to solution studies under closer to physiological conditions. The lower power optical studies can be used to argue that photobiology is not strongly affected by the crystal packing forces but not then extrapolate to high power. (Given the initial motions are localized in the protein, this point was known before hand.) In terms of going to higher peak powers, it is well known that at peak powers above 100 GW/cm^2 that multiphoton ionization and other artifacts begin to contribute for bR. (High power optics for lasers are usually specified for 10 GW/cm^2 to put this issue into perspective.) If so, any inferred motions are likely dominated by ionization effects as these motions involved are much larger amplitude and would correspond to the main motions observed within their resolution. For this reason, the authors need to compare the transient excited state spectra using a global fit (decay associated spectra). This*

requirement is now standard practice. It will show if the excited states are indeed the same under the much higher excitation conditions used in the XFEL experiments. If there is a pronounced red shift for example in solution phase studies under the same peak power (as a necessary comparison), it means there is even nonresonant ionization of waters to create solvated electrons and given the relative small absorption cross section would indicate a problem. This process would be occurring in addition to fully resonant ionization of the retinal chromophore in Br. If the decay associated spectra agree at the two peak power regimes then there is sufficient proof that the structural changes can be related to biologically relevant processes. I would further like to stress this condition is for the short time dynamics (0-10 ps range of most interest in this work). Germinate recombination will occur on longer time scales to excited state potentials and will be complicated to sort out.

As described in detail on page 28 we addressed all these points and identified a parallel photo-reaction (S-TPA process) in bR connected with the Trp charge transfer state.

◆ *If there is a problem, here it should be noted that the fraction of systems undergoing ionization would dominate the statistics for the short time scales in recovering the structural changes relative to the very small changes involving the retinal chromophore (a point again relevant to the fraction excited issue on the extracted structural changes). The structural changes in the case of ionizing the retinal or even protein residues would be expected to involve larger changes of the parent and solvation of a full charge (parent ion + e). These motions would be larger than the localized motions of the twisted I, J, K intermediates. The authors need to redo their control studies under the exact same laser excitation conditions and do a detailed comparison of the excited state spectra to show they are measuring biologically relevant processes under the excitation conditions used. This comparison should be made for both solution phase and similar crystal size distributions (re:light adapted bR problem).*

As mentioned below (page 28), it was unfortunately not possible to perform the spectroscopic investigations at the same laser excitations conditions used for the SFX experiments. We performed a spectroscopic power titration and compared the dynamics of bR purple membranes and microcrystals at similar excitation conditions. We show that the retinal dynamics of bR purple membranes and bR microcrystals are highly similar within the signal-to-noise ratio of the experiments. This is presented in Fig. 1c, Supplementary Fig. 5, and Supplementary Fig. 6. We explicitly characterize multiphoton processes by spectroscopy and QM. These results indicate that it is unlikely that multiphoton effects contribute to the sub-ps structural features observed by crystallography.

◆ *Simply put, the excitation conditions used in these experiments are not acceptable in femtosecond spectroscopy. There was a long history early on in the development of picosecond lasers that involved problems with excessive peak power and multi-photon absorption artifacts. It is a normal requirement to show that you are in the linear domain with respect to absorption and independence of spectral changes to ensure one is observing the excite state processes of interest. The same standards need to be applied to femtosecond time resolved x-ray diffraction experiments to avoid these pitfalls.*

This work is truly exceptional. I will fully support publication in Nature Communications if the peak power and fraction excited issues can be resolved.

We fully agree with the referee's final word regarding the amount of work provided in a multi-partner collaboration and bundled in the new manuscript, and with his request for performing off-line transient absorption experiments as a function of peak power density ("power titration") with the aim of mimicking as much as possible the unusually large values used for the TR-SFX. Most importantly, these "power titration" experiments, carried out for both PM suspensions and bR micro-crystals led us to a full reassessment of the multi-photon excitation conditions and of the identification of a new excitation scheme, a sequential 2-photon absorption (S-TPA). As a consequence, the revised manuscript is written in full awareness of these particular excitation conditions, as reflected by the 2nd sentence in the abstract: *"We use time-resolved crystallography ... to follow the structural changes in multiphoton-excited bR from 300 femtoseconds to 10 picoseconds,"*

The results of these new experiments as well as of the QM calculations are presented in

- The new figure 1d
- The new figs. 6 and 8
- The new Supp. Figs. 4,5,6, 7 and 9
- The new sub-section "*High power photoexcitation*"
- The modified sub-section "*Quantum chemistry analysis of the retinal interactions with the protein*"

In summary, we performed mid-IR and UV/VIS transient absorption spectroscopy (TAS) experiments, for peak intensities up to 1/3 of the ones used in TR-SFX. They reveal a new sequential two-photon process, which leads to the excitation of a Trp residue, most likely Trp86. This is in particular evident from the transient mid-IR spectra, showing the bleach of Trp vibrations (Fig. 6). The computed energy level diagrams (Fig. 8) rationalizes this process and explains new photo-induced bands observed in the UV/VIS TAS spectra. Indeed, Trp relaxes electronically within ~20ps and is at the origin of induced absorption bands in the UV/VIS TAS spectra, explaining the observed red-

shift in the 600-700 nm region, previously reported by other authors (new ref. Kraack et al., (2013)). Also, the observed increase of excited state lifetimes are qualitatively in agreement with previous reports (e.g. Schmidt et al., (2005)).

Most importantly, and against expectations raised by the paper of Polland et al, (1986), the relaxation of Trp does apparently NOT occur via energy transfer onto the RPSB. This would lead to re-excitation of retinal and to additional 13-cis isomers. However, as unambiguously shown by the vibrational bands of this species in the mid-IR, there is NO additional slow formation of the 13-cis species on a ~20ps time scale. Trp most likely relaxes in the excited state manifold including intermolecular CT states into non-productive pathways, which would still need to be identified in future studies. Therefore, and in agreement with the TR-SFX data, isomerization occurs only on a 0.5-0.6 ps time scale. Also, the observed oscillations are a manifestation of the coherent dynamics caused by strong coupling between the interconverting S_1 and CT state, and therefore unlikely to originate from the S-TPA. The crystal volume fractions excited in the 1-photon limit apparently dominate the prominent sub ps structural changes found by TR-SFX.

Note that we were also able to rule out the potential ionization of water for $P \leq 180$ GW/cm² (Supp. Fig. 7).

In conclusion, we are very grateful to the referee for having pointed out our error, it was an honest mistake. We hope that we have now addressed the issue of multi-photon excitation in sufficient detail that the referee can now recommend publication of our work.

Reviewers' comments:

Reviewer #2 (Remarks to the Author):

In our opinion, the authors have gone to great lengths to address our concerns. We feel that the new contributions significantly strengthen the conclusions. Furthermore, the revision also resolves the many misunderstandings we had in the first version. We therefore believe this manuscript will make a very valuable and timely contribution to our understanding of the photochemistry in bacteriorhodopsin and thus strongly support its publication.

We have a very few trivial comments on the revision that the authors may address in their final version and/or proof.

The meaning of the word 'coherent' seems to be used both in the context of the coherent vibrations, as well as coherent superposition between diabatic electronic states (e.g. the retinal-S1 and retinal-Trp-CT states). In the latter case, we assume that the authors refer to the adiabatic electronic surface connecting the diabatic retinal-S1 and retinal-Trp-CT states, on which a coherent nuclear wavepacket evolves back and forth until damped due to coupling to other vibrational modes?

Some of the authors are also thanked in the acknowledgements (i.e. KH), which is bit double.

In Figure S10, perhaps it is possible to include a small label to water402 in the pictures?

Is it possible to include the Cartesian coordinates of the QM-optimized cluster models in the SI?

There are a few typos, and omissions, e.g.:

Text:

Page1: crystallograpically -> crystallographically

page 11, energy is -> energy of

page 15: "Given that" appears twice.

SI:

page 14: replaces->replaced

page 15: minim-size -> minimal size

page 15: As we used -> We used, or remove "for which"

page 19: extracellular. -> extracellular side?

Page 33: select -> selected

Reviewer #3 (Remarks to the Author)

The revised manuscript "Three-dimensional view of ultrafast dynamics in photoexcited bacteriorhodopsin" by Kovacs et al. has responded dutifully to the request by the reviewer to do a power dependence to illustrate that the work was done under conditions dominated by nonlinear multiphoton processes – not relevant to biological functions. They have done a very careful study of the power dependence in which the major concern of the reviewer has been born out. However, the authors still insist they are observing biologically relevant structural changes due to 1-photon processes. For this reason, this paper is not acceptable for publication in Nature Communications nor in any journal until the major errors made in the use of high field excitation are corrected. The problem is not one of communication that can be corrected with clarification but a propagation of the same error under the guise of a new, previously undetected, excited state absorption process – unique to bacteriorhodopsin (bR). They insist, with no controls or experimental evidence to support their claim, that the transition to excited trp86 is the only multiphoton channel. They make this assertion even in the light of the enormous congestion of possible pathways for the obvious multiphoton processes, all of which would affect their conclusions. They are in the strong field excitation limit in which a broad distribution of multiphoton excited states is involved that are by definition not biologically relevant. It is an extraordinary claim that of all the multiple possible excitation pathways at such high excitation that they insist that they are still primarily observing 1-photon processes relevant to the biological response of bR. They are by definition not observing biologically relevant processes. My hope was that once they saw the effects on their spectrum at the high excitation conditions used, they would realize this is a serious problem and then discuss the structural changes they observe as consequences of high excitation and not extrapolate to biological processes. (Retinal is poised to undergo isomerization with relatively minor inputs of energy above the barrier. It should be no surprise if you deposit 10x more energy than needed one will see retinal has isomerized.)

The authors continue to make glaring errors assuming concepts from optical absorption in the weak field limit. The authors are again making unsubstantiated claims without proper controls or understanding of the processes involved. Under no circumstances can they be allowed to assert that the absorption of light near dielectric breakdown limits is at all biologically relevant. I am disappointed by their continued assertion that they are measuring biologically relevant processes despite all the evidence to the contrary in their own data and the very fact that they did little to educate themselves on the high field limits under which they did these experiments. They are clearly selectively interpreting their data and literature references without critically thinking about the results and prior reports in relation to their experiment. The confidence they have in the theoretical predictions to support their approach is unwarranted. Once you see the agreement between theory and experiment involves an artifact it should create some doubt - not new assertions of yet previously unknown effects to assert they are selectively observing biologically relevant processes.

I hope the authors can appreciate that the reviewer went to great lengths to explain the control experiments that they needed to do to point out the problem. This review saved them from what would have been a rather embarrassing error that would come to light in the near future. It needs to be fully appreciated that the reviewer also gave up a good fraction of his own

current research on this problem to point this out to them. It should be clear that they never would have done the proper controls and laser power dependence without explicit instructions from the last review on what was needed to be done. The peak power problem, including the red shift in the transient spectra at high excitation peak powers, and maximum peak power limit to stay within linear response were exactly as predicted by the reviewer. The authors are in no position to now assert they know what multiphoton processes are exclusively occurring and I would hope they will respect the advice below.

With the power dependence (requested by the reviewer), it is no longer a debatable point whether multiphoton absorption is occurring. I will point out again that the major problem with respect to multiphoton absorption is that it undoubtedly leads to larger amplitude motions (larger driving force from higher lying excited state) that are NOT biologically relevant by definition, and most likely some motions completely alien to the biological process. The very notion the authors would propose some new, previously, undetected multiphoton channel to assert they are only observing biological relevant 1-photon processes, with no controls, is not acceptable. This problem strictly excludes them from publishing this work in the present form.

I will provide a general response to their rebuttal to further reinforce the problem and provide some guidance that I hope they will follow to get the story right.

1. Multiphoton vs 1-photon contributions to their signal.

The authors argue exclusively on the basis of the QM calculations that there is a highly selective charge transfer band involving the transition from the retinal S1 excited state to the retinal-trp86 CT state calculated and no other possible pathways for multiphoton exist. They then argue that, since this highly excited state only contributes to relaxation within the excited trp86, it does not affect the retinal photochemistry and what is observed in the time resolved x-ray diffraction changes (from dark reference structures) is the 1-photon photoisomerization process that they link to biological function.

There are several problems with this assertion not the least that they did no control experiment to check on their QM calculation. These are the following:

- Figure 6a. The power dependence clearly demonstrates the onset of nonlinear absorption above 30 GW/cm². At this excitation, they document a change in absorption at 1513 cm⁻¹ as a direct measure of the fraction of cis photoproduct. The onset of nonlinear absorption sets in at approximately an excitation fraction of 30% (consistent with the estimated number of photons per retinal in the supplemental material). If one looks at the signal level at 180 GW/cm², the highest excitation done for the laser power dependence, the signal is approximately the same as at 30 GW/cm² and a factor of 50% less than at 60 GW/cm². The signal should have been at least as large as the signal at 60 GW/cm² which would be the saturation excitation. This single measurement definitely shows that there are other processes occurring that are depleting the 1-photon channel for absorption. The amount of cis photoproduct is going down not up in relation to other excitation channels. This measurement

was done at 180 GW/cm², which is a factor 3 less than the peak power used for the time resolved x-ray studies. If the nonlinear multiphoton process is being dominated by 2-photon processes, then the fraction of retinal excited by 2-photon processes would be 10x larger than the maximum peak power done in the laser titration. Taking 30% to be the fraction excited at 30 GW/cm², there would only be 3% of the retinal excited via 1-photon channels at the 500 GW/cm² used in the experiments. This control study shows the multiphoton absorption is not a minor channel but is the major photon absorption pathway. (It should be noted that 3-photon channels will make this situation worse.)

- Figure 6c. The changes in excited state spectra as a function of power clearly show that at 130 GW/cm², there is a new absorption band that now dominates. It is not a small fraction but the dominant fraction. The differential absorption change signaling the formation of cis-bR peaks at 630 nm for their PM suspension at excitation within linear 1-photon absorption. As the power increases, there is a red shift as pointed out by the reviewer. Please note, the depletion of the signal attributable to cis formation. It is nearly completely suppressed by the multiphoton generated photoproducts. There is only a minor inflection at the cis photoproduct maximum. This spectral change can be analyzed to determine the fraction cis formed at 130 GW/cm² but it is easy to see it is at least a factor of 5-10 decreased in amplitude as the sign for the absorption change is the same for the cis photoproduct and the new multiphoton product states. This change is observed for 130 GW/cm². Again at 500 GW/cm², there would be >10x (at least 16x) more multiphoton processes than 1-photon channels observed at 130 GW/cm². This data alone shows that their assignments to 1-photon processes is an error.

- The transient absorption band that appears at 670 nm has been assigned to a trp86 state through a speculative CT transition. Inspection of this band shows it to be too narrow to be an excited state transition. The excited state of the putative trp CT state would have transitions to the continuum states near the ionization threshold (trp S1 level is 4.5 eV and transitions at 670 nm from S1 take it close to ionization), which are very short lived and would give very broad S1-Sn transitions. The observed absorption band at 670 nm has comparable oscillator strength and bandwidth to ground state retinal. This assignment could easily be checked by exciting isolated trp or other proteins with trp and no chromophores to interfere with the trp excited state transitions to determine the excited state spectrum of trp. The shape of this band is unusual and is indicative of multiple but shifted transitions. One other likely possible assignment is to multiphoton ionized retinal or to a distribution of cis photoproducts created by much higher energy relaxation pathways than through the conical intersection (CI) in the S1 surface connecting the S1_trans to the S0_cis electronic surfaces. Here the authors need to appreciate that just like in the S1 surface there will be conical intersections all over the place in the excited state surfaces and rather than a single well defined pathway to the cis photoproduct there will be a very large number of other possible pathways and branching ratios, none of which are biologically relevant.

- the putative trp86 CT state is not excited trp and would involve some form of electron transfer and excited state relaxation with ultimately electron transfer to lower lying levels. The net

effect of this 2-photon channel is depletion (as with all multiphoton processes) of the ground retinal population from contributing to 1-photon processes. This is in fact observed.

- The peak power used in the time resolved experiment needs to be more carefully determined in the future. The authors state that they made a mistake in their original description of the experiment and left out that the beam was displaced by 25 microns down stream of the LCP injector flow to avoid light scattering from affecting the degree of trans retinal formed in their light adaption protocol. This error and another error in radius over diameter meant their peak powers were a factor of 2 down from their first submission. The problem is that they still need to take into account that they are using a 1 micron x-ray beam focus so the peak power is locally higher in their probe region than estimated assuming an effective uniform area for the Gaussian beam focus. The excitation conditions are closer to 1 TW/cm². They need to do these power dependences up to 1 TW/cm² to fully appreciate the seriousness of the problem. These studies are needed in any case to defend future applications for XFEL beam time. As it is, they only studied the power dependence up to 1/4th to 1/3rd the peak power used for the transient optical and transient IR data, respectively. This is not close. It is an order of magnitude down in population transfer via the different 1-photon and 2-photon channels. The excitation is high enough that 3-photon ionization is certainly occurring and may be a significant excitation channel. Without a detailed power dependence with a known 3-photon ionization state, it is not possible to determine relative 1-photon, 2-photon, and 3-photon absorption channels.

There is no question that multiphoton processes are dominating the excitation process and to such an extent that 3-photon ionization may be the most significant channel. Here it needs to be fully appreciated that the E field at the excitation used during the time resolved x-ray experiments is within a factor of 3 of complete ionization, i.e., being comparable to the E field binding the electrons to the atoms. It is this physics that leads to dielectric breakdown which appears universally around 10 TW/cm². (The E field at 1 TW/cm² is approx. 1/3rd the value at 10 TW/cm²). There is no way this excitation level can be treated as if one is in the weak perturbation limit as done by the authors. The electric field is enough to modulate significantly energy levels and mix new states as well as couple all possible resonant pathways involving all excited states up to the continuum or ionization limit. The use of a QM theory to calculate solely one pathway involving a putative CT state only involving the trp86 residue is simply wrong. The authors need to more carefully consider their own energy level diagram (figure 8). In the strong perturbation limit of these experiments, all possible electronic transitions in this figure will be resonantly enhanced to provide multiphoton pathways. The transition dipole moments are all within factors of 3 of one another and under strong field will mix. These will illicit the same photochemistry as they all involve electronic levels above the isomerization barrier as this is the labile photochemistry of retinal.

The data provided by the authors unequivocally demonstrates that the multiphoton channels dominate the absorption process at the excitation levels they used. This result illustrates the entire problem with the way the fraction excited is obtained from fitting crystallographic data, a main point made by the reviewer in the last review (see further comment on this below).

2. Oscillations in the data/frequencies and amplitudes.

It was pointed out in the last review that the extremely large amplitude oscillations (.2 Å) involving the surrounding amino acid residues of retinal would have strongly modulated the transient optical spectrum and these modes have never been observed for bR. The response to this point was *“In conclusion, the 90-120 cm⁻¹ oscillations we report for motion of the PSBR are consistent with the data of (Hou et al. Chem Phys Lett (2003), Wand et al., JACS (2011), Liebel et al., Phys Rev Lett (2014)) for bR, and with similar values reported for Anabaena Sensory rhodopsin in (Wand et al., J Phys Chem (2013), Wand et al., JACS (2011)).”*

I checked these references. None of the reported frequencies match the reported frequencies from the time resolved x-ray studies. It is only acceptable to compare bR data in any case. The closest was the 135 cm⁻¹ mode by Ruhman et al (ref. 23 in the paper). The dominant modes are observed at 170 cm⁻¹ and 210 cm⁻¹ for the better resolved experiments. These frequencies do not in fact match the observations, but more to the point, the low frequency modes that can be assigned to excited state displaced modes (i.e. modes involved in the reaction) are all damped with decay times between 200 – 300 fs. The data shown in figure 4 are clearly not damped on this time scale but more like 1 ps, within the limited dynamic range and sampled structural dynamics. The observed modes are definitely not modes observed under true 1-photon excitation conditions. If they were, they would not have been able to resolve them given the limited SNR and damping times of 200 – 300 fs of these modes. The authors were not critically looking at this data and were looking primarily to justify their assertion the observations were related to 1-photon processes. Instead, it should be clear to them that the amplitude is so large they would have to be seen in optical studies. The fact that the frequencies are very close to the impulse period given by their laser excitation pulse (145 fs) and that they appear to be under damped means these motions are not involved in the reaction nor related to the biologically relevant 1-photon process. The statement about the damping is very clear for Figure 4b (notably trp86, and all data shown in the right panel of figure 4b). The very fact that the Trp86 is supposed to have a CT band associated with it based on the QM calculations, the noted amplitude modulation in the distance shown in the figure would have to lead to significant modulation in the CT band. With typical electron wavefunction overlap decaying with beta factors on the order of 1Å, .2Å modulation would be a big effect on CT contribution to the retinal transitions.

The alternative explanation is multiphoton ionization that leads to longer range forces that are not dissipatively damped (and would be longer lived) as they would be if they were involved in relaxation along a reaction pathway. Longer range motions are needed to relax to screen the charge generated or electron recombination. This suggestion is speculative but could be checked by observing the Trp UV absorption band as it is sensitive to local E fields. The effect (re: new frequencies and observed large amplitudes) could simply arise from processes elicited from reaction pathways at higher energies and therefore larger driving forces. It is clear from the discussion above that the structural dynamics are being driven by higher lying electronic states/multiphoton processes and would be consistent with the observation of previously

unobserved frequencies/modes - than those observed under true 1-photon excitation conditions.

3. Fraction excited/fitting from crystallographic data

The above should point out the problem with this approach. The authors agreed in their rebuttal that the fraction excited affects the extrapolated amplitudes of the various motions from the differential change in diffracted intensities in relation to the reference unexcited or dark bR structure. The problem is that the light adapted fraction of bR in the trans configuration is at best 80%. The laser power dependence clearly shows suppression of the 1-photon excitation channel as more retinal is chewed up by 2-photon and 3-photon excitation channels. They observe 50% suppression at 180 GW/cm². Since the multiphoton scales at least quadratically, this would mean 10x more multiphoton processes relative to what is already a saturated 1-photon channel at about 30 GW/cm². The residual 1-photon channel could not be more than a few percent at 500 GW/cm² and as noted above the laser titration with a Gaussian probe of ½ the size the excitation needs to go to 1 TW/cm² to emulate the actual conditions used with a 1 micron x-ray probe spot size.

The use of the crystallographic data to fit to an effective average fraction excited is only acceptable in the true linear response regime where all crystallographic changes are linearly proportional, i.e. within linear response so that all structural changes are identical in all unit cells independent of fraction excited. In this nonlinear regime with a broad distribution of multiphoton reaction processes and amplitudes, this procedure can not be defended. The amplitudes will be off and certainly by the very nature of fitting a strongly excited system is not representative of biologically relevant motions.

4. Observation of long time structural intermediates

One of the main arguments put forward by the authors and the paper of Nogly et al (Science 2018) is that the observation of the M structural intermediate on the ms time scale coincides with previous measurements and this coincidence indicates they are measuring in the 1-photon regime. The previous measurements to which they compare (and also in this work) were done at low peak power and the intermediate structure is the accepted one. (Some of these studies were still conducted at very high photons per retinal levels.) There is a problem with this argument in that this observation is at the long time limit when there is enough time to relax to the global minimum no matter what pathway. The system is poised to undergo trans-cis isomerization. If one considers stilbene or azobenzene and locally had 3 photons absorbed within the solvation shell, the local temperature would be on the order of 2000 degrees, it will undergo thermally driven isomerization never mind photoisomerization. The issue will be the quantum yield. The high peak power creates excited states well above the barrier for isomerization that is quite unnatural. Some channels will simply involve excess heat to drive the process thermally from a very hot trans ground state. Also some will involve upper excited states and even ionization with subsequent recombination. It would be no surprise if you excited any system undergoing photoisomerization with excitation pulses at the S0 -> S1

transition or $S_0 \rightarrow S_n$ transitions, one would still see isomerization at both excitation conditions. All would photoisomerize but with different amplitudes of motion and different branching ratios. This is all this statement about observing the long lived intermediate proves. This discussion on this incorrect assertion again should highlight the importance of preparing well defined initial excited states that coincide with the biologically optimized states if you want to understand the biology.

5. Comparison to theory

The authors have put more faith in the theoretical predictions than warranted, especially at high powers. They have not critically evaluating the predictions in light of what I hope now is an obvious high power artifact in their data. The use of CASSCF and associated means of including the surrounding residues is the highest level calculation done to date. However, the selection of residues was done based on already known structural changes from longer time studies (I would expect anyway). It is hardly exhaustive. The claims of CT contributions only apply to the ground state structures in which some degree of electron borrowing or better polarization induced effects is occurring and contributes to the opsin shift. The very fact that the experiments have been done in the multiphoton regime and the authors still claim the results perfectly agree with theory shows that they went into to prove the theory not test the theory and look critically for differences. The theory heavily biased the data interpretation.

It should be noted equation 8 as part of the theory has been incorrectly applied. This relation is for the weak coupling, nonperturbative regime. It is basically a higher order correction to Fermi Golden Rule, which allows taking into account non-Condon effects. This relationship should have been applied to all possible CT states involving all residues within wavefunction overlap of the retinal (again see figure 8). The Mulliken-Hush relationship was really intended to estimate electronic coupling for electron transfer from experimental spectra in which the spectra show broaden red shifted transitions when the Donor and Acceptor are in close enough contact for significant wave function overlap as oppose to separated. As a Donor and Acceptor are brought closer and closer and provided there are electronic levels that are resonant for electron exchange, there will be a splitting in energy related to the electronic coupling, the inverse of which gives the electron transfer or hopping time between Donor and Acceptor. As this coupling exceeds the nuclear fluctuation time scale, one passes to the adiabatic limit for electron transfer. Once this coupling further approaches electronic decoherence timescales (in excess of a few 100 cm^{-1}), the electron associated with the transition is delocalized over both Donor and Acceptor states and one sees the onset of red shifted absorption band, that CT band, that is normally very broad. This effect would pertain to all residues within wavefunction overlap to the same degree as trp86 and it should be noted all will have levels resonant to the same degree as trp86. These other residues would also have to be seriously considered. Most problematic is that there was such a strong assertion that the trp86 was the only allowed 2-photon channel. This is clearly wrong and brings to question the theory and level of objectivity. These calculations were all done without the laser field present. They would need to be redone in the strong field limit in which the Hamiltonian includes explicitly the presence of a strong field corresponding to the actual conditions. The authors will find the presence of the strong

field mixes all the states and more states than those shown in figure 8. To state that there is only a single 2-photon absorption channel is clearly incorrect. In any case, the necessary level of rigor to make such a strong assertion is absent and shows the authors are not critically evaluating the theory but taking it as established.

The experimental conditions in no way correspond to those assumed for their QM treatment and any comparison to their theoretical predications is suspect. Their arguments that the excellent agreement with theory and use of their QM treatment to assert they are observing 1-photon processes further brings to question their objectivity. I hope the realization that they are really looking at multiphoton induced structural changes will sharpen their critical evaluation of the theoretical predictions. There are many features of the theory that do not represent the real system and one should not expect other than qualitative agreement, especially in terms of coupling of the retinal to the surrounding residues and waters as these motions would include correlated motions of the protein not included in these calculations. The dynamics localized to the retinal should be well reproduced. However, it is impossible from the resolution of these experiments to distinguish what would be a multiphoton photoisomerization process involving an upper excited level or the biologically relevant S1 state.

6. Suggested Ways Forward

The above points must be taken into account before publishing this work. There are a number of outstanding features that clearly merit publication once the correct story is pulled together and these strongly perturbing excitation conditions are not sold as being biologically relevant.

First, the authors are to be applauded for their work on the laser power dependence and dutifully responding to the reviewer's request. However, it must be stated that this power dependence should have been done before beam time to know exactly what laser excitation peak power is the maximum allowed for 1-photon processes and any claim to biologically relevant processes. It will be important for the authors to publish the laser power dependence on the crystals and other controls as a stand alone paper to point out the problem in fs time resolved x-ray diffraction studies in general. All the published XFEL work to date is at peak powers where it was known to be in the strong field, multiphoton dominated, regime. This point will not be fully appreciated by the XFEL user community until the authors publish their findings in this regard.

I would add here that this work should give them access to much more XFEL beam time rather than less. It is crucial to get the story right and not cave into pressure to try to publish high impact papers on putative biological relevance or other extraordinary claims for atomically resolving fundamental processes associated with linear (1-photon) response. There is no impact if the work is shown to be wrong years later (if any one could ever get beam time to repeat it) and leads science down wrong paths. Given the limited beam time for XFELs, there is a tendency to crank up the laser power to get signal, however, this is causing huge problems in the reported science. This work can make a major contribution to correcting the science. The

authors' work should be strongly promoted to ensure that all proposed beam times have done laser power dependences as part of the application process. It should be required homework for any further allocation of beam time. The problem with high peak power artifacts being promoted as extraordinary observations needs to stop. In this regard, the authors will do an enormous service to the community

The careful analysis of the excitation conditions and correction to their errors in the previous reported peak power estimates are all much appreciated. The work of Nogly et al (Science 2018) did not take into account the peak power problem in their data analysis. They asserted based on scattering that they were way below the peak power problem. However, the scattering is only large in the far field and the peak power is hardly affected within 1-2 photon scattering lengths, which is where the crystals are located. The authors' work on ray tracing and measuring the scattering is a very important contribution to demonstrate the argument by Nogly et al is incorrect. (*This work was done at even higher peak powers. So rather than being scooped by this prior work the authors are now in a position to get the story right and be poised for more beam time to go to lower excitation levels that are biologically relevant.) Again, this would be part of the above suggested stand alone publication on just the laser power dependence.

The authors have clearly observed structural changes. These changes, however, can in no way be related to biologically relevant processes. Nature has optimized the photophysics to create conditions such that very little dissipation is involved to direct the excited state to the target product state. The main questions here are the relative few modes needed for steering the photochemistry in relation to an enormous number of modes that are orthogonal to this channel that would otherwise lead to entropic losses in transducing the absorbed photon energy into biological functions (and reduced quantum yield or efficiency). In the present case, rather than a subtle tap that sinks a nail to hold steadfast a new structure, you have taken a sledge hammer. The remarkable thing is that the bR system still undergoes the same reaction from a number of different starting conditions above the barrier. I think the authors can discuss the importance of their work in this regard so it is clear the amplitudes and motions observed are not biologically relevant but show the system is geometrically constrained so that the photoisomerization process is conserved. The biologically relevant issue is to determine the amplitudes of the motions involved in steering under truly relevant biological conditions to give an idea of the degree of optimization of the reaction forces. The authors should definitely publish the time resolved diffraction analysis but be completely up front about the structural dynamics they have observed. The pitfalls of using a single parameter for the fraction excited needs to be clearly stated so that readers realize the motions and amplitudes are nonlinearly convoluted to a distribution and not uniquely defined motions for the isomerization. Most important, it should be clearly stated that they are observing strongly perturbed structural dynamics.

I appreciate the great effort the authors went to get this story straight. I hope they can equally appreciate the reviewer's work on this paper. The previous review illustrated a major flaw in the experiment and saved them from reporting on peak power artifacts. There are still

important new insights to be gained from this work and most important this work will establish rigorous experimental protocols for the field. The impact of this work will be greater for it.

Reviewer #2 (Remarks to the Author):

In our opinion, the authors have gone to great lengths to address our concerns. We feel that the new contributions significantly strengthen the conclusions. Furthermore, the revision also resolves the many misunderstandings we had in the first version. We therefore believe this manuscript will make a very valuable and timely contribution to our understanding of the photochemistry in bacteriorhodopsin and thus strongly support its publication.

We thank the referee for her/his positive comments and are glad that she/he is satisfied with the way we addressed the points.

We have a very few trivial comments on the revision that the authors may address in their final version and/or proof.

The meaning of the word 'coherent' seems to be used both in the context of the coherent vibrations, as well as coherent superposition between diabatic electronic states (e.g. the retinal-S1 and retinal-Trp-CT states). In the latter case, we assume that the authors refer to the adiabatic electronic surface connecting the diabatic retinal-S1 and retinal-Trp-CT states, on which a coherent nuclear wavepacket evolves back and forth until damped due to coupling to other vibrational modes?

This is correct. Indeed, we suggest that the oscillation observed might be of the electronic origin, that is, a coherent nuclear wavepacket evolves back and forth on the adiabatic electronic surface connecting the diabatic S1 and CT state energy minima.

Some of the authors are also thanked in the acknowledgements (i.e. KH), which is bit double.

In Figure S10, perhaps it is possible to include a small label to water402 in the pictures?

Is it possible to include the Cartesian coordinates of the QM-optimized cluster models in the SI?

There are a few typos, and omissions, e.g.:

Text:

Page1: crystallograpically -> crystallographically

page 11, energy is -> energy of

page 15: "Given that" appears twice.

SI:

page 14: replaces->replaced

page 15: minim-size -> minimal size

page 15: As we used -> We used, or remove "for which"

page 19: extracellular. -> extracellular side?

Page 33: select -> selected

We are very grateful for the careful reading of the entire manuscript; we corrected the errors and added a statement about data availability.

Reviewer #3

***Introduction.** The revised manuscript "Three-dimensional view of ultrafast dynamics in photoexcited bacteriorhodopsin" by Kovacs et al. has responded dutifully to the request by the reviewer to do a power dependence to illustrate that the work was done under conditions dominated by nonlinear multiphoton processes – not relevant to biological functions. They have done a very careful study of the power dependence in which the major concern of the reviewer has been born out. However, the authors still insist they are observing biologically relevant structural changes due to 1-photon processes. For this reason, this paper is not acceptable for publication in Nature Communications nor in any journal until the major errors made in the use of high field excitation are corrected.*

It is what it is; there is no way around the fact that the SFX experiment was performed in the multiphoton excitation regime. This is now spelled out clearly from the very beginning. The crystallographic data are interpreted as apparent conformations, representing the averaged conformation of all light-induced species. The structural dynamics is interpreted as evolution of a highly excited system, with electronic and vibrational relaxation ultimately leading to isomerization. The spectroscopic and quantum chemical studies characterize what changes between single and multiphoton excitation, providing new insight in both processes. We think that the manuscript largely follows the suggestions given by the referee (Point 6 below,

“Suggested ways forward”), although we came to this insight independently. It is nice to see that the referee’s view and ours converge. Hopefully the referee agrees.

The problem is not one of communication that can be corrected with clarification but a propagation of the same error under the guise of a new, previously undetected, excited state absorption process – unique to bacteriorhodopsin (bR). They insist, with no controls or experimental evidence to support their claim, that the transition to excited trp86 is the only multiphoton channel. They make this assertion even in the light of the enormous congestion of possible pathways for the obvious multiphoton processes, all of which would affect their conclusions. They are in the strong field excitation limit in which a broad distribution of multiphoton excited states is involved that are by definition not biologically relevant. It is an extraordinary claim that of all the multiple possible excitation pathways at such high excitation that they insist that they are still primarily observing 1-photon processes relevant to the biological response of bR. They are by definition not observing biologically relevant processes.

We fully agree with the referee that we are not following the biologically relevant reaction.

My hope was that once they saw the effects on their spectrum at the high excitation conditions used, they would realize this is a serious problem and then discuss the structural changes they observe as consequences of high excitation and not extrapolate to biological processes. (Retinal is poised to undergo isomerization with relatively minor inputs of energy above the barrier. It should be no surprise if you deposit 10x more energy than needed one will see retinal has isomerized.) The authors continue to make glaring errors assuming concepts from optical absorption in the weak field limit. The authors are again making unsubstantiated claims without proper controls or understanding of the processes involved. Under no circumstances can they be allowed to assert that the absorption of light near dielectric breakdown limits is at all biologically relevant. I am disappointed by their continued assertion that they are measuring biologically relevant processes despite all the evidence to the contrary in their own data and the very fact that they did little to educate themselves on the high field limits under which they did these experiments. They are clearly selectively interpreting their data and literature references without critically thinking about the results and prior reports in relation to their experiment. The confidence they have in the theoretical predictions to support their approach is unwarranted. Once you see the agreement between theory and experiment involves an artifact it should create some doubt - not new assertions of yet previously unknown effects to assert they are selectively observing biologically relevant processes.

I hope the authors can appreciate that the reviewer went to great lengths to explain the control experiments that they needed to do to point out the problem. This review saved them from what would have been a rather embarrassing error that would come to light in the near future. It needs to be fully appreciated that the reviewer also gave up a good fraction of his own current research on this problem to point this out to them. It should be clear that they never would have done the proper controls and laser power dependence without explicit instructions from the last review on what was needed to be done. The peak power problem, including the red shift in the transient spectra at high excitation peak powers, and maximum peak power limit to stay within linear response were exactly as predicted by the reviewer. The authors are in no position to now assert they know what multiphoton processes are exclusively occurring and I would hope they will respect the advice below.

We very much appreciate the insight, expertise, effort and time of the referee. As detailed below we did not mean to imply that we know which multiphoton processes are occurring exclusively. The high excitation intensity-induced red shift of the peaks has been reported several times in the bR literature. So far, the molecular origin was unknown; we now offer an explanation in terms of a sequential 2-photon absorption process.

With the power dependence (requested by the reviewer), it is no longer a debatable point whether multiphoton absorption is occurring. I will point out again that the major problem with respect to multiphoton absorption is that it undoubtedly leads to larger amplitude motions (larger driving force from higher lying excited state) that are NOT biologically relevant by definition, and most likely some motions completely alien to the biological process. The very notion the authors would propose some new, previously, undetected multiphoton channel to assert they are only observing biological relevant 1-photon processes, with no controls, is not acceptable. This problem strictly excludes them from publishing this work in the present form. I will provide a general response to their rebuttal to further reinforce the problem and provide some guidance that I hope they will follow to get the story right.

We thank the referee for recognising the advances made due to the spectroscopic power titration experiments. We agree that the amplitude of the motions may be different, but that does by *no means rule out* the presence of oscillations in 1-photon conditions. To really access to what extend our observations occur under 1-photon excitation, further TR-SFX experiments need to be performed in the future. Results from spectroscopic experiments on solutions, suspensions or thin films performed under lower excitation intensities can simply not be directly translated into diffraction experiments on crystals. We are glad that the referee recognises that this work is essential for the future direction of the field.

1. Multiphoton vs 1-photon contributions to their signal.

The authors argue exclusively on the basis of the QM calculations that there is a highly selective charge transfer band involving the transition from the retinal S1 excited state to the retinal trp86 CT state calculated and no other possible pathways for multiphoton exist. They then argue that, since this highly excited state only contributes to relaxation within the excited trp86, it does not affect the retinal photochemistry and what is observed in the time resolved x-ray diffraction changes (from dark reference structures) is the 1-photon photoisomerization process that they link to biological function. There are several problems with this assertion not the least that they did no control experiment to check on their QM calculation.

We disagree with the referee. We do NOT state “*no other possible pathways for multiphoton exist*”. Instead, our QM calculations show that these CT states are likely to be populated upon single photon excitation and furthermore, can contribute to the sequential two photon processes. We identify a possible pathway of a two-photon excitation process leading to excitation of a tryptophan residue that is strongly electronically coupled to the retinal, but we do not claim that this is the only possibility. Experimentally, however, the mid-IR data clearly show the dominant activation of Trp’s via the predicted two-photon process. **[REDACTED]**

These are the following:

- Figure 6a – new Figure 2. The power dependence clearly demonstrates the onset of nonlinear absorption above 30 GW/cm². At this excitation, they document a change in absorption at 1513 cm⁻¹ as a direct measure of the fraction of cis photoproduct. The onset of nonlinear absorption sets in at approximately an excitation fraction of 30% (consistent with the estimated number of photons per retinal in the supplemental material). If one looks at the signal level at 180 GW/cm², the highest excitation done for the laser power dependence, the signal is approximately the same as at 30 GW/cm² and a factor of 50% less than at 60 GW/cm². The signal should have been at least as large as the signal at 60 GW/cm² which would be the saturation excitation.

The referee has a good point. We therefore checked the spectra again. The photoproduct signal decreases due to newly emerging overlapping vibrational bands (New Supplementary Fig. 1d,e). Due to this, it is not straightforward to decide at what laser intensity the absorption becomes nonlinear. Nevertheless, it is very clear that the photoproduct yield decreases. This was also published previously by Prokhorenko et al who determined a limit of 80 GW/cm² for the linear regime.

This single measurement definitely shows that there are other processes occurring that are depleting the 1-photon channel for absorption. The amount of cis photoproduct is going down not up in relation to other excitation channels. This measurement was done at 180 GW/cm², which is a factor 3 less than the peak power used for the time resolved x-ray studies. If the nonlinear multiphoton process is being dominated by 2-photon processes, then the fraction of retinal excited by 2-photon processes would be 10x larger than the maximum peak power done in the laser titration. Taking 30% to be the fraction excited at 30 GW/cm², there would only be 3% of the retinal excited via 1-photon channels at the 500 GW/cm² used in the experiments. This control study shows the multiphoton absorption is not a minor channel but is the major photon absorption pathway. (It should be noted that 3-photon channels will make this situation worse.)

We agree that at 500 GW/cm² multiphoton absorption is the major pathway and that single photon photoproduct production is negligible.

(Just for completeness: The pump laser beam will be reflected at the interface vacuum- LCP stream, both at the front and the backside of the LCP stream. We estimate 6 % reflection, corresponding to ~0.3 uJ. In this case, for an average crystal diameter of 5.9 um, we would expect that 66 % of the (already excited) retinals absorb one photon. The isomerization reaction induced by the reflected light would be delayed by ~ 0.7 ps, assuming the crystal to be located in the middle of the 0.1 mm diameter stream and an index of refraction of ~1.4. This scenario might contribute to phase III in Fig. 7.)

- Figure 6c – Now figure 2a. The changes in excited state spectra as a function of power clearly show that at 130 GW/cm², there is a new absorption band that now dominates. It is not a small fraction but the dominant fraction. The differential absorption change signalling the formation of cis-bR peaks at 630 nm for their PM suspension at excitation within linear 1-photon absorption. As the power increases, there is a red shift as pointed out by the reviewer. Please note, the depletion of the signal attributable to cis formation. It is nearly completely suppressed by the multiphoton generated photoproducts. There is only a minor inflection at the cis photoproduct maximum. This spectral change can be analysed to determine the fraction cis formed at 130 GW/cm² but it is easy to see it is at least a factor of 5-10 decreased in amplitude as the sign for the absorption change is the same for the cis photoproduct and the new multiphoton product states. This change is observed for 130 GW/cm². Again at 500 GW/cm², there would be >10x (at least 16x) more multiphoton processes than 1-photon channels observed at 130 GW/cm². This data alone shows that their assignments to 1-photon processes is an error.

We agree.

- The transient absorption band that appears at 670 nm has been assigned to a trp86 state through a speculative CT transition. Inspection of this band shows it to be too narrow to be an excited state transition. The excited state of the putative trp CT state would have transitions to the continuum states near the ionization threshold (trp S1 level is 4.5 eV and transitions at 670 nm from S1 take it close to ionization), which are very short lived and would give very broad S1- Sn transitions. The observed absorption band at 670 nm has comparable oscillator strength and bandwidth to ground state retinal. This assignment could easily be checked by exciting isolated trp or other proteins with trp and no chromophores to interfere with the trp excited state transitions to determine the excited state spectrum of trp. The shape of this band is unusual and is indicative of multiple but shifted transitions. One other likely possible assignment is to multiphoton ionized retinal or to a distribution of cis photoproducts created by much higher energy relaxation pathways than through the conical intersection (CI) in the S1 surface connecting the S1_trans to the S0_cis electronic surfaces. Here the authors need to appreciate that just like in the S1 surface there will be conical intersections all over the place in the excited state surfaces and rather than a single well defined pathway to the cis photoproduct there will be a very large number of other possible pathways and branching ratios, none of which are biologically relevant.

- the putative trp86 CT state is not excited trp and would involve some form of electron transfer and excited state relaxation with ultimately electron transfer to lower lying levels. The net effect of this 2-photon channel is depletion (as with all multiphoton processes) of the ground retinal population from contributing to 1-photon processes. This is in fact observed.

We thank the referee for his/her suggestion. We have clarified the discussion of the Trp ESA accordingly. The referee's prediction of a "very broad" Trp ESA can be checked against published data. We did this and thereby justify the assignment of the absorption bands at 460 and 670 to a Trp ESA, on the basis of Leonard et al., *Phys Chem Chem Phys* **12**, 15744-15750 (2010), where Trp's ESA was identified at 450 and 550-620 nm in aqueous solution and when bound to a peptide, but partially water exposed. The different spectral positions are most probably due to the lower dielectric constant of the protein environment. A precise comparison of the spectral width is however not possible since, in Leonard et al., 2010 ., the Trp ESA sits on a dominant broad unstructured background due to solvated electrons. It is important to note that even for the highest excitation powers used, the bR data do not show solvated electrons, meaning multi-photon ionisation of the protein or the buffer is not detected.

As shown in Supplementary Figures 8,9 (SI), both two-photon induced signatures decay on a 15-20 ps time scale, in agreement with their assignment to Trp ESA. Note that thereafter (48 ps spectra in Suppl. Fig. 8), the K difference spectrum remains. Its slight red shift (10-20 nm) seen for the highest excitation powers may be due to incomplete vibrational relaxation at these delays.

Concerning the alternative assignment as a spectrum representing a “distribution of *cis* photo-products created by much higher energy relaxation pathways than through the conical intersection (CI)...”, one may envisage that this is indeed the case for the highest laser intensities used for TR-SFX, but not for the two-photon absorption regime for which the discussed transient UV/VIS spectra were obtained. The reduced *cis* photoproduct C=C stretch signal at 1513 cm⁻¹ leads us to disfavour this scenario, but rather to support a non-productive relaxation pathway back to a hot all-*trans* ground state, in line with the reduced isomerisation quantum yield, reported by Prokhorenko et al.

Note also that photo-ionisation of PSBR is unlikely even with 3 photon absorption (≈ 7 eV), since the ionisation energy of the PSBR cation is rather in the 10 eV range.

- The peak power used in the time resolved experiment needs to be more carefully determined in the future. The authors state that they made a mistake in their original description of the experiment and left out that the beam was displaced by 25 microns down stream of the LCP injector flow to avoid light scattering from affecting the degree of trans retinal formed in their light adaption protocol. This error and another error in radius over diameter meant their peak powers were a factor of 2 down from their first submission. The problem is that they still need to take into account that they are using a 1 micron x-ray beam focus so the peak power is locally higher in their probe region than estimated assuming an effective uniform area for the Gaussian beam focus.

We did not assume an effective uniform area. We calculated the power of the Gaussian beam at the position ($r = \text{center}-25 \mu\text{m}$) of the 1 μm X-ray beam (see below).

The excitation conditions are closer to 1 TW/cm². They need to do these power dependences up to 1 TW/cm² to fully appreciate the seriousness of the problem. These studies are needed in any case to defend future applications for XFEL beam time. As it is, they only studied the power dependence up to 1/4th to 1/3rd the peak power used for the transient optical and transient IR data, respectively. This is not close. It is an order of magnitude down in population transfer via the different 1-photon and 2-photon channels. The excitation is high enough that 3-photon ionization is certainly occurring and may be a significant excitation channel. Without a detailed power dependence with a known 3-photon ionization state, it is not possible to determine relative 1-photon, 2-photon, and 3-photon absorption channels.

Concerning the power density calculations, we have indeed made errors in the initial submission and are extremely grateful to the referee for catching them. In our revised version, we had included a detailed table with all parameters and considerations that go into the calculation (Supplementary Table 7). It clearly states there that the displacement of 25 μm is between the pump laser and X-ray focus, not between the beam and the LCP injector. This results in a power density reduced by a factor of 0.6 from the peak power density value, thus we are **not measuring at 1 TW/cm²**, but at around 600 GW/cm². Taking into account the experimentally determined reduction by scattering at the various surfaces, the sample is exposed to 500 GW/cm². The referee has used this power density above (page 3) in his arguments; we are surprised why he/she argues here with the 1 TW/cm² from the previous version of the manuscript.

We agree that the spectroscopic experiments do not reach the intensity range of the TR-SFX experiments; we were not able to go any higher due to technical limitations. We would again like to emphasize that even if we did a perfect spectroscopic power dependence knowing the

3-photon ionization state and could determine the relative 1-photon, 2-photon, and 3-photon absorption channels, it would not directly translate to the TR-SFX conditions due to the inhomogeneous excitation. The only solution is to perform TR-SFX experiments at different excitation intensities. We hope that our work will highlight the importance of power titrations to the community and that facilities will allocate additional beamtime to such experiments in order to avoid SFX pump probe experiments in the multiphoton excitation regime. We fully agree that is a real problem in the field.

There is no question that multiphoton processes are dominating the excitation process and to such an extent that 3-photon ionization may be the most significant channel.

Photo-ionisation of PSBR is unlikely even with 3 photon absorption (≈ 7 eV), since the ionisation energy of the PSBR cation is in the 10 eV range.

Here it needs to be fully appreciated that the E field at the excitation used during the time resolved x-ray experiments is within a factor of 3 of complete ionization, i.e., being comparable to the E field binding the electrons to the atoms. It is this physics that leads to dielectric breakdown which appears universally around 10 TW/cm². (The E field at 1 TW/cm² is approx. 1/3rd the value at 10 TW/cm²).

As pointed out above, the referee misunderstood the geometry of our setup. While our excitation intensity is high, it is not 1 TW/cm² but 600 GW/cm². Taking into account the contribution of scattering, the power density is 500 GW/cm² at the sample/X-ray interaction position.

There is no way this excitation level can be treated as if one is in the weak perturbation limit as done by the authors. The electric field is enough to modulate significantly energy levels and mix new states as well as couple all possible resonant pathways involving all excited states up to the continuum or ionization limit. The use of a QM theory to calculate solely one pathway involving a putative CT state only involving the trp86 residue is simply wrong. The authors need to more carefully consider their own energy level diagram (figure 8 [Now figure 4]). In the strong perturbation limit of these experiments, all possible electronic transitions in this figure will be resonantly enhanced to provide multiphoton pathways. The transition dipole moments are all within factors of 3 of one another and under strong field will mix. These will illicit the same photochemistry as they all involve electronic levels above the isomerization barrier as this is the labile photochemistry of retinal.

See above concerning power density and excitation conditions.

The referee will agree with us that the pathway calculations are complex and highly demanding. We characterized the CT states of two prominent electron donors characterized by the largest wavefunction overlap or electronic coupling (Trp86 and Tyr185). This pathway is important because we (and previous studies) obtained experimental evidence that it plays a role in bR multiphoton excitation. We do not agree with the referee that it is wrong to calculate one of the possible pathways. It is an important start and we do not claim that it is the only channel. This is now explicitly stated. The pathway investigated is consistent with previous experimental data and with our own assignment of the IR spectral dynamics. Even at the highest intensities, the mid-IR data do not show any sign of long-lived (> 50 ps) bleach bands in the vibrational bands of PSBR or amino acids, other than the K form, which would be indicative of photo-ionisation.

The data provided by the authors unequivocally demonstrates that the multiphoton channels dominate the absorption process at the excitation levels they used. This result illustrates the entire problem with the way the fraction excited is obtained from fitting crystallographic data, a main point made by the reviewer in the last review (see further comment on this below).

We agree, multiphoton channels most likely dominate the crystallographic data.

2. Oscillations in the data/frequencies and amplitudes.

It was pointed out in the last review that the extremely large amplitude oscillations (.2 Å) involving the surrounding amino acid residues of retinal would have strongly modulated the transient optical spectrum and these modes have never been observed for bR. The response to this point was “In conclusion, the 90-120 cm⁻¹ oscillations we report for motion of the PSBR are consistent with the data of (Hou et al. Chem Phys Lett (2003), Wand et al., JACS (2011), Liebel et al., Phys Rev Lett (2014)) for bR, and with similar values reported for Anabaena Sensory rhodopsin in (Wand et al., J Phys Chem (2013), Wand et al., JACS (2011)).”

I checked these references. None of the reported frequencies match the reported frequencies from the time resolved x-ray studies. It is only acceptable to compare bR data in any case. The closest was the 135 cm⁻¹ mode by Ruhman et al (ref. 23 in the paper). The dominant modes are observed at 170 cm⁻¹ and 210 cm⁻¹ for the better resolved experiments.

We are glad that the referee now agrees that low frequency modulations of the transient optical spectrum have indeed been observed. Frequency values of the oscillations in Fig. 6 and 8 contain the error estimate– which is on the order of ± 10 -20 cm⁻¹. Further, the analysis of the oscillations is handicapped by the way the data was binned, which introduces further errors (Supplementary Figure 17). We therefore consider the values of $\sim 80 - 130$ cm⁻¹ as consistent with literature, in fact surprisingly close. Had we planned to look for oscillations in particular and had we had more beamtime, we would have collected more data to reduce these errors and determine the oscillation frequencies more accurately. Follow-up work will focus exactly on this, but using single photon excitation conditions.

These frequencies do not in fact match the observations,

We disagree, see above

but more to the point, the low frequency modes that can be assigned to excited state displaced modes (i.e. modes involved in the reaction) are all damped with decay times between 200 – 300 fs. The data shown in figure 4 [now figure 8] are clearly not damped on this time scale but more like 1 ps, within the limited dynamic range and sampled structural dynamics. The observed modes are definitely not modes observed under true 1-photon excitation conditions. If they were, they would not have been able to resolve them given the limited SNR and damping times of 200 – 300 fs of these modes.

Stimulated by the comment of the referee we revisited our analysis approach. Due to the inherent timing-jitter of the XFEL/pump pulses, one cannot collect diffraction data of a well-defined time-delay but instead collects data at nominal time-delays (e.g. $\Delta t=0.5$ ps) and sorts them afterwards using a timing tool (e.g. $0.2 \leq t_i \leq 0.8$ ps). Due to temporal drifts, the nominal time-delays Δt may cover larger or smaller actual time-ranges. Together, both effects result in an uneven distribution of the number of collected diffraction patterns $P(t_i)$ for the various chosen time-delays. Since the number of diffraction images/dataset strongly affects data quality and thus comparability of the various datasets one keeps the number of diffraction images/dataset constant. Practically, this means that the datasets corresponding to different time-delays t_1, t_2, t_3, \dots average diffraction images spanning time-windows of different width. As now described in the Methods section this results in a weighting of the observed electron densities with the binning function (e.g. if one bins the amplitudes of all time points describing half a sine function, the average temporal value is ~ 0.6 of the amplitude and not the full amplitude). Since the binning distribution is known (Supplementary Fig. 17), one can correct its effect of the amplitudes of a known function, e.g. an oscillation. This procedure allows one to get a rough lower estimate of the real amplitudes. It is clear that the amplitudes of the oscillations are not the same for the few cycles that we can observe but that they decay with time. Correcting for the binning and fitting a damped oscillation results in most case in a damping constant of several hundred fs. Obviously, this is just an indication since the accuracy is compromised by the binning, few oscillations and lack of longer time-delays.

Since we are clearly in the multiphoton excitation regime, it is most likely that we observe vibrational relaxation of the excited electronic states S_n . Nevertheless, they show vibrational coupling of retinal and surrounding amino acids and waters.

We have no way of knowing whether these oscillations also occur in the single photon excitation regime. However, our QM indicate strongest coupling in the 1-photon regime. In line with this we note, that these 1.5-2 period oscillation patterns agree fairly well with oscillatory spectroscopic transients (e.g. Hou 2003, Kraack 2011, Wand 2013) obtained under 1-photon excitation. Oscillations of amino acids are typically not visible in the TA experiments or Vis-pump IR-probe experiments, even in cases of strongly coupled amino acids to the chromophore, since the resonant response of the chromophore dominates.

The authors were not critically looking at this data and were looking primarily to justify their assertion the observations were related to 1-photon processes. Instead, it should be clear to them that the amplitude is so large they would have to be seen in optical studies.

See above. Oscillations of amino acid residues are weakly coupled in optical studies and oscillations of the chromophore modulating optical signals have been observed.

The fact that the frequencies are very close to the impulse period given by their laser excitation pulse (145 fs) and that they appear to be under damped means these motions are not involved in the reaction nor related to the biologically relevant 1- photon process.

The observed retinal oscillations match the reported oscillations within the error margin (see above). Thus, it is suggestive that the retinal oscillations display the same eigenfrequencies as the relevant 1-photon process. The more important coherent retinal oscillations at higher frequencies at 170 cm^{-1} and 210 cm^{-1} could not be observed, due to limited time resolution. Amino acid oscillations and movements induced by the instantaneous electric field change of the excited chromophore are typically not or only weakly coupled to the optical transition. Therefore, the oscillations are not or barely visible in TA experiments, and much weaker than the chromophore oscillations. In contrast, X-ray crystallography is very sensitive to amino acid oscillations and movements. The reviewer raises the point that the oscillations are due to impulsive coherent Raman scattering, resulting in oscillations around the ground state geometry, and thus not relevant for the isomerization process. Firstly, if the optical transition would allow excitation of the amino acid ground state vibrations via impulsive coherent Raman scattering, these oscillations would show up in high-precision TA experiments of bR (P.J.M. Johnson et al. Phys. Chem. Chem. Phys., 2014, 16, 21310--21320) which is not the case. Moreover, for impulsive coherent Raman scattering we would expect the phase of the oscillations at time zero to be the same for all oscillations and start around zero. This is not the case as depicted in Figure 4.

We changed the description of the oscillations in the manuscript to clarify these points.

The statement about the damping is very clear for Figure 4b [now Figure 8] (notably trp86, and all data shown in the right panel of figure 4b [now Figure 8]). The very fact that the Trp86 is supposed to have a CT band associated with it based on the QM calculations, the noted amplitude modulation in the distance shown in the figure would have to lead to significant modulation in the CT band. With typical electron wavefunction overlap decaying with beta factors on the order of 1Å , $.2\text{Å}$ modulation would be a big effect on CT contribution to the retinal transitions.

The referee probably overlooked that for Trp86, an oscillation in the torsion angle is indicated – magnitude of $\pm 1^\circ$. We believe that this relieves his concern about wavefunction overlap.

The alternative explanation is multiphoton ionization that leads to longer range forces that are not dissipatively damped (and would be longer lived) as they would be if they were involved in relaxation along a reaction

pathway. Longer range motions are needed to relax to screen the charge generated or electron recombination. This suggestion is speculative but could be checked by observing the Trp UV absorption band as it is sensitive to local E fields. The effect (re: new frequencies and observed large amplitudes) could simply arise from processes elicited from reaction pathways at higher energies and therefore larger driving forces. It is clear from the discussion above that the structural dynamics are being driven by higher lying electronic states/multiphoton processes and would be consistent with the observation of previously unobserved frequencies/modes - than those observed under true 1-photon excitation.

See above

3. Fraction excited/fitting from crystallographic data

The above should point out the problem with this approach. The authors agreed in their rebuttal that the fraction excited affects the extrapolated amplitudes of the various motions from the differential change in diffracted intensities in relation to the reference unexcited or dark bR structure. The problem is that the light adapted fraction of bR in the trans configuration is at best 80%. The laser power dependence clearly shows suppression of the 1- photon excitation channel as more retinal is chewed up by 2-photon and 3-photon excitation channels. They observe 50% suppression at 180 GW/cm². Since the multiphoton scales at least quadratically, this would mean 10x more multiphoton processes relative to what is already a saturated 1-photon channel at about 30 GW/cm². The residual 1-photon channel could not be more than a few percent at 500 GW/cm² and as noted above the laser titration with a Gaussian probe of ½ the size the excitation needs to go to 1 TW/cm² to emulate the actual conditions used with a 1 micron x-ray probe spot size.

The referee rises several points. First, it is standard crystallographic procedure to identify the fraction of “B” in a dataset collected of a crystal containing a mixture of “A+B” by subtracting the respective diffraction intensities of a crystal containing only “A”. This assumes that the two datasets are isomorphous and the data properly scaled. This is what we did. There is a significant fraction of “B” in our photoexcited bR data. The more interesting question in our case is, what does “B” correspond to (A being the dark adapted state). Clearly, it is not the single photon photoproduct.

The use of the crystallographic data to fit to an effective average fraction excited is only acceptable in the true linear response regime where all crystallographic changes are linearly proportional, i.e. within linear response so that all structural changes are identical in all unit cells independent of fraction excited. In this nonlinear regime with a broad distribution of multiphoton reaction processes and amplitudes, this procedure can not be defended. The amplitudes will be off and certainly by the very nature of fitting a strongly excited system is not representative of biologically relevant motions.

Here we do **not** agree with the referee. There are two issues in this point, the fitting of electron densities and the assignment of the resulting model to a certain functional state. The referee assumes that the former is no problem for bR in the single photon excitation regime but this is not true for the ultrafast time-points. These contain contributions of both the forward reaction and the back reaction (see Fig 1b in the previous version of the manuscript). The resolution of the bR diffraction data does not allow fitting of two models, representing these two reactions. Instead, the electron density corresponding to the photoexcited molecules is modeled with one structure that corresponds to the weighted average of the two independent conformations. This is conceptionally equivalent to the situation for multiphoton excitation. In this case, it is potentially more than two processes and thus conformations, but the electron density is fit with one model. The alternative in both cases (single/multiple photon excitation) would be not to fit the density at all, which does not help. This approach is standard crystallographic practice. The problem arises only when attaching a functional label to the fitted “apparent” model. In the single photon case, it does NOT correspond to the forward reaction, in the multiphoton case it is less straightforward what the average conformation reflects. *“of fitting a strongly excited system is not representative of biologically relevant motions”* mixes the fitting aspect (technically OK), and labeling (biologically relevant, not OK).

4. Observation of long time structural intermediates

One of the main arguments put forward by the authors and the paper of Nogly et al (Science 2018) is that the observation of the M structural intermediate on the ms time scale coincides with previous measurements and this coincidence indicates they are measuring in the 1-photon regime.

To the referee it may sound like splitting hairs, but we do not say that. There is no way that we are measuring in the 1-photon regime and we had made that point very clearly in the revised version of the manuscript, thanks to the referee. We wrote “This strongly suggests that the crystallographically observed structural changes are part of the functional reaction coordinate”. What we are saying is that the multiphoton excited states not only isomerize but eventually end up on a reaction coordinate that resembles that triggered by 1-photon absorption (instead of being e.g. ionized and not developing further). We weakened this further in the revised version. “... shows that the ultrafast intermediates formed after multiphoton absorption can proceed along the photocycle”.

The previous measurements to which they compare (and also in this work) were done at low peak power and the intermediate structure is the accepted one. (Some of these studies were still conducted at very high photons per retinal levels.) There is a problem with this argument in that this observation is at the long time limit when there is enough time to relax to the global minimum no matter what pathway. The system is poised to undergo trans-cis isomerization. If one considers stilbene or azobenzene and locally had 3 photons absorbed within the solvation shell, the local temperature would be on the order of 2000 degrees, it will undergo thermally driven isomerization never mind photoisomerization. The issue will be the quantum yield. The high peak power creates excited states well above the barrier for isomerization that is quite unnatural. Some channels will simply involve excess heat to drive the process thermally from a very hot trans ground state. Also some will involve upper excited states and even ionization with subsequent recombination. It would be no surprise if you excited any system undergoing photoisomerization with excitation pulses at the $S_0 \rightarrow S_1$ transition or $S_0 \rightarrow S_n$ transitions, one would still see isomerization at both excitation conditions. All would photoisomerize but with different amplitudes of motion and different branching ratios. This is all this statement about observing the long lived intermediate proves. This discussion on this incorrect assertion again should highlight the importance of preparing well defined initial excited states that coincide with the biologically optimized states if you want to understand the biology.

We agree with the referee. Concerning the long-lived intermediate, this is the statement we can make and we tried to make this clear now (see above). Concerning the isomerization mechanism, we have the data that we have, and this is in the multiphoton excitation regime. Interestingly, when analysing the temporal changes of apparent C13=C14 torsion angle, we can distinguish several phases that we tentatively link to electronic relaxation ($S_n \rightarrow S_1$). In addition to this reactive pathway, the multiphoton excitations seem to lead primarily into non-reactive pathways, as shown by the spectroscopy already for the lower intensity 2-photon absorption. This leads to the low occupancy and the reduced isomerisation quantum yield at highest excitation densities powers. While not the biologically relevant reaction, we may be following both electronic and vibrational relaxation of multiphoton excited retinal, ultimately leading to isomerization.

5. Comparison to theory

The authors have put more faith in the theoretical predictions than warranted, especially at high powers. They have not critically evaluating the predictions in light of what I hope now is an obvious high power artifact in their data.

Energies and properties of the low-lying states (i.e. states accessible via a single and two-photon excitation at the wavelength of the bR absorption maximum) are in the focus of our computational study. In addition, we attempted to include in our consideration states that explain spectral signatures of excited-state intermediates suggested by the time-resolved IR and VIS studies. The information provided by our computational study is critical for 1) extending models of bR beyond the properties of the chromophore; 2) establishing that excited-state calculations explicitly characterizing protein chromophore interactions (i.e.

computations of excited states that consider electron transfer between the protein and chromophore) may complement time resolved spectroscopy studies, and can be especially helpful for the assignment of the transient spectra, 3) characterizing excited-state charge transfer, which might be directly involved not only in color tuning but also in control of retinal photoisomerization. Obviously, the electronic states characterized in our study for the first time, among other excited states, underlie single-, two-photon and multiphoton dynamics.

We never stated that we addressed all relevant states (neither for single-photon nor for multiphoton processes). We do insist, however, that we characterized the CT states of two prominent electron donors characterized by the largest wavefunction overlap or electronic coupling (Trp86 and Tyr185). We suggest that the states we computed may contribute to the photodynamics of bR even in low-power regime corresponding to the physiologically relevant single-photon excitation. We do not exclude other electronic states of CT character being involved in excited-state relaxation, especially under multiphoton high-power condition. Nonetheless, the contribution of other states does not discredit the role of charge transfer from Tyr185 and Trp86 discussed in our manuscript in some detail. We also suggest that the CT-W86 state may contribute to the population of the excited W86 under two-photon excitation conditions. This suggestion, again, does not rule out other multiphoton pathways. Interestingly, our computational study revealed that reaction coordinates associated with the CT excited states involve rather extended hydrogen bonding network in the binding pocket of the retinal. Furthermore, such dynamics is expected to occur on an ultrafast timescale (may precede photoisomerization). All these predictions are based on computed energies and properties at the ground- and excited-state optimized geometries. The accuracy and reliability of our energy calculations is demonstrated by extensive energy comparison provided as Supplementary Information to our paper (Suppl. Fig. 22, Suppl. Tabs 8 and 9). We are fully aware that our computational approach has limitations. We know that large molecular systems (such as “realistic” models of the protein active sites) remain extremely challenging for advanced quantum-chemistry and density-functional theory methods. Yet, many examples in the literature demonstrate that certain properties of biomolecules can be predicted with sufficient accuracy and thus, such results contribute to our better understanding of biomolecular mechanisms. We also would like to refer to our publication record in computational characterization of the CT states in flavin-containing photoreceptor proteins. In the current study, we transferred our previous experience to characterize, for the first time, the CT states of bR.

Apparently, the referee recognizes the possibility of coupling in terms of CT in bR, however, in order to accept computational results in the present study, requests us performing theoretical simulations of the bR photodynamics at high power. We very much doubt that such simulations would be feasible at present. However, our excited-state calculations of the retinal excited states as well as of the CT states should be viewed as first step towards quantum-dynamics simulations of bR under physiological and non-physiological conditions.

The use of CASSCF and associated means of including the surrounding residues is the highest level calculation done to date. However, the selection of residues was done based on already known structural changes from longer time studies (I would expect anyway). It is hardly exhaustive.

We selected all residues that surround the protonated Schiff-base moiety and the so-called aromatic triad coupled to the two counterions. In our opinion, this is the minimum model that probably includes at least all strong interactions at the active site. Unfortunately, the referee did not specify which of the critical residues in his opinion are still missing from our model.

The claims of CT contributions only apply to the ground state structures in which some degree of electron borrowing or better polarization induced effects is occurring and contributes to the opsin shift.

This is not true according to Hayashi et al., *Biophysics* **8**, 67-72 (2012) and Yanai et al., *J Chem Theory Comput* **14**, 3643-3655 (2018). The S1 state is also affected by charge transfer according to these references and our own results presented here. Furthermore, as we demonstrated, there is a CT excited state manifold coupled to the S1 state. We note that we included the excited-state optimized geometry in our study (from these geometries we identify the CT reaction coordinate associated with large-scale motions at the active site (Suppl. Fig.7)).

The very fact that the experiments have been done in the multiphoton regime and the authors still claim the results perfectly agree with theory shows that they went into to prove the theory not test the theory and look critically for differences. The theory heavily biased the data interpretation.

Unfortunately, the referee did not specify where in the manuscript he found such a claim. This must be a misunderstanding (as several other points of criticism such as excluding other CT states based on our calculations and incorrect application of the Mulliken Hush method). In the manuscript, we indeed link the computed properties of the CT states with the previous and current experimental observations in order to highlight a possible role of these states in bR photodynamics. We did not suggest any “new theory” which needs to be proven. Rather, we complemented our experimental results characterizing retinal-protein interactions with computational results obtained by employing state of art quantum-chemistry methods with known limitations.

To the best of our knowledge, no one uses as large models treated at the QM level of theory as we do. The theory shows that the CT states are there – not only for Trp86, but also for Tyr185. There is no way to prove a CT state crystallographically, thereby test the theory. Importantly, the computational results help rationalize some of the spectroscopic results. It is known that there is a strong coupling between the interconverting CT and S₁ that may launch oscillations (e.g. according to Egorova and Domcke, *J. Photochem. Photobiol. A* (2004)) – thus, this is a result, not biased interpretation.

It should be noted equation 8 as part of the theory has been incorrectly applied. This relation is for the weak coupling, nonperturbative regime. It is basically a higher order correction to Fermi Golden Rule, which allows taking into account non-Condon effects. This relationship should have been applied to all possible CT states involving all residues within wavefunction overlap of the retinal (again see figure 8 [now figure 4]).

We would like to remind that the wavefunction overlap depends not only on the distance, but also on the orientation of the overlapping densities (electronic states associated with the same nucleus may have a zero overlap). To compare the wavefunction overlap for the Trp86 and Tyr185 electron donors, we employed the generalized Mulliken Hush method (not the same as the original Mulliken Hush method described by the referee). Moreover, we did not state that the CT states addressed by our current calculations are the only relevant/accessible CT states. As already mentioned, Trp86 and Tyr185 arguably are the most prominent electron donors to retinal.

The Mulliken-Hush relationship was really intended to estimate electronic coupling for electron transfer from experimental spectra in which the spectra show broaden red shifted transitions when the Donor and Acceptor are in close enough contact for significant wave function overlap as oppose to separated. As a Donor and Acceptor are brought closer and closer and provided there are electronic levels that are resonant for electron exchange, there will be a splitting in energy related to the electronic coupling, the inverse of which gives the electron transfer or hopping time between Donor and Acceptor.

We did not use the Mulliken-Hush method to estimate electronic coupling. We used the generalized Mulliken-Hush method (GMH). The description of the method given in Cave et al., *Chem. Phys. Lett.* **249**, 15-19 (1996) addresses the concerns of the referee. GMH provides a way to relate adiabatic energies and properties (obtained from excited-state quantum-chemical calculations) (see Cave et al., or e.g. Hsu, *Acc. Chem. Res.* 2008, 509-518) to diabatic parameters (the so-called donor and acceptor state energies and coupling) that are typically employed in theories predicting electron-transfer rates. In our paper we use couplings estimated using GMH to compare the two most prominent electron donors. Such a comparison is qualitative rather than quantitative. We do not derive any electron-transfer rates based on our estimated couplings. However, qualitatively the couplings are rather strong (which is not surprising given the interactions at the active site). The strong coupling indeed indicates an adiabatic process, which is consistent with CT contribution to the excited-state dynamics.

As this coupling exceeds the nuclear fluctuation time scale, one passes to the adiabatic limit for electron transfer. Once this coupling further approaches electronic decoherence timescales (in excess of a few 100 cm⁻¹), the electron associated with the transition is delocalized over both Donor and Acceptor states and one sees the onset of red shifted absorption band, that CT band, that is normally very broad. This effect would pertain to all residues within wavefunction overlap to the same degree as trp86 and it should be noted all will have levels resonant to the same degree as trp86. These other residues would also have to be seriously considered.

Clearly, the referee recognizes the possibility of electronic coupling in terms of charge transfer. Indeed, the wavefunction overlap (measured for instance, by the GMH electronic coupling) might be considered a key parameter. We agree that all residues with the suitable overlap/coupling will contribute. Among those residues are Trp86 and Tyr185, for which we provided, for the first time, computational evidence. It is not clear to us, how the fact that we demonstrated possible charge transfer for two prominent candidates and not for all putative electron donors (which is computationally really demanding) discredits our work. Trp182 or Tyr57 would be also possible candidates.

Most problematic is that there was such a strong assertion that the trp86 was the only allowed 2- photon channel. This is clearly wrong and brings to question the theory and level of objectivity.

We did not assert or write in our manuscript that the Trp86 pathway is the only allowed two photon pathway, but based on experimental evidence (Schenk et al., *Science* **309**, 917-920 (2005).) this is the most probable candidate. The referee questions our computational approach and objectivity based on an unjustified claim.

These calculations were all done without the laser field present. They would need to be redone in the strong field limit in which the Hamiltonian includes explicitly the presence of a strong field corresponding to the actual conditions. The authors will find the presence of the strong field mixes all the states and more states than those shown in figure 8. To state that there is only a single 2-photon absorption channel is clearly incorrect. In any case, the necessary level of rigor to make such a strong assertion is absent and shows the authors are not critically evaluating the theory but taking it as established. The experimental conditions in no way correspond to those assumed for their QM treatment and any comparison to their theoretical predications is suspect. Their arguments that the excellent agreement with theory and use of their QM treatment to assert they are observing 1- photon processes further brings to question their objectivity. I hope the realization that they are really looking at multiphoton induced structural changes will sharpen their critical evaluation of the theoretical predictions. There are many features of the theory that do not represent the real system and one should not expect other than qualitative agreement, especially in terms of coupling of the retinal to the surrounding residues and waters as these motions would include correlated motions of the protein not included in these calculations. The dynamics localized to the retinal should be well reproduced. However, it is impossible from the resolution of these experiments to distinguish what would be a multiphoton photoisomerization process involving an upper excited level or the biologically relevant S1 state.

See above: To the best of our knowledge, nobody performed this kind of calculation on a protein active site either in a strong or in the weak field limit. Not only the size of the electronic problem is prohibitive, but also the number of nuclear degrees of freedom is far too high. Further, the question arises – how to model the inhomogeneous excitation conditions in the theoretical calculations? There seems to be no simple answer to this as well. We agree that the experimental conditions do not match the theoretical ones – we are very well aware of that (see above). We do not claim that other 2-photon channels are not possible – we simply tried our best to get at least some specific information on what multiphoton effects are likely to be present.

6. Suggested Ways Forward

The above points must be taken into account before publishing this work. There are a number of outstanding features that clearly merit publication once the correct story is pulled together and these strongly perturbing excitation conditions are not sold as being biologically relevant.

First, the authors are to be applauded for their work on the laser power dependence and dutifully responding to the reviewer's request. However, it must be stated that this power dependence should have been done before beam time to know exactly what laser excitation peak power is the maximum allowed for 1-photon processes and any claim to biologically relevant processes. It will be important for the authors to publish the laser power dependence on the crystals and other controls as a stand alone paper to point out the problem in fs time resolved x-ray diffraction studies in general. All the published XFEL work to date is at peak powers where it was known to be in the strong field, multiphoton dominated, regime. This point will not be fully appreciated by the XFEL user community until the authors publish their findings in this regard.

We thank the referee for this positive rating. We completely agree that a power titration should have been done as the first step; unfortunately, we had not enough time for it.

I would add here that this work should give them access to much more XFEL beam time rather than less. It is crucial to get the story right and not cave into pressure to try to publish high impact papers on putative biological relevance or other extraordinary claims for atomically resolving fundamental processes associated with linear (1-photon) response. There is no impact if the work is shown to be wrong years later (if any one could ever get beam time to repeat it) and leads science down wrong paths. Given the limited beam time for XFELs, there is a tendency to crank up the laser power to get signal, however, this is causing huge problems in the reported science. This work can make a major contribution to correcting the science. The authors' work should be strongly promoted to ensure that all proposed beam times have done laser power dependences as part of the application process. It should be required homework for any further allocation of beam time. The problem with high peak power artifacts being promoted as extraordinary observations needs to stop. In this regard, the authors will do an enormous service to the community.

We completely agree with the referee. However, an optical power titration is only a first start; it needs to be followed by a power titration at the XFEL. Interestingly, an optical analysis was done on PYP microcrystals (Hutchinson et al 2016) establishing the excitation protocol and an optimal laser pulse energy of 0.25 mJ/mm², however the SFX experiment used 0.8mJ/mm² (5.7 GW/mm²). No explanation was provided (Pande et al 2016).

*The careful analysis of the excitation conditions and correction to their errors in the previous reported peak power estimates are all much appreciated. The work of Nogly et al (Science 2018) did not take into account the peak power problem in their data analysis. They asserted based on scattering that they were way below the peak power problem. However, the scattering is only large in the far field and the peak power is hardly affected within 1-2 photon scattering lengths, which is where the crystals are located. The authors' work on ray tracing and measuring the scattering is a very important contribution to demonstrate the argument by Nogly et al is incorrect. (*This work was done at even higher peak powers. So rather than being scooped by this prior work the authors are now in a position to get the story right and be poised for more beam time to go to lower excitation levels that are biologically relevant.) Again, this would be part of the above suggested stand alone publication on just the laser power dependence.*

We are glad to see that the referee recognizes our efforts to get this right, given the limitation of not having more XFEL beamtime. We appreciate that the referee recognizes the importance of the scattering quantification, another big issue in the field.

We also agree with the referee that it would be important to publish our work, in particular in view of the highly visible related Science publication by Nogly et al. [REDACTED] While we made a number of mistakes that the referee fortunately caught, we were at least aware of the problems, and, inspired by the referee, spent a lot of effort to address the issue experimentally after the fact. [REDACTED] We are very grateful to the referee to help us improving our manuscript. We hope that he/she now support publishing the re-revised version.

The authors have clearly observed structural changes. These changes, however, can in no way be related to biologically relevant processes. Nature has optimized the photophysics to create conditions such that very little dissipation is involved to direct the excited state to the target product state. The main questions here are the relative few modes needed for steering the photochemistry in relation to an enormous number of modes that are orthogonal to this channel that would otherwise lead to entropic losses in transducing the absorbed photon energy into biological functions (and reduced quantum yield or efficiency). In the present case, rather than a subtle tap that sinks a nail to hold steadfast a new structure, you have taken a sledge hammer. The remarkable thing is that the bR system still undergoes the same reaction from a number of different starting conditions above the barrier. I think the authors can discuss the importance of their work in this regard so it is clear the amplitudes and motions observed are not biologically relevant but show the system is geometrically constrained so that the photoisomerization process is conserved.

We fully agree with the referee. This is the focus of the re-revised manuscript. We describe multiphoton excitation, and try to get as much mechanistic information from the structural data. We speculate that we observe electronic relaxation of the highly excited retinal followed by vibrational relaxation to the ground state. While this is not the reaction that we intended to study we nevertheless consider it very interesting, also in view of previous publications investigating multiphoton excitation. In addition, by characterizing the system by spectroscopy and quantum chemistry in both the single and multiphoton regime, we have obtained new information on electronic and vibrational coupling. This insight helps explain a number of published observations.

The biologically relevant issue is to determine the amplitudes of the motions involved in steering under truly relevant biological conditions to give an idea of the degree of optimization of the reaction forces. The authors should definitely publish the time resolved diffraction analysis but be completely up front about the structural dynamics they have observed. The pitfalls of using a single parameter for the fraction excited needs to be clearly stated so that readers realize the motions and amplitudes are nonlinearly convoluted to a distribution and not uniquely defined motions for the isomerization. Most important, it should be clearly stated that they are observing strongly perturbed structural dynamics. I appreciate the great effort the authors went to get this story straight. I hope they can equally appreciate the reviewer's work on this paper. The previous review illustrated a major flaw in the experiment and saved them from reporting on peak power artifacts. There are still important new insights to be gained from this work and most important this work will establish rigorous experimental protocols for the field. The impact of this work will be greater for it.

We have reorganized the manuscript. We did not follow the recommendation of the reviewer to split the work in two parts (spectroscopy, QM vs SFX) but decided to use the former to explore both single and multiphoton excitation, followed by a crystallographic study performed using clear multiphoton excitation.

We appreciate that the reviewer recognizes our efforts to get the story correct. We are truly grateful to the reviewer for his/her continued interest in our work and the huge amount of effort and care he/she put into the reviews. The reviewer has made major contributions and we would be most happy to accept him/her as a coauthor if this were possible.

REVIEWERS' COMMENTS:

Reviewer #3 (Remarks to the Author):

I appreciate the great work that went into reevaluating the findings in light of my comments. It is critical that the peak power used in XFEL experiments be reduced to the true 1-photon regimes to make any connection to the relevant to biology (or chemistry). The very fact that the authors observe "Interestingly, as in our case, negative difference densities are much stronger than the corresponding positive densities, indicating significant disorder of the developing states compared to the initial state." This effect is expected in multiphoton processes in which there will be a distribution of potential surfaces explored in the relaxation process. Multiphoton ionization will lead to electron being trapped at numerous polar sites...giving rise to disordering effects on structure parameters as well.

I would also like to point out one further error in your discussion. The fact that the ionization threshold is 7 eV does not rule out ionization. You can't use nonresonant multiphoton arguments to rule out multiphoton ionization. You have used excitation conditions for which the 3 photon process is fully resonant AND you have also shown that you have many more photons absorbed per retinal...if 1 photon. The cross sections for $S1 \rightarrow S_n$ and $S1 \rightarrow S_+$ are similar and will be fully resonantly coupled. These transitions will be resonantly excited. The fact you do not detect solvated electrons is that the electrons do not escape the protein (escape depth is order 1 nm in the polar environs of the rhodopsin). It is not an argument against resonant 3+ photon ionization contributions. The very fact that significant (10% pump depletion) nonresonant 3photon ionization of water has been observed under peak powers lower than the conditions used makes this point clear (Prokopenko et al). You need to consider this channel in future work. These ionization effects and larger potential gradients from multiphoton excited states undoubtedly give rise to the oscillations observed. I am still not convinced about the decay period but give you that it is difficult to assess giving your binning procedure.

I would suggest you make minor modifications to your text to make it clear you can not rule out multiphoton ionization and you are dealing with a distribution of excited states. Your main conclusion is good.

Note, figure 2d. The bottom point is blue and should be black.

Final comment, I greatly appreciate the effort that went into this work to get the science right. In my opinion, this work is the most thorough study done to date and will have a major impact in the field in both getting the science right and setting standards to ensure such.

I congratulate you on a very thorough study showing that bR is amazingly wired up to take energy of any form and drive isomerization. I strongly support publishing this work in Nature Comm.

REVIEWERS' COMMENTS:

Reviewer #3 (Remarks to the Author):

I appreciate the great work that went into reevaluating the findings in light of my comments. It is critical that the peak power used in XFEL experiments be reduced to the true 1-photon regimes to make any connection to the relevant to biology (or chemistry). The very fact that the authors observe "Interestingly, as in our case, negative difference densities are much stronger than the corresponding positive densities, indicating significant disorder of the developing states compared to the initial state." This effect is expected in multiphoton processes in which there will be a distribution of potential surfaces explored in the relaxation process. Multiphoton ionization will lead to electron being trapped at numerous polar sites...giving rise to disordering effects on structure parameters as well.

I would also like to point out one further error in your discussion. The fact that the ionization threshold is 7 eV does not rule out ionization. You can't use nonresonant multiphoton arguments to rule out multiphoton ionization. You have used excitation conditions for which the 3 photon process is fully resonant AND you have also shown that you have many more photons absorbed per retinal...if 1 photon. The cross sections for $S1 \rightarrow S_n$ and $S1 \rightarrow S_+$ are similar and will be fully resonantly coupled. These transitions will be resonantly excited. The fact you do not detect solvated electrons is that the electrons do not escape the protein (escape depth is order 1 nm in the polar environs of the rhodopsin). It is not an argument against resonant 3+ photon ionization contributions. The very fact that significant (10% pump depletion) nonresonant 3photon ionization of water has been observed under peak powers lower than the conditions used makes this point clear (Prokohenko et al). You need to consider this channel in future work. These ionization effects and larger potential gradients from multiphoton excited states undoubtedly give rise to the oscillations observed. I am still not convinced about the decay period but give you that it is difficult to assess giving your binning procedure.

I would suggest you make minor modifications to your text to make it clear you cannot rule out multiphoton ionization and you are dealing with a distribution of excited states. Your main conclusion is good.

We thank the referee for appreciating our efforts to get this right. We agree with the suggestion for the minor modification and extended the manuscript accordingly at several places, mainly in the discussion by adding:

"This effect is expected in multiphoton processes in which there will be a distribution of potential surfaces explored in the relaxation process. Moreover, multiphoton ionization may result in electrons being trapped at polar sites, resulting in further disordering."

Note, figure 2d. The bottom point is blue and should be black.

We thank the referee for catching this. The dot is blue because the quantification of the photoproduct at this intensity is complicated by overlapping bands arising. We adapted the figure caption.

Final comment, I greatly appreciate the effort that went into this work to get the science right. In my

opinion, this work is the most thorough study done to date and will have a major impact in the field in both getting the science right and setting standards to ensure such.

We are grateful for the extensive effort that the referee put into shaping of this manuscript. We are glad about this very positive assessment of our work.

I congratulate you on a very thorough study showing that bR is amazingly wired up to take energy of any form and drive isomerization. I strongly support publishing this work in Nature Comm.